🔓 | **Open Peer Review** | Bacteriology | Research Article

# Genomic variability of lipooligosaccharide biosynthesis locus and sequence type among *Campylobacter jejuni* isolated from patients with Guillain-Barré syndrome

**Md. Abu Jaher Nayeem,**[1] **Shoma Hayat,**[1] **Asaduzzaman Asad,**[1] **Md. Golam Mostafa,**[1] **Shah Nayeem Faruque,**[1] **Ruma Begum,**[1] **Mosabbir Ahmed,**[1] **Yearul Kabir,**[2] **Israt Jahan,**[1,3] **Zhahirul Islam**[1]

**ABSTRACT** Lipooligosaccharide (LOS) is a crucial component of the *Campylobacter jejuni (C. jejuni)* cell membrane that mimics human gangliosides and can induce autoimmune neuropathies like Guillain-Barré syndrome (GBS). Approximately one in 1,000 *C. jejuni* infections leads to GBS, with LOS variations influencing immune response and disease induction. We aimed to investigate the sequence type and genomic variation of the LOS region of *C. jejuni* strains isolated from patients with GBS and enteritis. We isolated 38 GBS- and 24 enteritis-associated *C. jejuni* strains from Bangladesh, sequenced them on the Illumina platform, and compared their LOS regions using various bioinformatic approaches. Comparative genomic analysis between GBS-associated and enteritis-associated strains revealed that LOS classes A, B, and C were prevalent ($P = 0.0001$) in GBS-associated *C. jejuni* strains. Sequence types ST-43 and ST-2042 and clonal complexes ST-403 cc and ST-21 cc were overrepresented in GBS compared to enteritis-associated strains. All GBS-associated strains carrying LOS class C belonged to clonal complex ST-21 cc. Nucleotide variation within the LOS region differs significantly ($P = 0.005$) among GBS-associated and enteritis-associated strains. In GBS strains, phase variation (PV) within the LOS region was significantly higher ($P = 0.004$). In conclusion, the predominance of GBS-associated LOS classes, characterized by class-specific nucleotide changes and elevated phase variation, may play a critical role in the pathogenesis of GBS. However, further research on integrating DNA epigenetic analysis and metabolic profiling is needed to establish the *C. jejuni*-specific risk factors of GBS.

**IMPORTANCE** The lipooligosaccharide (LOS) biosynthesis region is the most variable and virulent genomic region of *Campylobacter jejuni*. This study addressed the critical gap in understanding the genomic features contributing to GBS by analyzing the LOS region of *C. jejuni* strains from GBS and enteritis patients. Our analysis of *C. jejuni* isolates from GBS and enteritis cases revealed genomic variations in the sequence type and LOS region, including distinct LOS classes (A, B, and C), higher prevalence of the *cst-II* gene, and significant presence of phase variability among GBS-associated *C. jejuni* strains. The findings highlight the association of *C. jejuni* LOS region in the pathogenesis of GBS. In addition, the results of this study will facilitate future comparisons by identifying genomic differences in lipooligosaccharide biosynthesis and sequence types between *C. jejuni* strains linked to GBS. This will enhance our understanding of the genomic variation of the LOS region and their role in GBS development.

**KEYWORDS** Guillain-Barré syndrome, *Campylobacter jejuni*, lipooligosaccharide, comparative genomics, phase variation

**Peer Reviewers** Willi Quino, Instituto Nacional de Salud, Lima, Peru; Isabelle Bernaquez, Laboratoire de sante publique du Quebec, Sainte-Anne-de-Bellevue, Québec, Canada

Address correspondence to Zhahirul Islam, zislam@icddrb.org.

The authors declare no conflict of interest.

See the funding table on p. 12.

*C*ampylobacter jejuni (C. jejuni) is a leading cause of gastroenteritis and has been causatively associated with the onset of peripheral autoimmune neuropathy such as Guillain-Barré syndrome (GBS) (1). This association is linked to specific genetic and structural variations within the bacterial strains, particularly within the lipooligosaccharide (LOS) locus. The LOS is the key virulence factor of *C. jejuni*, structurally mimicking human peripheral nerve gangliosides and triggering a cross-reactive immune response that can lead to the development of autoimmunity in GBS (2). However, only a very small proportion of *C. jejuni*-infected patients (one in 1,000–5,000 cases) develop GBS, and the molecular mechanisms that trigger autoreactivity are still poorly understood (3). In Bangladesh, *C. jejuni* is recognized as the most prevalent pathogen associated with GBS (4). Previous studies have reported that *C. jejuni* LOS loci classes A, B, and C, as well as the serotype *C. jejuni* HS:23 and clonal complex ST-403 cc, were associated with a high frequency of *C. jejuni*-associated GBS (5–7). Moreover, no studies have conducted a comparative sequence-based approach to examine genomic variations within the LOS region of *C. jejuni*.

Previously, a comparative genotyping approach revealed remarkable diversity among the sequence types (STs) and clonal complexes (CCs) of various *C. jejuni* strains isolated from patients with enteritis and patients with GBS in Bangladesh (5). This diversity highlights the genetic variability of *C. jejuni* strains and their different associations with gastrointestinal and neurological diseases (8). Cohort-based studies on GBS-associated *C. jejuni* cases have indicated that genomic variation in the LOS region of *C. jejuni* may be responsible for the endemicity of GBS in Bangladesh (4, 5). These studies suggest that specific genetic differences within the LOS locus contribute to the high prevalence of GBS in this region, underscoring the need for further research into these genomic variations. Biochemical and structural analyses of the LOS outer core oligosaccharides in *C. jejuni* have identified sialylated moieties that closely resemble several human nerve gangliosides (2, 9), such as GM1, GD1a, and GQ1b at the molecular level. The sialyltransferase gene *cst-II* influences the addition of neuraminic acid to LOS, potentially triggering the immune system to produce cross-reactive antibodies that contribute to the development of GBS (2, 9–12). During infection, LOS-induced antibodies may cross-react with nerve gangliosides, triggering complement activation and leading to the onset of GBS (13).

*C. jejuni* LOS biosynthesis region displays considerable genetic variation among strains, featuring notable differences in the arrangement of genes which encode LOS carbohydrate moieties and their linkages. The LOS gene locus undergoes phase fluctuations, leading to structural diversity among strains, particularly in the outer oligosaccharide region (14–16). The LOS biosynthesis region in *C. jejuni* NCTC 11168 reference genome, spanning genes Cj1131c(*gal*E) to Cj1151c (*rfa*D) with low GC content, is highly variable across strains and plays a key role in ganglioside mimicry (14, 17–19). A cross-reactive antibody response is induced in nerve tissue due to the molecular similarity between *Campylobacter* LOS and nerve gangliosides; however, the precise pathophysiology of post-*Campylobacter* neuropathy, like GBS, is unclear (10). The variations in LOS class, gene alterations, mutations, and mechanisms, such as phase variation (PV) within the LOS locus, contribute to structural variations in the ganglioside mimics (14, 15, 20). Gilbert et al. reported that two types of *cst-II* gene alleles cause the translated enzyme to express either threonine (Thr) or asparagine (Asn) at position 51[st], resulting in monofunctional or bifunctional activity with one or two sialic acids (20).

The acute motor axonal neuropathy (AMAN) variant of GBS has an unusually high frequency of occurrence in Bangladesh, which is associated with the presence of pathogenic GM1 or GD1a antibodies in serum and preceding *C. jejuni* infections (4, 19). Previously, Islam et al. reported the dominance of clonal complex ST-403 cc and concordance between LOS class B and ST-403 cc complex among GBS-associated *C. jejuni* strains isolated from Bangladesh (5). The genetic heterogeneity of *C. jejuni* strains linked to GBS has been extensively studied in the developed world. However, little is known about the significant diverse genomic variability of the LOS region of *C. jejuni*

strains associated with GBS in low- and middle-income countries. Therefore, the present study aimed to explore genomic variations within the LOS region and perform sequence typing of *C. jejuni* isolated from patients with GBS and enteritis (ENT) in Bangladesh.

## RESULTS

### Distribution of *C. jejuni* sequence types and clonal complexes

A total of 15 different sequence types were found within the 38 GBS-associated strains. Among them, ST-43 (16%, 6/38) and ST-2042 (13%, 5/38) were more prevalent (Fig. 1A and B). Within the enteritis-associated *C. jejuni* strains, we identified 19 distinct sequence types out of the 24 strains analyzed. Specifically, ST-2042 was present in three strains (13%), while ST-985, ST-986, and ST-2109 were each found in two strains (9%). The remaining strains exhibited a diverse range of sequence types, contributing to a total of 17 unique sequence types among the analyzed strains (Fig. 1B). Clonal complexes ST-403 cc (29%) and ST-21 cc (16%) were more prevalent in GBS-associated *C. jejuni* strains. Five types of clonal complexes (GBS: ENT; ST-403 cc [11:3], ST-42 cc [3:2], ST-362 cc [3:1], ST-22 cc [3:1], and ST-45 cc [1:2], respectively) were identified among both GBS-associated and enteritis-associated *C. jejuni* strains (Fig. 1C). Two types of clonal complexes ST-41 cc (2/38) and ST-61 cc (1/28) were found only among GBS-associated *C. jejuni* strains. The clonal complex of 8 GBS-associated *C. jejuni* strains and 9 enteritis-associated *C. jejuni* strains remained unassigned.

### Characterization of LOS classes and phylogenetic analysis of the *C. jejuni* LOS region

Comparative genomic analysis between GBS-associated and enteritis-associated strains revealed that LOS classes A, B, and C were significantly prevalent ($P = 0.0001$) among *C. jejuni* strains isolated from patients with GBS (GBS: 32/38, ENT = 6/24) (Fig. 2). LOS classes A and B locus were found in 32% (12/38) and 37% (14/38) of GBS-associated *C. jejuni* strains compared to 17% (4/24) and 13% (3/24) of enteritis-associated *C. jejuni* strains. LOS class C locus was found in 16% (6/38) of GBS-associated *C. jejuni* strains, whereas no LOS class C locus was found among enteritis-associated *C. jejuni* strains. The LOS class of two GBS-associated *C. jejuni* strains remained undetermined, whereas three other LOS classes (LOS class F [$n = 1$]; LOS class M [$n = 2$]; and LOS class W [$n = 1$]) were identified among 4/38 GBS-associated *C. jejuni* strains. Among enteritis-associated *C. jejuni* strains, LOS class G was present in 17% (4/24) strains, LOS class I in 8% (2/24) strains, LOS class W in 8% (2/24) strains, LOS class H in 4% (1/24), and LOS class Z in 4% (1/24) strains. Moreover, the LOS class locus of 29% (7/24) enteritis-associated *C. jejuni* strains remained undetermined (Fig. 1A and 2). Phylogenetic analysis of the LOS region revealed a phylogenetic relationship based on LOS classes rather than the isolation sources (GBS or enteritis). LOS classes B and A strains were clustered (Cluster_1) within the same branch, indicating a closer phylogenetic relationship between these groups (Fig. 2). Most LOS regions corresponding to locus class A were clumped within a specific quadrant of the phylogenetic tree. All LOS class C sequences were clustered within the same clade (Cluster_3) phylogenetic tree. LOS class G (Cluster_2), LOS class I, and LOS class H were also clustered in separate clades and were found only in the enteritis-associated strains of *C. jejuni*.

### Comparison of gene alignment within the LOS region

Comparison between LOS regions extracted from GBS-associated and enteritis-associated *C. jejuni* strains was performed to determine deletions, insertions, or genomic rearrangements in the LOS biosynthesis locus of *C. jejuni*. The comparative analysis of LOS regions from different *C. jejuni* strains revealed hypervariability based on gene patterns between LOS regions, but no genes or regions were identified within LOS that were unique to GBS-associated isolates (Fig. 3A and B). However, the genes in the LOS region of GBS-associated strains showed a distinct pattern comparable to that of

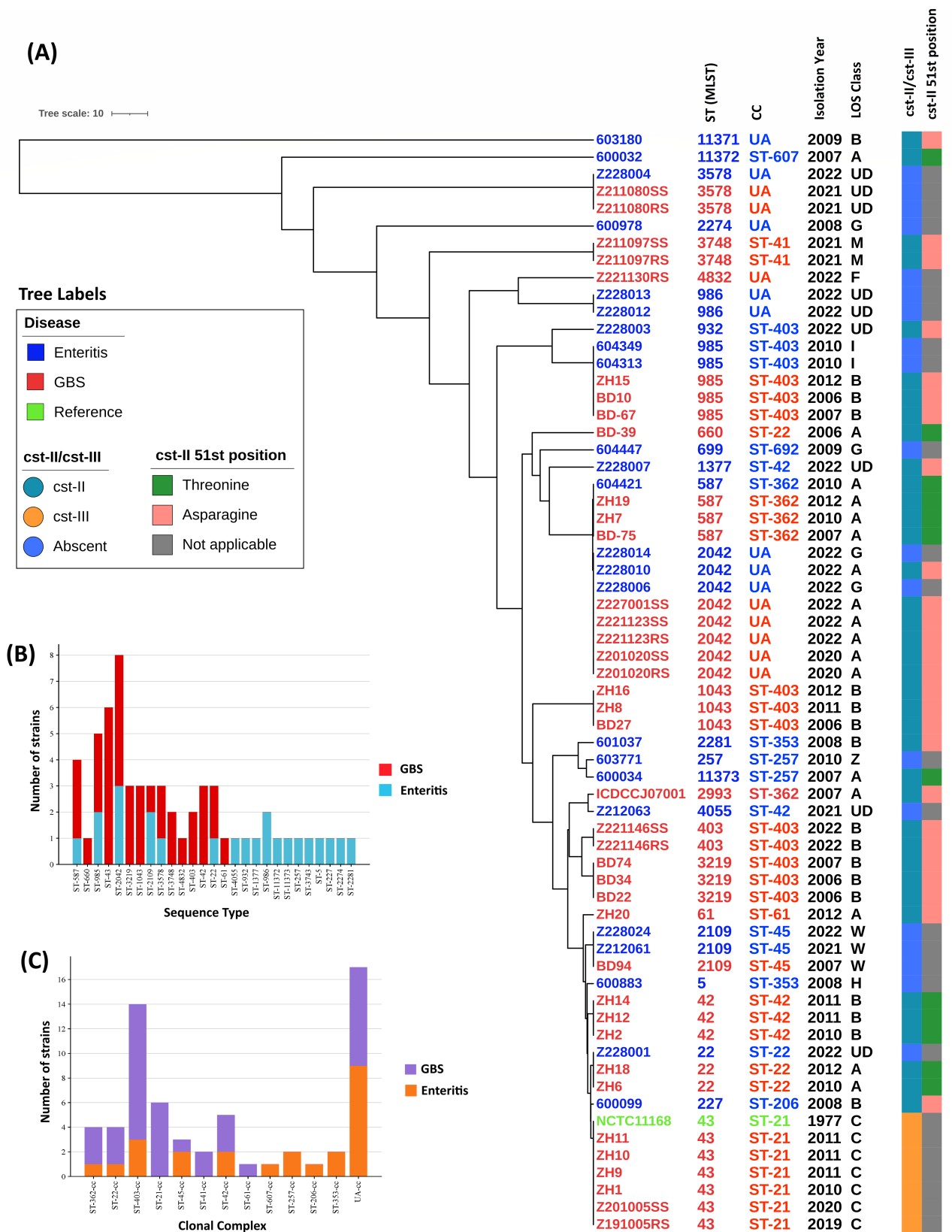

**FIG 1**  (A) Phylogenetic tree based on concatenated nucleotide sequence loci of *Campylobacter jejuni* allele from PubMLST database based on seven MLST loci. The tree also includes metadata, such as clonal complex, year of isolation, LOS class, sialyltransferase genes (*cst-II/cst-III*), and the 51st position of *cst-II*. (B) Bar chart of sequence type (ST) diversity among GBS-associated and enteritis-associated *C. jejuni* strains. (C) Bar chart of clonal complex (CC) diversity among GBS-associated and enteritis-associated *C. jejuni* strains.

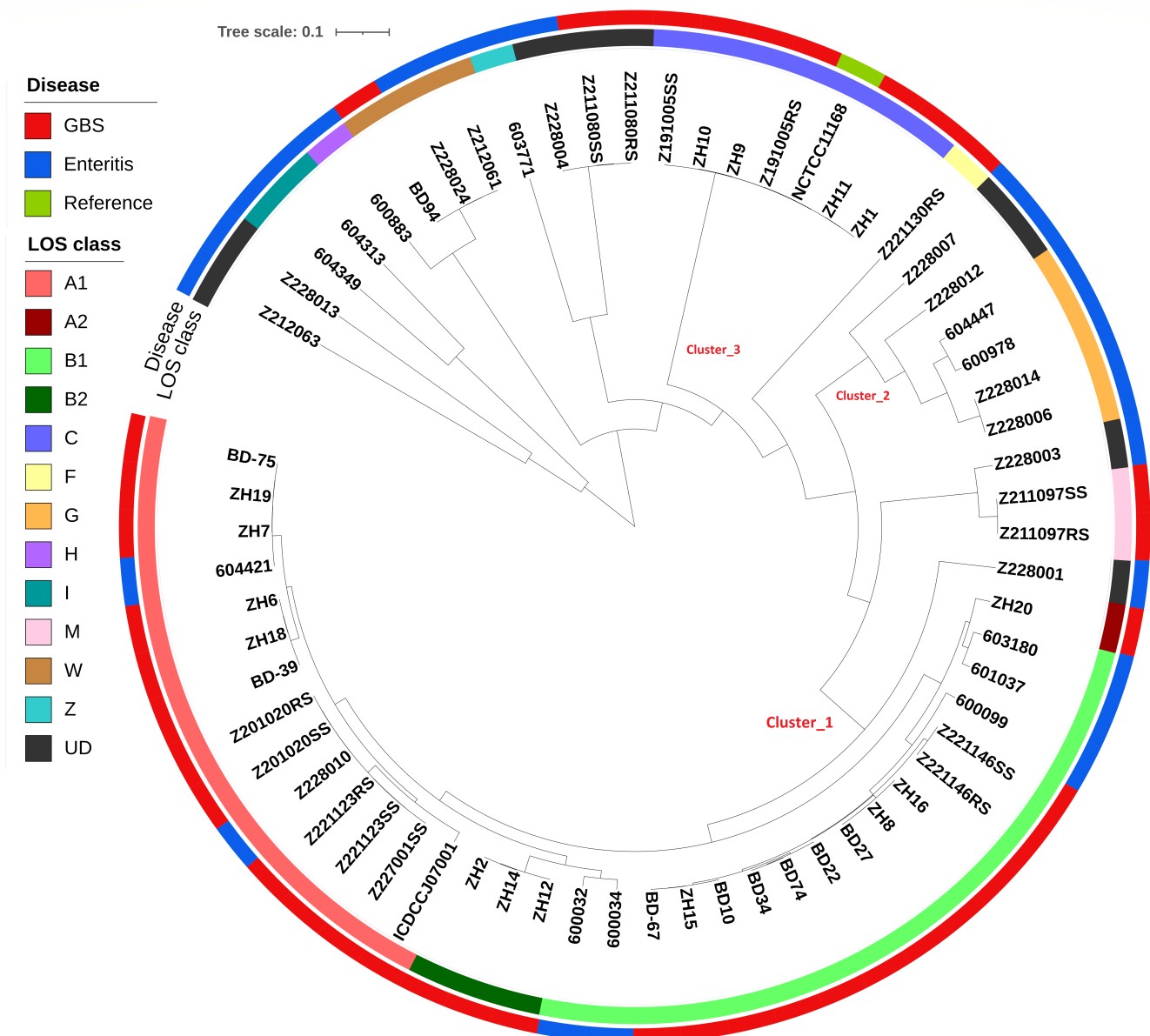

**FIG 2** Phylogenetic tree of the LOS region of *C. jejuni* strains constructed using the unweighted pair group method with arithmetic mean (UPGMA) methods among GBS-associated and enteritis-associated strains. The scale bar represents evolutionary distances in terms of nucleotide substitutions per site. LOS classes, along with the disease groups of the studied isolates, are visually presented in the tree using specific color codes. Clusters within the tree were defined arbitrarily based on genetic similarity and phylogenetic relatedness among the strains.

enteritis-associated strains. The LOS region of GBS-associated strains showed a similar pattern of gene alignment, with the majority of them (84%) belonging to LOS classes A, B, and C, whereas, the LOS regions of enteritis-associated strains exhibited a more varied gene pattern with higher presence of hypothetical proteins compared to GBS-associated strains, which belong to seven different LOS classes (Fig. 3A and B).

## Nucleotide variation in the LOS region

The overall distribution and frequency of nucleotide variations within the LOS region of *C. jejuni* were significantly higher ($P = 0.005$) among strains isolated from GBS patients compared to strains isolated from enteritis patients (Fig. 4A). Among nucleotide variations in the LOS region, single nucleotide polymorphism (SNP), multiple nucleotide

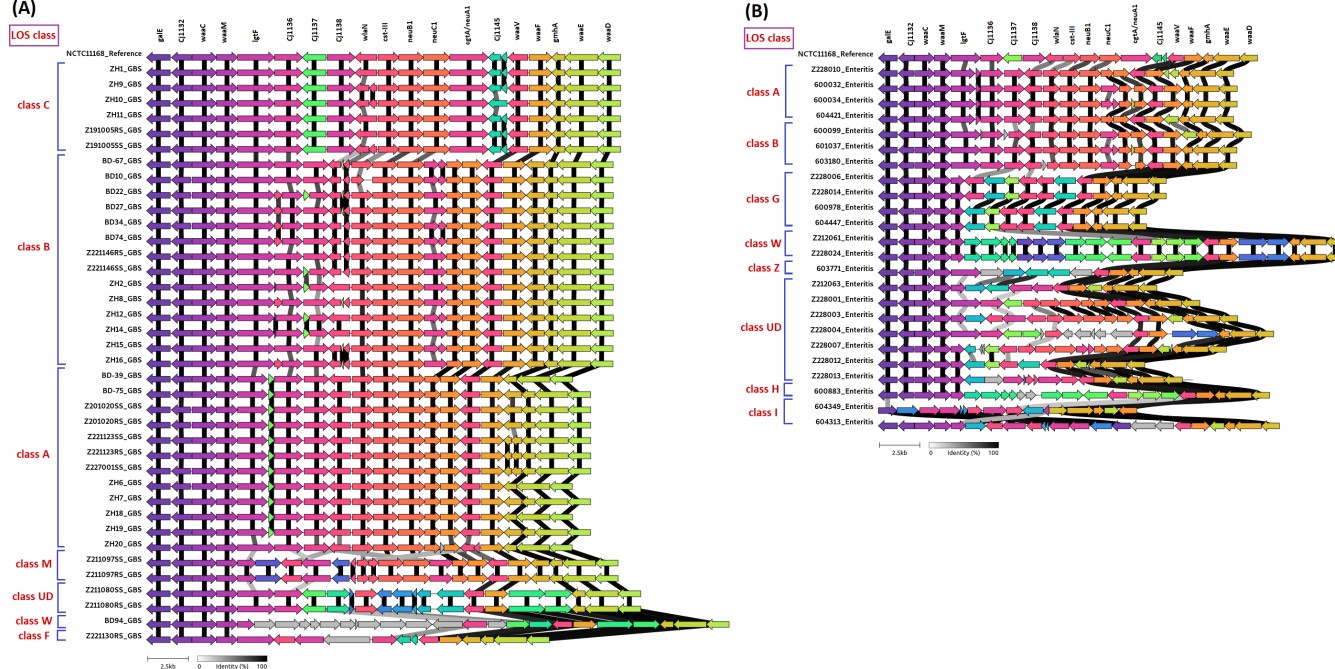

**FIG 3** Comparative genome alignment of LOS biosynthesis locus region between GBS-associated (A) and enteritis-associated *C. jejuni* strains (B). LOS region of *C. jejuni* NCTC11168 was used as the reference genome. Colored arrows indicate the coding sequences and various color-reflected homolog groups identified by the clinker. Gray/white arrows represent the hypothetical proteins. The black and gray bars represent the percentage of amino acid identity as indicated in the figure.

polymorphisms (MNP), and complex nucleotide variations (SNP/MNP) were found to be significantly higher in GBS-associated strains compared to enteritis-associated strains with *P*-values of 0.0006, 0.001, and 0.04, respectively (Fig. 4B through D).

## Variation in sialyltransferase gene

The sialyltransferase (*cst*) gene was detected in 34 out of 38 GBS-associated *C. jejuni* strains and nine out of 24 enteritis-associated strains. The *cst-II* gene (*n* = 28/38) was predominant in GBS-associated strains compared to the *cst-III* gene (*n* = 6/38). The *cst-II* with asparagine (Asn51) amino acid at the 51st position was frequently 68% (19/28) present in GBS-associated strains harboring the sialyltransferase gene. The *cst-II* gene with threonine at the 51st position (Thr51) was present in 32% (9/28) of GBS-associated strains (Fig. 1A). Among enteritis-associated strains, the *cst-II* gene was present in all sialyltransferase-carrying strains, while the *cst-III* gene was absent. Among the nine *cst-II*-carrying enteritis-associated strains, the asparagine (Asn51) amino acid at the 51st position was present in seven strains, while threonine (Thr51) was present in two strains.

## Phase variation within the LOS region

Thirteen phase-variable (PV) gene groups collectively harbored 108 phase-variable sites within the LOS region of *C. jejuni* strains. Among these, 107 were genic phase variations, while one was located within an intergenic repeat sequence (Table S3). Phase variation among major genes within the LOS region, which were grouped based on similar gene functions, is significantly higher (*P* = 0.004) in GBS-associated *C. jejuni* compared to enteritis-associated *C. jejuni* strains (Fig. 5). In the "On vs. length comparison for G/C tracts" analysis among 107 repeats, 95% (102/107) repeats were poly-G, 3% (3/107) were poly-TA, and 2% (2/107) were poly-T repeats. Within 103 poly-G repeats (genic, *n* = 102, and intergenic, *n* = 1); 8 bp, 9 bp, 10 bp, and 11 bp poly-G repeats were "ON" length for 10/103, 55/103, 33/103, and 5/103, respectively (Table S4). Poly-G tracts

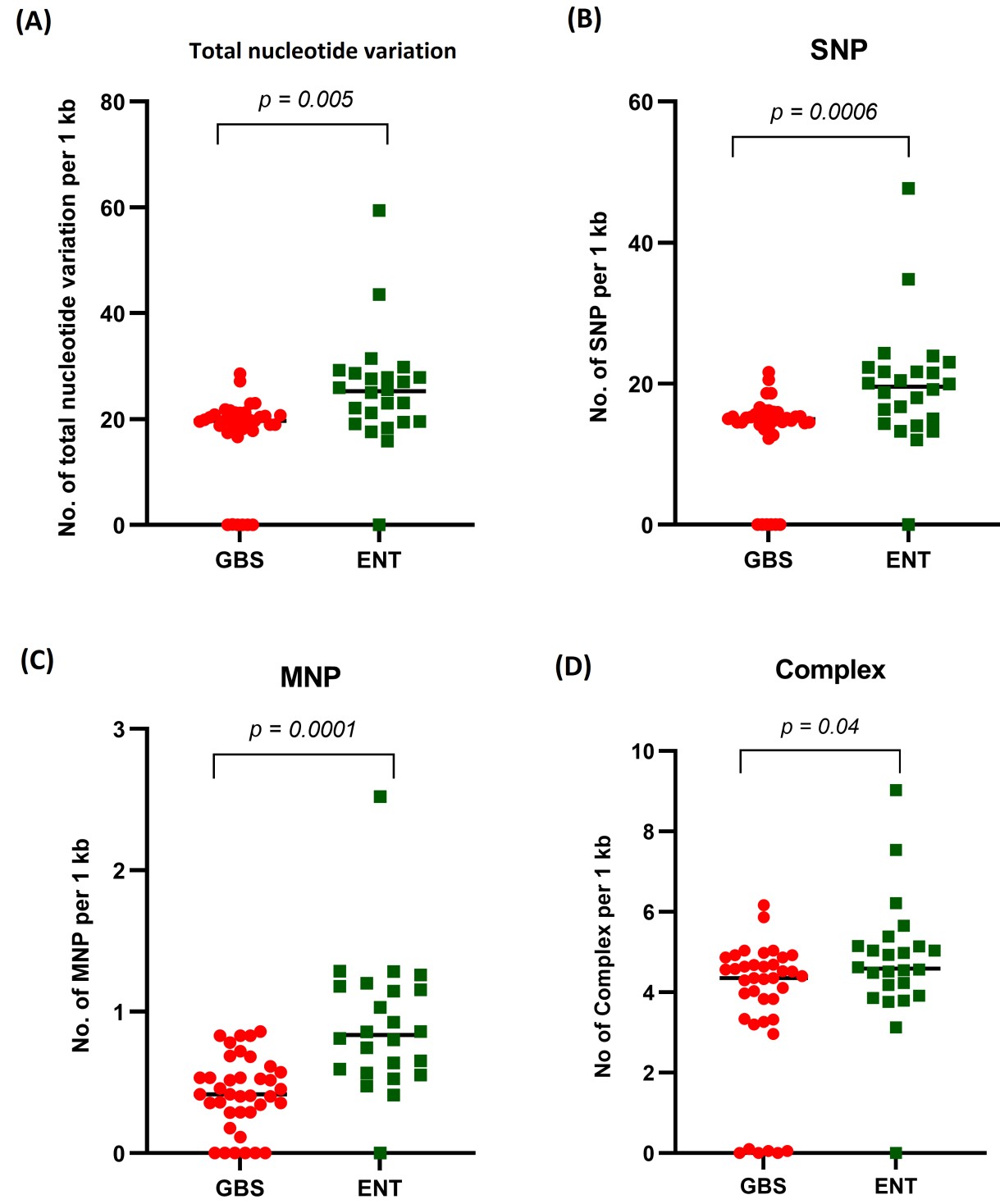

**FIG 4** Distribution of nucleotide variations within the LOS region between GBS-associated and enteritis-associated *C. jejuni*. (A) Distribution of total nucleotide variations, (B) distribution of SNPs, (C) distribution of MNPs, and (D) distribution of complex (combination of SNP/MNP).

were detected in the LOS region in 84% (32/38) of GBS-associated and 79% (19/24) of enteritis-associated *C. jejuni* strains. The frequency of poly-G was higher within the LOS region among GBS-associated strains compared to enteritis-associated *C. jejuni* strains (*P* = 0.004). Poly-TA tracts were detected in the LOS region of 3 GBS-associated *C. jejuni* strains (BD-75, ZH7, and ZH19), while poly-T tracts were found in the LOS region of two enteritis-associated *C. jejuni* strains (601037 and 603180) (Table S4).

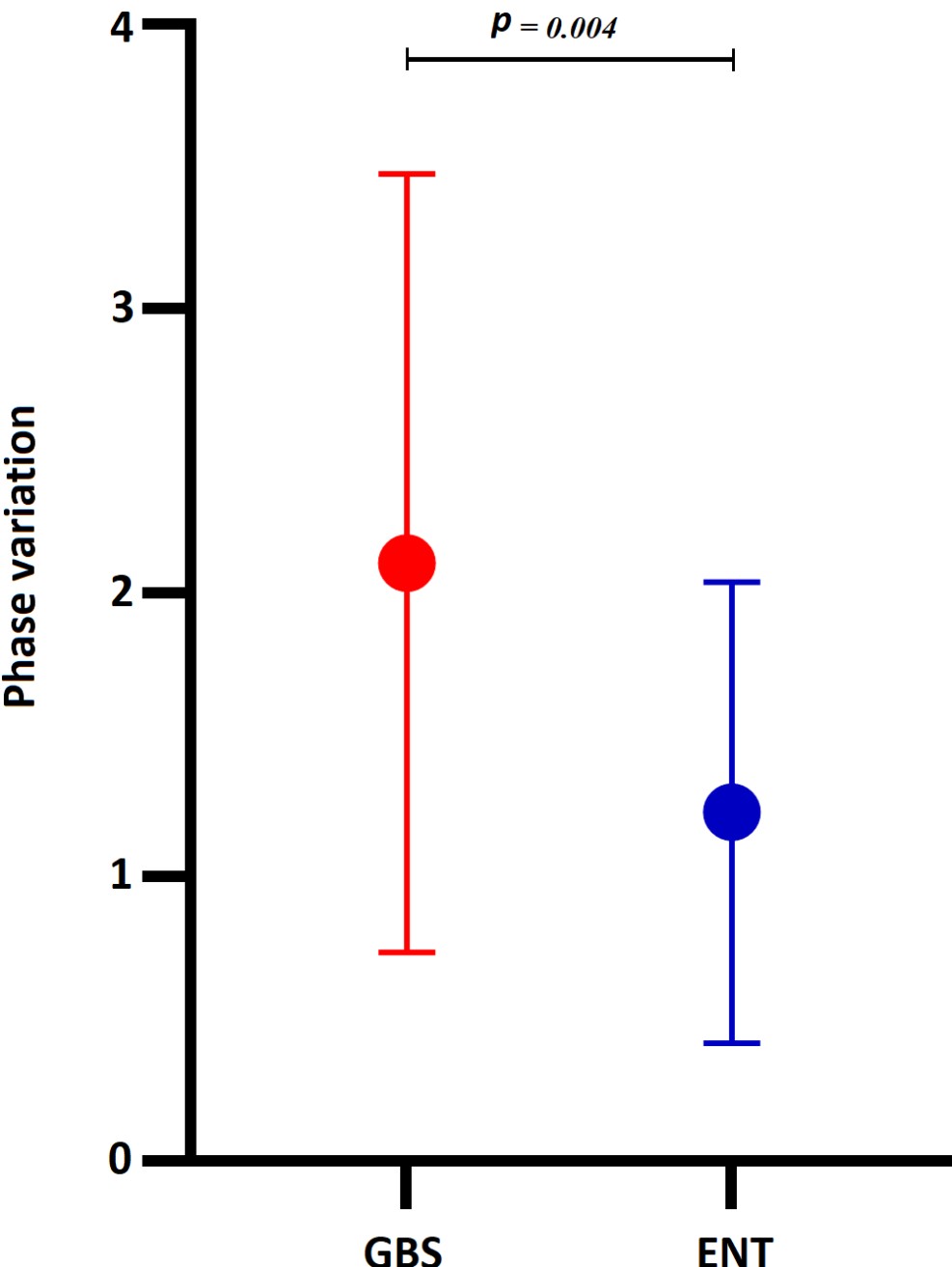

**FIG 5** The presence of phase variation within the LOS region of GBS-associated and enteritis-associated strains.

## DISCUSSION

The current comparative genomic study of *C. jejuni* investigated the in-depth strain level variation of LOS biosynthesis locus and sequence type among strains isolated from patients with GBS and enteritis in Bangladesh. This study revealed that GBS-associated *C. jejuni* strains frequently exhibit LOS classes A, B, and C, with class-specific nucleotide variation, elevated rates of phase variation, and the presence of the *cst-II* gene within the LOS locus, compared to enteritis-associated strains. Our analysis also revealed a greater prevalence of clonal complex ST-403 cc and ST-21 cc among GBS-associated strains. Furthermore, our research indicates that LOS class, sialyltransferase variation, and phase variation within the LOS region might be the significant contributing factors for inducing GBS.

*Campylobacter* is the leading bacterial cause of gastroenteritis worldwide, and its incidence is especially high in low- and middle-income countries (LMICs). Islam et al. reported that a prior infection of *C. jejuni* is linked to the high incidence of GBS in Bangladesh. Bacteria use several genetic mechanisms to achieve structural variability in cell surface oligosaccharides, including differences in gene content and sequence variation, resulting in changes in gene function and expression (15). In the present study, the comparative genomic variation analysis of the LOS region between GBS-associated and enteritis-associated *C. jejuni* strains did not reveal any unique gene that is specifically linked to GBS; however, the findings indicate the positioning of the genes in the LOS region might be the determining factor. The results are consistent with earlier findings from a comparative examination of *C. jejuni* using microarray technology, which showed that the LOS region had genetic diversity but no genes or regions that could distinguish between strains linked to enteritis and GBS (21, 22). However, the current study provides a clear overview of similar gene alignment patterns among GBS-associated strains, which are distinguishable from enteritis-associated strains.

This study identified different LOS classes among *C. jejuni* strains through comparative bioinformatic approaches. We found that the majority of GBS-associated *C. jejuni* strains possess LOS classes A, B, and C loci, while a smaller proportion of enteritis-associated *C. jejuni* strains carry the same LOS classes. Comparison of the LOS loci of various *C. jejuni* strains has demonstrated that only the classes A, B, and C LOS loci possess the genes strongly associated with ganglioside-mimicking structures, which were also previously identified in GBS-associated *C. jejuni* from other countries, including the Netherlands (6, 15, 23). The development of ganglioside-mimicking structures in *C. jejuni's* LOS regions is thought to be crucial for the production of autoantibodies that cause GBS (24). Some *C. jejuni* strains remain unclassified for LOS class due to the high variability and recombination within the LOS locus, leading to novel or mosaic gene arrangements. Additionally, gene loss or the presence of uncharacterized LOS gene combinations can result in undefined LOS types outside current classification schemes. Furthermore, phylogenetic analysis of LOS regions demonstrated that their genomic arrangement is predominantly influenced by structural locus classification rather than by the clinical association with GBS or enteritis.

Multi-locus sequence typing (MLST) and clonal complex were determined to examine the overall genomic diversity among GBS-associated and enteritis-associated *C. jejuni* strains. Specific sequence types (ST-43, ST-2042) and clonal complexes (ST-403 cc, ST-21 cc) were observed to have differing distributions between the two clinical groups, with certain clonal complexes (ST-403 cc) showing a stronger association with GBS-associated strains. Earlier research reported a consistent link between LOS class B and the ST-403 cc clonal complex among GBS-associated *C. jejuni* strains in Bangladesh (5). The current study reaffirms previous reports and strengthens the evidence for the link between LOS class B and clonal complex ST-403 cc. Additionally, we found that LOS class C is associated with clonal complex ST-42 cc among GBS-associated strains. However, it was previously reported that clonal complexes ST-21 cc and ST-22 cc were prevalent among GBS-associated *C. jejuni* strains isolated from the Netherlands, while ST-362 cc was predominant in GBS-associated strains from Peru (6, 23, 25).

The overall distribution and frequency of nucleotide variations in the *C. jejuni* LOS region substantially differ among GBS-associated strains compared to those isolated from enteritis patients. However, no specific nucleotide variation was found to be substantially linked with only GBS-associated strains. Heikema et al. described that LOS loci sequence alignment did not differentiate between GBS-associated and enteritis-associated strains (6). Several genes of the *C. jejuni* LOS biosynthesis locus are known to be significantly associated with ganglioside mimicry in neuropathic diseases such as GBS (7, 20). The sialyltransferase genes (*cst-II* and *cst-III*) of the *C. jejuni* LOS biosynthesis locus are present among most of the GBS-associated strains. Four GBS-associated strains lack the presence of any sialyltransferase genes. The previous study also reported that neuropathogenic strains with no sialyltransferase genes also exist (7).

Phase variation (PV) is a phenomenon common to a variety of bacterial species for niche adaptation and is also involved in the generation of the ganglioside GM1-like lipooligosaccharide of *C. jejuni* (17). This higher phase variability among GBS-associated strains may indicate the increased possibility of genetic changes and host ganglioside mimicry. As "ON" length poly-G containing PV genes may express one of the multiple antigenic forms of the protein, phase variation of gene off/on can cause structure variations of genes. The number of G residues in the LOS region's gene affected the length of the open reading frame, and these variations in the LOS region's gene correlated to a shift in the LOS structure from GM2 to GM3 ganglioside mimicry (26). Our analysis revealed that most of the phase variations in the lipooligosaccharide regions are poly-G repeats. Bacteria have been demonstrated to produce continual genetic diversity by a mechanism known as phase variation, in which gene on/off switching alters cell-surface LOS structures and functions (26).

In the current study, we conducted a comprehensive genomic analysis of bacterial genome sequences, utilizing various genome-based approaches, but did not assess the effect of these differences in lipooligosaccharide production, which is one of our limitations. Furthermore, phase variation can alter transcription or gene expression, which can result in the production of a truncated and non-functional protein. Additionally, host factors, including genetic predispositions and immune responses, critically influence individual susceptibility to GBS following infection, even when the infection is a known trigger (27). These factors play a central role in regulating inflammatory processes in GBS pathogenesis, with evidence highlighting the involvement of cytokines and Toll-like receptors in immune-mediated demyelination and axonal damage (28).

In *C. jejuni*, GBS-specific LOS classes and increased variability within the LOS region may be key factors contributing to the induction of GBS. Insights gained from this comparative genomic approach could lead to the development of more precise diagnostic tools for identifying specific *C. jejuni* strains linked to GBS and identifying specific genomic variations associated with GBS that can facilitate mitigation of the risk of developing GBS following *C. jejuni* infection. Overall, this study lays the groundwork for future research to elucidate the molecular mechanisms underlying the association between *C. jejuni* genomic variability and pathogenesis of GBS.

## MATERIALS AND METHODS

### Bacterial strains isolation and identification

In this study, we included 38 *Campylobacter jejuni* strains isolated from stool and rectal swab specimens of GBS patients in Bangladesh. Additionally, 12 *C. jejuni* strains isolated from stool specimens of enteritis patients in Bangladesh and 12 other enteritis-associated *C. jejuni* sequences retrieved from the National Center for Biotechnology Information (NCBI) that had been previously isolated from diarrheal patients in Bangladesh were included for comparison study. *C. jejuni* strains were isolated from stool and rectal specimens using standard microbiological procedures. The isolates were selected and enriched for 48 h at 42°C under microaerobic conditions in CHROMagar (*Campylobacter*) and blood agar with 5% sheep blood, respectively. Selected *C. jejuni* isolates were confirmed by species-specific PCR (29, 30). The identified *C. jejuni* was enriched in Brain-Heart Infusion Broth (BHIB) and stored using 30% glycerol at −80°C until sequencing.

### Extraction of genomic DNA and whole-genome sequencing

The genomic DNA of 50 *C. jejuni* strains (GBS strain, $n = 38$ and enteritis strains, $n = 12$) was extracted using the Wizard genomic DNA purification kit (Promega, USA), according to the manufacturer's instructions. The quality of DNA was assessed using a NanoDrop spectrophotometer (Thermo Scientific, USA) and quantified with a Qubit 2.0 fluorimeter (Life Technologies). Whole-genome sequencing was performed using the Illumina

NextSeq 500 system at the icddr,b Genome Centre (iGC). The raw sequence data were processed and assembled using a previously described approach (30). Sequence reads were aligned to the reference genome using the Burrows-Wheeler Aligner (BWA) v0.7.18, and genome coverage was calculated using SAMtools v1.20 (31, 32). The quality of the assembled genome was assessed using QUAST v5.2.0 (33). The assemblies were further filtered to remove short contigs < 200 bp in length. Kraken v1.1.1 was used to confirm the genus of sequenced bacterial isolates (34). The genome sequences are publicly available on the NCBI-SRA database under BioProjects PRJNA378546, PRJNA717137, and PRJNA607058, respectively (Table S1). Annotation was performed on the NCBI Prokaryotic Genome Annotation Pipeline (PGAP) v6.7 and rapid annotation using subsystem technology (RASTtk) (https://rast.nmpdr.org/rast.cgi) platform using default parameters (35, 36). The multi-locus sequence type (MLST) and clonal complex (CC) were determined through the PubMLST database, utilizing the sequences of seven housekeeping genes: *aspA*, *glnA*, *gltA*, *glyA*, *pgm*, *tkt*, and *uncA* (37).

## Identification of LOS biosynthesis gene cluster

The LOS region was extracted from each whole genome by aligning it with *C. jejuni* reference genome NCTC11168 using minimap2 (-c --cs -x asm20) (38). The LOS region in the *Campylobacter jejuni* reference genome (NCTC11168) is located between two specific genes: the *Cj1131c* (*gal*E) gene and the *Cj1151c* (*rfa*D) gene (14). The coordinate range (AL11168.1:1064895-1084736) was then lifted over using the minimap2 utility paftools.js (with "liftover"), and the specified genomic region was separated from each genome with SAMtools (39). Multiple sequence alignment of LOS loci was performed for all GBS, enteritis, and reference genomes NCTC11168 and ICDCCJ07001. The extracted LOS region sequences were annotated in the RAST annotation server (36). After annotation, each open reading frame (ORF) was manually verified for their homology to specific genes from the NCBI database using blastp (https://blast.ncbi.nlm.nih.gov/Blast.cgi).

## *In silico* LOS classification

The genome sequences of the LOS region from reference strains and subject strains (GBS and enteritis) were aligned using nucleotide BLAST (https://blast.ncbi.nlm.nih.gov). The LOS classification was performed following the methodology and references described by Hameed et al. (2020) (40). The LOS classes were confirmed by sequence alignment with ≥90% coverage and ≥90% nucleotide identity.

## Genome alignment and phylogenetic analysis of LOS biosynthesis locus

The annotated data of each LOS region was compared with blastn and visualized in clinker and clustermap.js using the GenBank (GBK) file of the specified LOS region of each *C. jejuni* genome (41). A phylogenetic tree was developed to understand the genomic variation of the LOS region in GBS-associated and enteritis-associated *C. jejuni* strains. The LOS regions of *C. jejuni* isolate were aligned on MAFFT online server version 7 using default parameters, and UPGMA clustering was performed on the sequences of the LOS locus (21). The phylogenetic tree was visualized using iTOL v6.8.1 (22).

## Variation of nucleotide in *Campylobacter jejuni* LOS biosynthesis locus

Nucleotide variations in the LOS biosynthesis locus region were detected through Snippy v4.6.0 (https://github.com/tseemann/snippy) using NCTC 11168 as the reference genome. Snippy was used to perform local alignment of sequencing reads using the BWA-MEM algorithm. Alignment files in SAM/BAM format were processed using SAMtools, and variant calling was carried out using FreeBayes, integrated within the Snippy pipeline (42). The number of nucleotide variations was normalized based on the length of the genome aligned to the reference genome. An unpaired *t*-test was performed using GraphPad Prism v9 to compare the number of unique single nucleotide polymorphisms (SNPs), multiple nucleotide polymorphisms (MNPs), complex

polymorphisms (combinations of SNPs/MNPs), and total nucleotide variations per 1 kb alignment within the LOS region between GBS-associated and enteritis-associated strains (Table S2).

## Identification of phase-variable genes within the LOS locus of *C. jejuni*

PhasomeIt tool was used for the identification of phase-variable (PV) genes within the LOS locus of the *C. jejuni* genome by analyzing associations with simple sequence repeats (43). To identify all putatively phase-variable loci in *C. jejuni* LOS regions, a cutoff value of "7 6 0 5 5" and filter cutoff of "W9" were used (43). We compared the number of "G/C" tracts containing PV genes in the "ON" state within LOS region of strains isolated from GBS and enteritis patients. Statistical significance was assessed based on the major gene groups (Table S3) using an unpaired *t*-test in GraphPad Prism v9.

## ACKNOWLEDGMENTS

This research activity was funded by the Fogarty International Center (FIC) and the National Institute of Neurological Disorders and Stroke (NINDS) of the National Institutes of Health (NIH), USA, under Award Number K43TW011447. icddr,b gratefully acknowledges the commitment of the Government of the People's Republic of Bangladesh and Global Affairs Canada for their unrestricted support. We acknowledge all study personnel who contributed to patient enrollment and specimen collection, data management, and laboratory support. We are indebted to the neurologists who referred their patients to this genomic study. We are thankful to the patients who participated in the study and provided their valuable data.

Z.I., M.A.J.N., and S.H. conceptualized and designed the study. The research methodology and execution plan were conceived by M.A.J.N., S.H., and A.A. M.A.J.N., A.A., and M.G.M. contributed to data acquisition and genomic analysis. ZI contributed to the acquisition of funding. M.A.J.N., S.N.F., R.B., M.A., and A.A. did material acquisition, laboratory experiments, and data acquisition. M.A.J.N. drafted the manuscript, which was critically reviewed by S.H., A.A., M.G.M., S.N.F., R.B., M.A., Y.K., I.J., and Z.I. for intellectual content. All authors read and approved the final manuscript before submission.

The correspondence and material requests would be addressed to Z.I.

## AUTHOR AFFILIATIONS

[1]Gut-Brain Axis Laboratory, Infectious Diseases Division (IDD), icddr,b, Dhaka, Bangladesh
[2]Department of Biochemistry and Molecular Biology, University of Dhaka, Dhaka, Bangladesh
[3]Department of Medical Microbiology and Infectious Diseases, Erasmus University Medical Center, Rotterdam, the Netherlands

## AUTHOR ORCIDs

Md. Abu Jaher Nayeem  http://orcid.org/0000-0002-2620-1762
Shoma Hayat  http://orcid.org/0000-0003-1064-9009
Asaduzzaman Asad  http://orcid.org/0000-0001-6894-0910
Md. Golam Mostafa  http://orcid.org/0009-0008-4587-2239
Shah Nayeem Faruque  http://orcid.org/0000-0001-6316-1459
Israt Jahan  http://orcid.org/0000-0001-9594-0705
Zhahirul Islam  http://orcid.org/0000-0003-0935-8079

## FUNDING

| Funder | Grant(s) | Author(s) |
| --- | --- | --- |
| Fogarty International Center | K43TW011447 | Zhahirul Islam |

## AUTHOR CONTRIBUTIONS

Md. Abu Jaher Nayeem, Conceptualization, Data curation, Formal analysis, Investigation, Methodology, Software, Validation, Visualization, Writing – original draft, Writing – review and editing | Shoma Hayat, Conceptualization, Data curation, Investigation, Project administration, Supervision, Validation, Writing – review and editing | Asaduzzaman Asad, Data curation, Formal analysis, Methodology, Software, Validation, Writing – review and editing | Md. Golam Mostafa, Formal analysis, Methodology, Software, Writing – review and editing | Shah Nayeem Faruque, Data curation, Formal analysis, Methodology, Writing – review and editing | Ruma Begum, Data curation, Methodology, Writing – review and editing | Mosabbir Ahmed, Data curation, Methodology, Writing – review and editing | Yearul Kabir, Investigation, Supervision, Writing – review and editing | Israt Jahan, Investigation, Project administration, Supervision, Writing – review and editing | Zhahirul Islam, Conceptualization, Funding acquisition, Investigation, Project administration, Resources, Supervision, Writing – review and editing

## DATA AVAILABILITY

The authors confirm that all supporting data, and protocols have been provided within the article or through supplementary data files. The raw sequencing data for this project and primary assemblies are made publicly available under projects PRJNA378546, PRJNA717137, and PRJNA607058. The genome accession numbers and other metadata are provided in the supplementary file.

## ETHICS APPROVAL

Each participant provided written informed consent, and all studies were carried out following relevant guidelines and regulations. The study protocol was reviewed and approved by the Institutional Review Board (IRB) and the Ethical Review Committee (ERC) of icddr,b, Dhaka, Bangladesh, under protocol number PR-19048.

## ADDITIONAL FILES

The following material is available online.

### Supplemental Material

**Supplemental tables (Spectrum00062-25-s0001.docx).** Tables S1 to S4.

### Open Peer Review

**PEER REVIEW HISTORY (review-history.pdf).** An accounting of the reviewer comments and feedback.

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
