## [Reviewer comments · Microbiology Spectrum]

Microbiology Spectrum

Genomic Variability of Lipooligosaccharide Biosynthesis Locus and Sequence Type among *Campylobacter jejuni* Isolated from Patients with Guillain-Barre´ Syndrome

Md. Abu Jaher Nayeem, Shoma Hayat, Asaduzzaman Asad, Md. Golam Mostafa, Shah Faruque, Ruma Begum, Mosabbir Ahmed, Yearul Kabir, Israt Jahan, and Zhahirul Islam

Corresponding Author(s): Zhahirul Islam, International Centre for Diarrhoeal Disease Research Bangladesh

Review Timeline:

Submission Date:	January 7, 2025
Editorial Decision:	March 31, 2025
Revision Received:	May 4, 2025
Editorial Decision:	May 15, 2025
Revision Received:	May 29, 2025
Accepted:	May 30, 2025

Editor: Sadjia Bekal

Reviewer(s): Disclosure of reviewer identity is with reference to reviewer comments included in decision letter(s). The following individuals involved in review of your submission have agreed to reveal their identity: Willi Quino (Reviewer #1); Isabelle Bernaquez (Reviewer #2)

Transaction Report:

DOI: <https://doi.org/10.1128/spectrum.00062-25>

Re: Spectrum00062-25 (Genomic Variability of Lipooligosaccharide Biosynthesis Locus and Sequence Type among *Campylobacter jejuni* Isolated from Patients with Guillain-Barre´ Syndrome)

Dear Dr. Zahirul Islam:

Thank you for the privilege of reviewing your work. Below you will find my comments, instructions from the Spectrum editorial office, and the reviewer comments.

Revision Guidelines

Sincerely,
Sadjia Bekal
Editor
Microbiology Spectrum

Reviewer #1 (Editor):

The study is relevant, however some gaps related to genome quality assessment are present and could affect analysis and conclusions

1. Authors should detail the quality of selected genomes (number of contigs, N50, length of genomes), that should reflect the robustness of analysis

2. Authors used genomes with low coverage (5X). This is very low, considering the importance of the study. I recommend to re-analyse genomes, using only those with higher coverage 10X and more then 15X and more, to provide a more robust analysis.

Reviewer #2 (Comments for the Author):

The study is relevant, however some gaps related to genome quality assessment are present and could affect analysis and conclusions

1. Authors should detail the quality of selected genomes (number of contigs, N50, length of genomes), that should reflect the robustness of analysis

2. Authors used genomes with low coverage (5X). This is very low, considering the importance of the study. I recommend to re-analyse genomes, using only those with higher coverage 10X and more then 15X and more, to provide a more robust analysis.

Introduction

1. The molecular-mimicry phenomenon that explains how certain *Campylobacter* can provoke a cross-reactive antibody response is repeated at many points in the Introduction. I would recommend reviewing this section to make it more concise.

Results

2. I find Figure 1a would be easier to analyze if the tree was displayed as a rectangle. I know that we usually see circularized trees in papers, but that is mostly due to the large number of sequences which would make it impossible to fit rectangularly in one page. But in your case, as there are only 63 sequences, I think it could fit well within figure dimensions and could provide extra space to include other metadata to the tree (ex. All information from Table 1).

3. Is the phylogenetic tree in Figure 1A based only on the 7 MLST loci, the cgMLST scheme or the wgMLST scheme in PubMLST? Many different schemas for *Campylobacter* exist and it was not clear in the Methods section which scheme was used, as it seems to be only roughly mentioned in Figure 1's description. No distance scale is provided either to interpret the branch lengths. Why not use Snippy on the whole genome instead?

4. I think Figure 1a and 1b could be better represented as a compiled bar graph with the STs and CCs included in the figure, which would make it easier to visualize the commonalities and differences between the two groups (GBS and enteritis).

5. Line 142: Why were the LOS classes not determined for some of the strains? Did they not correspond to the classes defined by Hameed et al. (2020)? An explanation here could be useful to better understand the results.

6. Line 148: Did you mean to say "undetermined" instead of "undermined" here?

7. Lines 162-168: It wasn't clear what reference genome was used with Snippy to identify the nucleotide variations within the LOS locus. What LOS class did the reference genome have? Could this have biased your results?

8. Lines 168-175: I feel this section describes Figure 2, the phylogeny of the LOS locus of your 63 strains, which should be mentioned earlier in the manuscript, when Figure 2 is first mentioned. I don't see why the phylogenetic analysis and the nucleotide variation is described together here. I would recommend adding it to "Identification of frequency of *C. jejuni* LOS locus classes" and to modify the title, or to give it its own subsection.

9. In Figure 2, do you know why LOS classes A and B cluster into different branches? In other words, classes A and B are not monophyletic, as is the case for the other LOS classes presented here. Is this expected? Do the LOS classes A and B have a lot of genetic similarity and overlap?

10. Were the clusters 1, 2 and 3 in Figure 2 defined statistically by Snippy, or were they arbitrarily defined by the authors?

11. Do the *cst* gene profiles (II vs III and Asn51 vs Thr51) correlate with the LOS classes or ST/CC? For example, are all *cst*-II (Asn51) within LOS class A? It could be useful to include the *cst* gene information as metadata in one of the phylogenetic trees (Figure 1 or 3) to improve visualization instead of only describing them in Table 1.

12. Line 188: There doesn't seem to be many genes inside the LOS locus, but you claim 108 phase variable genes were found. Perhaps you meant to say "sites" instead of "genes"? Later on, you mention 13 different groups of PV genes. Were these 108 sites found within 13 different genes inside the LOS locus? Providing the name of these genes is suggested. This paragraph should be reworked to improve comprehension.

13. Again, do the different polymeric tracts correlate with the different LOS classes and/or ST/CC observe between the GBS and

enteritis groups? Are they dependent on the other variables?

14. I'm not convinced that there is more nucleotide variation inside the LOS locus in GBS-associated strains compared to enteritis strains. Could you explain in more detail how the differences were counted? Does the reference genome used for Snippy include all possible genes that could be included in the LOS locus? Because since Snippy is a reference-based alignment tool, if an LOS locus gene is absent from the reference genome, no SNPs will be found for this gene. Is it possible that the LOS locus in the reference genome shared more resemblance to the primary LOS classes detected in GBS (class A and B), and that's why you observe more nucleotide variation? Also, some LOS classes illustrated in Figure 3b (ex. class G) seem to have less gene content than the other LOS classes and is more represented in enteritis than in GBS. With that being said, perhaps the frequency of nucleotide variation in the LOS locus should be normalized on the full length of the LOS locus. This might explain why more variation is observed in GBS-associated strains due to the longer LOS locus observed on average. Same goes for the frequency in phase variation in Figure 5.

15. I would recommend including the outputs of Snippy and Phasomelt or creating a summary table describing their results as supplementary tables.

Discussion

16. Line 208: Parentheses not needed.

17. Lines 210-212: "our research indicates that LOS class, sialyltransferase variation, and phase variation within the LOS region might be the significant contributing factors for inducing GBS" ... However, it was not demonstrated if these 3 factors are co-dependent. Do only certain LOS classes (ex. A and B) have more significant sialyltransferase and phase variation? How do these 3 factors differ from one another?

18. Lines 260-262: I would recommend placing this text in the Introduction instead.

19. I would suggest mentioning that other host-related factors can influence the development of GBS (ex. state of immune system, age, etc.). Based on previous research, genotypes alone do not seem to be able to predict with 100% accuracy the development of GBS.

Methods

20. Line 313: I don't understand what has been filtered from the assemblies here... Did contigs under 5x coverage stay within the assembly? 5x coverage is not very high to generate a good quality genome assembly. What were the range of coverages in your genomes? What amount of N bases did you have per genome on average? How many contigs were in your final assemblies? This information could be added to support the quality of the sequences generated.

21. Line 336: The Hameed reference is dated in 2020, not 2022.

22. Line 356: I would be careful with the use of the term "novel" here to describe Phasomelt, as the paper was published 6 years ago...

**Genomic Variability of Lipooligosaccharide Biosynthesis Locus and Sequence Type among**
***Campylobacter jejuni* Isolated from Patients with Guillain-Barre´ Syndrome**

Md. Abu Jaher Nayeem,^a Shoma Hayat,^a Asaduzzaman Asad,^a Md. Golam Mostafa,^a Shah
Nayeem Faruque,^a Ruma Begum,^a Mosabbir Ahmed,^a Yearul Kabir,^b Israt Jahan,^{a,c} Zhahirul
Islam,^{a*}

7 ^aGut-Brain Axis Laboratory, Infectious Diseases Division (IDD), icddr,b, Dhaka 1212,
Bangladesh.

9 ^bDepartment of Biochemistry and Molecular Biology, University of Dhaka, Dhaka 1000,
Bangladesh.

11 ^cDepartment of Medical Microbiology and Infectious Diseases, Erasmus University Medical
Center, Rotterdam, The Netherlands.

Running Title: Lipooligosaccharide and sequence types in *C. jejuni* associated with GBS

***Corresponding author**

Zhahirul Islam, Ph.D.

Gut-Brain Axis Laboratory

Infectious Diseases Division (IDD)

icddr,b, Dhaka, Bangladesh

68, Shaheed Tajuddin Ahmad Sarani, Mohakhali, Dhaka-1212, Bangladesh

Phone: +880 2 9886464, Fax: +880 2 8812529

E-mail: zislam@icddr.org

**ABSTRACT**

Lipooligosaccharide (LOS) is a crucial component of the *Campylobacter jejuni* (*C. jejuni*) cell
membrane that mimics human gangliosides and can induce autoimmune neuropathies like
Guillain-Barré syndrome (GBS). Approximately 1 in 1000 *C. jejuni* infections lead to GBS, with
LOS variations influencing immune response and disease induction. We aimed to investigate the
sequence type and genomic variation of the LOS region of *C. jejuni* strains isolated from patients
with GBS and enteritis. We isolated 38 GBS- and 24 enteritis-associated *C. jejuni* strains from
Bangladesh, sequenced them on the Illumina platform and compared their LOS regions using
various bioinformatics approaches. Comparative genomic analysis between GBS-associated and
enteritis-associated strains revealed that LOS class A, B and C were prevalent ($p = 0.0001$) in
GBS-associated *C. jejuni* strains. Sequence types ST-43 and ST-2042 and clonal complexes ST-
403 cc and ST-21 cc were overrepresented in GBS compared to enteritis-associated strains. All
GBS-associated strains carrying LOS class C belonged to clonal complex ST-21 cc. A
significantly higher ($p = 0.025$) number of nucleotide variations (NVs) were detected within the
LOS region of GBS-associated strains. In GBS strains, phase variation (PV) within the LOS
region was significantly higher ($p = 0.004$). In conclusion, the over-representation of GBS-
specific LOS classes with higher rates of nucleotide and phase variations might be the
determining factor for GBS. However, further research on integrating DNA epigenetic analysis
and metabolic profiling is needed to establish the *C. jejuni*-specific risk factors of GBS.

**IMPORTANCE**

The lipooligosaccharide (LOS) biosynthesis region is the most variable and virulent genomic
region of *Campylobacter jejuni*. This study addressed the critical gap in understanding the
genomic features contributing to GBS by analyzing the LOS region of *C. jejuni* strains from

GBS and enteritis patients. Our analysis of *C. jejuni* isolates from GBS and enteritis cases
revealed genomic variations in the sequence type and LOS region, including distinct LOS classes
(A, B, C), higher prevalence of the *cst-II* gene and significant presence of phase variability
among GBS-associated *C. jejuni* strains. The findings highlight the association of *C. jejuni* LOS-
region in the pathogenesis of GBS. In addition, the results of this study will facilitate future
comparisons by identifying genomic differences in lipooligosaccharide biosynthesis and
sequence types between *C. jejuni* strains linked to GBS. This will enhance our understanding of
the genomic variation of the LOS region and their role in GBS development.

**KEYWORDS**

Guillain-Barré syndrome, *Campylobacter jejuni*, lipooligosaccharide, comparative genomics,
phase variation

Introduction

*Campylobacter jejuni* (*C. jejuni*) is a leading cause of gastroenteritis and has been causatively
associated with the onset of peripheral autoimmune neuropathy such as Guillain-Barré syndrome
(GBS) (1). This association is linked to specific genetic and structural variations within the
bacterial strains, particularly within the lipooligosaccharide (LOS) locus. LOS is the most
important virulence factor of *C. jejuni*, structurally mimicking human gangliosides and thereby
triggering an autoimmune response that can lead to GBS (2). *C. jejuni* infection may develop
GBS in approximately 1 in 1000 enteritis (ENT) patients as a result of a cross-reactive immune
response that induces anti-ganglioside antibodies (3–5). In Bangladesh, *C. jejuni* is recognized as
the most prevalent pathogen associated with GBS (3). The onset of autoimmune neuropathies
following *C. jejuni* infection is primarily due to molecular mimicry between LOS on *C. jejuni*'s
cell surface and gangliosides on human peripheral nerves (6,7). Previous studies have reported
that *C. jejuni* LOS loci class A, B, and C, as well as the serotype *C. jejuni* HS:23 and clonal
complex ST-403 cc, were associated with a high frequency of *C. jejuni*-associated GBS (8–10).
Moreover, no studies have conducted a comparative sequence-based approach to examine
genomic variations within the LOS region of *C. jejuni*.

Previously, a comparative genotyping approach revealed remarkable diversity among the
sequence types (STs) and clonal complexes (CCs) of various *C. jejuni* strains isolated from
patients with enteritis and patients with GBS in Bangladesh (8). This diversity highlights the
genetic variability of *C. jejuni* strains and their different associations with gastrointestinal and
neurological diseases (11). Cohort-based studies on GBS-associated *C. jejuni* cases have
indicated that genomic variation in the LOS region of *C. jejuni* may be responsible for the
endemicity of GBS in Bangladesh (3,8). These studies suggest that specific genetic differences

within the LOS locus contribute to the high prevalence of GBS in this region, underscoring the
need for further research into these genomic variations. Biochemical and structural analyses of
the LOS outer core oligosaccharides of *C. jejuni* have revealed sialylated moieties structurally
similar to various gangliosides (2,12). This structural resemblance, known as molecular mimicry,
facilitates the production of anti-ganglioside antibodies during infection and contributes to the
onset of GBS (2,13,14). The LOS region of *C. jejuni* isolated from GBS patients has been shown
to mimic the peripheral nerve gangliosides, mainly GM1, GD1a, and GQ1b (15). The
sialyltransferase (*cst-II*) gene is a key in adding neuraminic acid to LOS which is linked to GBS.
Strains with *cst-II* (Thr51) express GM1- and GD1a-like LOS, while *cst-II* (Asn51) variants
produce GT1a- and GD1c-like LOS, influencing immune responses and GBS risk (16). During
infection, the induction of these LOS may prompt antibodies to bind to similar ganglioside
structures on peripheral nerves. This cross-reactivity can trigger complement-mediated immune
activation, resulting in nerve damage and the onset of GBS (17).

*C. jejuni* LOS biosynthesis region displays considerable genetic variation among strains,
featuring notable differences in the arrangement of genes which encode LOS carbohydrate
moieties and their linkages. Furthermore, there is significant structural variety between strains
due to the susceptibility of the LOS gene locus to periodic phase fluctuations (18,19). There are
notable differences, especially in the outer section of the oligosaccharide moiety, with variations
in lipid A phosphorylation and the number of amide linkages (20). The complete genome
sequence analysis of *C. jejuni* NCTC 11168 revealed the involvement of a cluster of genes that
extends from Cj1131c (*galE*) to Cj1151c (*rfaD*) with lower GC content (18,21,22). The LOS
biosynthesis locus, a gene cluster that is genetically quite varied and interchangeable between
strains, harbors the *C. jejuni* genes involved in ganglioside mimicry (23). A cross-reactive

antibody response is induced in nerve tissue due to the molecular similarity between
*Campylobacter* LOS and nerve gangliosides; although, the precise pathophysiology of post-
*Campylobacter* neuropathy, like GBS, is unclear (13). The variations in LOS class, gene
alterations, mutations, and mechanisms such as phase variation (PV) within the LOS locus
contribute to structural variations in the ganglioside mimics (18,19,23).

[revised manuscript text omitted]

**Extraction of genomic DNA and whole-genome sequencing**

The genomic DNA of 50 *C. jejuni* strains (GBS strain, $n = 38$ and enteritis strains, $n = 12$) were
extracted using the Wizard® genomic DNA purification kit (Promega, USA) according to the
manufacturer's instructions. The quality of DNA was assessed using NanoDrop
spectrophotometer (Thermo Scientific, USA) and quantified with Qubit 2.0 fluorimeter (Life
Technologies). Whole-genome sequencing was performed using Illumina technology at the
icddr,b Genome Centre (iGC), and the sequences were processed and assembled using a
previously described approach (33). Sequence coverage was checked by mapping the reads to
reference sequence with Burrows-Wheeler Aligner (BWA) v0.7.18 and coverage was calculated

using SAMtools v1.20 (34,35). The quality of the assembled genome was assessed using
QUAST v5.2.0 (36). Confirmation of genus and detection of possible contaminated contigs was
carried out on Kraken v1.1.1 (37). The assemblies were further filtered to remove contigs <
200bp with a k-mer coverage of $\geq 5X$. The genome sequences are publicly available on the
NCBI-SRA database and are associated with the BioProjects PRJNA378546, PRJNA717137,
and PRJNA607058 respectively (Table S1). Annotation was performed on the NCBI Prokaryotic
Genome Annotation Pipeline (PGAP) v6.7 and rapid annotation using subsystem technology
(RASTtk) (<https://rast.nmpdr.org/rast.cgi>) platform using default parameters (38,39). Sequence
type and clonal complex were determined using the PubMLST database (40).

**Identification of LOS biosynthesis gene cluster**

The LOS region was extracted from each whole genome through alignment with *C. jejuni*
reference genome NCTC11168 using minimap2 (-c --cs -x asm20) (41). The LOS region in the
*Campylobacter jejuni* reference genome (NCTC11168) is located between two specific genes:
the *Cj1131c* (*galE*) gene and the *Cj1151c* (*rfaD*) gene (18). The coordinate range
(AL11168.1:1064895-1084736) was then lifted over using minimap2 utility paftools.js (with
“liftover”) and the specified genomic region was separated from each genome with SAMtools
(42). Multiple sequence alignment of LOS loci was performed for all GBS, enteritis, and
reference genomes NCTC11168 and ICDCJ07001. The extracted LOS region sequences were
annotated in the RAST annotation server (39). After annotation, each open reading frame (ORF)
was manually verified for their homology to specific genes from the NCBI database using blastp
(<https://blast.ncbi.nlm.nih.gov/Blast.cgi>).

***In silico* LOS classification**

The genome sequences of the LOS region from reference strains and subject strains (GBS and
enteritis) were aligned using Nucleotide BLAST (<https://blast.ncbi.nlm.nih.gov>). The LOS
classification was performed following the methodology and references described by *Hameed et*
*al.* 2022 (43). The LOS classes were confirmed by sequence alignment with $\geq 90\%$ coverage and
$\geq 90\%$ nucleotide identity.

**Genome alignment and phylogenetic analysis of LOS biosynthesis locus**

The annotated data of each LOS region was compared with blastn and visualized in clinker and
clustermap.js using the GenBank (GBK) file of the specified LOS region of each *C. jejuni*
genome (44). A phylogenetic tree was developed to understand the genomic variation of the LOS
region in GBS-associated and enteritis-associated *C. jejuni* strains. The LOS regions of *C. jejuni*
isolate were aligned on MAFFT online server version 7 using default parameters, and UPGMA
clustering was performed on the sequences of the LOS locus (45). The phylogenetic tree was
visualized using iTOL v6.8.1 (46).

**Variation of nucleotide in *Campylobacter jejuni* LOS biosynthesis locus**

Nucleotide variations in the LOS biosynthesis locus region were detected through SNIPPY
v4.6.0 (<https://github.com/tseemann/snippy>). Snippy uses the BWA-MEM algorithm to perform
local alignment, and SAMtool for parsing and manipulating alignment in SAM/BAM format.
Freebayes was used for variant calling (47). An unpaired t-test was performed using GraphPad
Prism v9 to compare the number of unique single nucleotide polymorphisms (SNPs), multiple
nucleotide polymorphisms (MNPs), complex polymorphisms (combinations of SNPs/MNPs),
and total nucleotide variations within the LOS region between GBS-associated and enteritis-
associated strains.

**Identification of phase variable genes within the LOS locus of *C. jejuni***

PhasomeIt, a novel program which identifies phase variable genes in *C. jejuni* genome through
association with simple sequence repeats was used to detect Phase Variable (PV) genes in the
LOS locus. To identify all putatively phase-variable loci in *C. jejuni* LOS regions, a cut-off value
of '7 6 0 5 5' and filter cutoff of 'W9' was used (48). Fisher's exact test was performed for 'G/C'
tracts containing PV genes of strains isolated from GBS and enteritis patients, depending on
whether their PV genes are ON or OFF.

***Acknowledgements***

This research activity was funded by the Fogarty International Center (FIC), National Institute of
Neurological Disorders and Stroke (NINDS) of the National Institutes of Health (NIH), USA
under Award Number K43TW011447. icddr,b gratefully acknowledges the commitment of the
Government of The People's Republic of Bangladesh and Global Affairs Canada for their
unrestricted support. We acknowledge all study personnel who contributed to patient enrollment
and specimen collection; data management and laboratory support. We are indebted to the
neurologists who referred their patients to this genomic study. We are thankful to the patients
who participated in the study and provided their valuable data.

***Author Contributions***

ZI, MAJN and SH conceptualized and designed the study. The research methodology and
execution plan were conceived by MAJN, SH and AA. MAJN, AA and MGM contributed to
data acquisition and genomic analysis. ZI contributed to the acquisition of funding. MAJN, SNF,
RB, MA, and AA did materials acquisition, laboratory experiments and data acquisition. MAJN
drafted the manuscript, which was critically reviewed by SH, AA, MGM, SNF, RB, MA, YK, IJ,
and ZI for intellectual content. All authors read and approved the final manuscript before
submission.

***Materials and Correspondence***

The correspondence and material requests would be addressed to ZI.

***Conflicts of interest***

The authors declare that there are no conflicts of interest.

***Ethical statement***

Each participant provided written informed consent, and all studies were carried out following
relevant guidelines and regulations. The study protocol was reviewed and approved by the
Institutional Review Board (IRB) and the Ethical Review Committee (ERC) of icddr,b, Dhaka,
Bangladesh under protocol number PR-19048.

**Data availability**

The authors confirm that all supporting data, and protocols have been provided within the article
or through supplementary data files. The raw sequencing data for this project and primary
assemblies are made publicly available under projects PRJNA378546, PRJNA717137, and
PRJNA607058. The accession numbers are provided in supplementary file **Table S1** which is
available in the online version of this article. Other metadata about study subjects and genome
sequences are provided in **Table 1**.

References

- 1. Malik A, Brudvig JM, Gadsden BJ, Ethridge AD, Mansfield LS. *Campylobacter jejuni*
induces autoimmune peripheral neuropathy via Sialoadhesin and Interleukin-4 axes. *Gut*
*Microbes*. 2022;14(1).
- 2. Islam Z, Gilbert M, Mohammad QD, Klaij K, Li J, van Rijs W, et al. Guillain-Barré
Syndrome-Related *Campylobacter jejuni* in Bangladesh: Ganglioside Mimicry and Cross-
Reactive Antibodies. Bereswill S, editor. *PLoS One* [Internet]. 2012 Aug 27;7(8):e43976.
Available from: <https://dx.plos.org/10.1371/journal.pone.0043976>
- 3. Islam Z, Jacobs BC, Van Belkum A, Mohammad QD, Islam MB, Herbrink P, et al.
Axonal variant of Guillain-Barré syndrome associated with *Campylobacter* infection in
Bangladesh. *Neurology*. 2010;74(7):581–7.
- 4. Allos BM. Association between *Campylobacter* infection and Guillain-Barre syndrome. *J*
*Infect Dis*. 1997;176(6 SUPPL.).
- 5. Mccarthy N, Andersson Y, Jormanainen V, Gustavsson O, Giesecke J. The risk of
Guillain-Barre syndrome following infection with *Campylobacter jejuni*. *Epidemiol*
*Infect*. 1999;122(1):15–7.
- 6. Godschalk PCR, Gilbert M, Jacobs BC, Kramers T, Tio-Gillen AP, Ang CW, et al. Co-
infection with two different *Campylobacter jejuni* strains in a patient with the Guillain-
Barré syndrome. *Microbes Infect* [Internet]. 2006 Jan;8(1):248–53. Available from:
<https://linkinghub.elsevier.com/retrieve/pii/S1286457905002601>
- 7. Godschalk PCR, Kuijff ML, Li J, St. Michael F, Ang CW, Jacobs BC, et al. Structural
characterization of *Campylobacter jejuni* lipooligosaccharide outer cores associated with
Guillain-Barré and Miller Fisher syndromes. *Infect Immun*. 2007;75(3):1245–54.
- 8. Islam Z, van Belkum A, Wagenaar JA, Cody AJ, de Boer AG, Tabor H, et al.
Comparative genotyping of *Campylobacter jejuni* strains from patients with Guillain-
Barré syndrome. *PLoS One*. 2009;4(9).
- 9. Heikema AP, Strepis N, Horst-Kreft D, Huynh S, Zomer A, Kelly DJ, et al. Biomolecule
sulphation and novel methylations related to Guillain-Barré syndrome-associated
*Campylobacter jejuni* serotype HS: 19. *Microb Genomics*. 2021;7(11):660.
- 10. Godschalk PCR, Heikema AP, Gilbert M, Komagamine T, Ang CW, Glerum J, et al. The
crucial role of *Campylobacter jejuni* genes in anti-ganglioside antibody induction in
Guillain-Barre syndrome. *J Clin Invest*. 2004;114(11):1659–65.
- 11. Djeghout B, Bloomfield SJ, Rudder S, Elumogo N, Mather AE, Wain J, et al.
Comparative genomics of *Campylobacter jejuni* from clinical campylobacteriosis stool
specimens. *Gut Pathog* [Internet]. 2022 Dec 7;14(1):45. Available from:
<https://gutpathogens.biomedcentral.com/articles/10.1186/s13099-022-00520-1>
- 12. Bowes T, Wagner ER, Boffey J, Nicholl D, Cochrane L, Benboubetra M, et al. Tolerance
to Self Gangliosides Is the Major Factor Restricting the Antibody Response to
Lipopolysaccharide Core Oligosaccharides in *Campylobacter jejuni* Strains Associated

- with Guillain-Barré Syndrome. *Infect Immun* [Internet]. 2002 Sep;70(9):5008–18.
Available from: <https://journals.asm.org/doi/10.1128/IAI.70.9.5008-5018.2002>
- 13. Yuki N. Molecular mimicry between gangliosides and lipopolysaccharides of
*Campylobacter jejuni* isolated from patients with Guillain-Barre syndrome and Miller
Fisher syndrome. *J Infect Dis*. 1997;176(6 SUPPL.):S150–3.
- 14. Yu RK, Usuki S, Ariga T. Ganglioside Molecular Mimicry and Its Pathological Roles in
Guillain-Barré Syndrome and Related Diseases. *Infect Immun* [Internet]. 2006
Dec;74(12):6517–27. Available from: <https://journals.asm.org/doi/10.1128/IAI.00967-06>
- 15. Hughes RAC, Cornblath DR. Guillain-barre syndrome. *Lancet*. 2005;366(9497):1653–66.
- 16. Yuki N. *Campylobacter* sialyltransferase gene polymorphism directs clinical features of
Guillain–Barré syndrome. *J Neurochem* [Internet]. 2007 Nov 6;103(s1):150–8. Available
from: <https://onlinelibrary.wiley.com/doi/10.1111/j.1471-4159.2007.04707.x>
- 17. Laman JD, Huizinga R, Boons G-J, Jacobs BC. Guillain-Barré syndrome: expanding the
concept of molecular mimicry. *Trends Immunol* [Internet]. 2022 Apr;43(4):296–308.
Available from: <https://linkinghub.elsevier.com/retrieve/pii/S147149062200028X>
- 18. Parker CT, Horn ST, Gilbert M, Miller WG, Woodward DL, Mandrell RE. Comparison of
*Campylobacter jejuni* Lipooligosaccharide Biosynthesis Loci from a Variety of Sources. *J*
*Clin Microbiol* [Internet]. 2005 Jun;43(6):2771–81. Available from:
<https://journals.asm.org/doi/10.1128/JCM.43.6.2771-2781.2005>
- 19. Parker CT, Gilbert M, Yuki N, Endtz HP, Mandrell RE. Characterization of
lipooligosaccharide-biosynthetic loci of *Campylobacter jejuni* reveals new
lipooligosaccharide classes: Evidence of mosaic organizations. *J Bacteriol*.
2008;190(16):5681–9.
- 20. Stephenson HN, John CM, Naz N, Gundogdu O, Dorrell N, Wren BW, et al.
*Campylobacter jejuni* Lipooligosaccharide Sialylation, Phosphorylation, and Amide/Ester
Linkage Modifications Fine-tune Human Toll-like Receptor 4 Activation. *J Biol Chem*
[Internet]. 2013 Jul;288(27):19661–72. Available from:
<https://linkinghub.elsevier.com/retrieve/pii/S0021925820456832>
- 21. Linton D, Karlyshev A V., Hitchen PG, Morris HR, Dell A, Gregson NA, et al. Multiple
N-acetyl neuraminic acid synthetase (*neuB*) genes in *Campylobacter jejuni*: Identification
and characterization of the gene involved in sialylation of lipo-oligosaccharide. *Mol*
*Microbiol*. 2000;35(5):1120–34.
- 22. Parkhill J, Wren BW, Mungall K, Ketley JM, Churcher C, Basham D, et al. The genome
sequence of the food-borne pathogen *Campylobacter jejuni* reveals hypervariable
sequences. *Nature* [Internet]. 2000;403(6770):665–8. Available from:
[file://localhost/Users/cliftonfranklund/Documents/Journal Articles/Parkhill\(2000\).pdf](file://localhost/Users/cliftonfranklund/Documents/Journal%20Articles/Parkhill(2000).pdf)
- 23. Gilbert M, Karwaski MF, Bernatchez S, Young NM, Taboada E, Michniewicz J, et al. The
genetic bases for the variation in the lipo-oligosaccharide of the mucosal pathogen,
*Campylobacter jejuni*. Biosynthesis of sialylated ganglioside mimics in the core
oligosaccharide. *J Biol Chem*. 2002;277(1):327–37.

- 24. Semchenko EA, Day CJ, Moutin M, Wilson JC, Tiralongo J, Korolik V. Structural
heterogeneity of terminal glycans in *Campylobacter jejuni* Lipooligosaccharides. *PLoS*
*One*. 2012;7(7).
- 25. Parker CT, Gilbert M, Yuki N, Endtz HP, Mandrell RE. Characterization of
lipooligosaccharide-biosynthetic loci of *Campylobacter jejuni* reveals new
lipooligosaccharide classes: Evidence of mosaic organizations. *J Bacteriol*.
2008;190(16):5681–9.
- 26. Heikema AP, Islam Z, Horst-Kreft D, Huizinga R, Jacobs BC, Wagenaar JA, et al.
*Campylobacter jejuni* capsular genotypes are related to Guillain-Barré syndrome. *Clin*
*Microbiol Infect*. 2015;21(9):852.e1-852.e9.
- 27. Koga M, Gilbert M, Takahashi M, Li J, Koike S, Hirata K, et al. Comprehensive analysis
of bacterial risk factors for the development of Guillain-Barré syndrome after
*Campylobacter jejuni* enteritis. *J Infect Dis*. 2006;193(4):547–55.
- 28. Heikema AP, Strepis N, Horst-Kreft D, Huynh S, Zomer A, Kelly DJ, et al. Biomolecule
sulphation and novel methylations related to guillain-barré syndrome-associated
*campylobacter jejuni* serotype hs:19. *Microb Genomics*. 2021;7(11).
- 29. Quino W, Caro-Castro J, Mestanza O, Hurtado V, Zamudio ML, Cruz-Gonzales G, et al.
Emergence and Molecular Epidemiology of *Campylobacter jejuni* ST-2993 Associated
with a Large Outbreak of Guillain-Barré Syndrome in Peru. Denes TG, editor. *Microbiol*
*Spectr* [Internet]. 2022 Oct 26;10(5). Available from:
<https://journals.asm.org/doi/10.1128/spectrum.01187-22>
- 30. Gilbert M, Karwaski MF, Bernatchez S, Young NM, Taboada E, Michniewicz J, et al. The
genetic bases for the variation in the lipo-oligosaccharide of the mucosal pathogen,
*Campylobacter jejuni*. Biosynthesis of sialylated ganglioside mimics in the core
oligosaccharide. *J Biol Chem*. 2002;277(1):327–37.
- 31. Guerry P, Szymanski CM, Prendergast MM, Hickey TE, Ewing CP, Pattarini DL, et al.
Phase variation of *Campylobacter jejuni* 81-176 lipooligosaccharide affects ganglioside
mimicry and invasiveness in vitro. *Infect Immun*. 2002;70(2):787–93.
- 32. Islam Z, Nabila FH, Asad A, Begum R, Jahan I, Hayat S, et al. Draft Genome Sequences
of Three Strains of *Campylobacter jejuni* Isolated from Patients with Guillain-Barré
Syndrome in Bangladesh. Rasko D, editor. *Microbiol Resour Announc* [Internet]. 2021
Apr 29;10(17). Available from: <https://journals.asm.org/doi/10.1128/MRA.00005-21>
- 33. Hayat S, Nabila FH, Asad A, Begum R, Jahan I, Endtz HP, et al. Draft Genome
Sequences of Four Strains of *Campylobacter jejuni* Isolated from Patients with Axonal
Variant of Guillain-Barré Syndrome in Bangladesh. Putonti C, editor. *Microbiol Resour*
*Announc* [Internet]. 2022 Feb 17;11(2). Available from:
<https://journals.asm.org/doi/10.1128/mra.01146-21>
- 34. Heng L, Richard D. Fast and accurate short read alignment with Burrows-Wheeler
Transform. *Bioinformatics*. 2009;25(14):1754–60.
- 35. Li H, Handsaker B, Wysoker A, Fennell T, Ruan J, Homer N, et al. The Sequence

- Alignment/Map format and SAMtools. *Bioinformatics* [Internet]. 2009 Aug
15;25(16):2078–9. Available from:
<https://academic.oup.com/bioinformatics/article/25/16/2078/204688>
- 36. Gurevich A, Saveliev V, Vyahhi N, Tesler G. QUASt: Quality assessment tool for
genome assemblies. *Bioinformatics*. 2013;29(8):1072–5.
- 37. Wood DE, Salzberg SL. Kraken: ultrafast metagenomic sequence classification using
exact alignments. *Genome Biol* [Internet]. 2014 Mar 3;15(3):R46. Available from:
<https://genomebiology.biomedcentral.com/articles/10.1186/gb-2014-15-3-r46>
- 38. Tatusova T, DiCuccio M, Badretdin A, Chetvernin V, Nawrocki EP, Zaslavsky L, et al.
NCBI prokaryotic genome annotation pipeline. *Nucleic Acids Res* [Internet]. 2016 Aug
19;44(14):6614–24. Available from: [https://academic.oup.com/nar/article-](https://academic.oup.com/nar/article-lookup/doi/10.1093/nar/gkw569)
[lookup/doi/10.1093/nar/gkw569](https://academic.oup.com/nar/article-lookup/doi/10.1093/nar/gkw569)
- 39. Aziz RK, Bartels D, Best A, DeJongh M, Disz T, Edwards RA, et al. The RAST Server:
Rapid annotations using subsystems technology. *BMC Genomics*. 2008;9.
- 40. Jolley KA, Bray JE, Maiden MCJ. Open-access bacterial population genomics: BIGSdb
software, the PubMLST.org website and their applications. *Wellcome Open Res*
[Internet]. 2018 Sep 24;3:124. Available from:
<https://wellcomeopenresearch.org/articles/3-124/v1>
- 41. Li H. New strategies to improve minimap2 alignment accuracy. *Bioinformatics*.
2021;37(23):4572–4.
- 42. Danecek P, JK B, J L, J M, V O, MO P, et al. Twelve years of SAMtools and BCFtools.
*Gigascience*. 2021;10(2).
- 43. Hameed A, Woodacre A, Machado LR, Marsden GL. An Updated Classification System
and Review of the Lipooligosaccharide Biosynthesis Gene Locus in *Campylobacter jejuni*
[Internet]. Vol. 11, *Frontiers in Microbiology*. 2020. p. 677. Available from:
<https://www.frontiersin.org/article/10.3389/fmicb.2020.00677>
- 44. Gilchrist CLM, Chooi YH. Clinker & clustermap.js: Automatic generation of gene cluster
comparison figures. *Bioinformatics*. 2021;37(16):2473–5.
- 45. Katoh K, Standley DM. MAFFT Multiple Sequence Alignment Software Version 7:
Improvements in Performance and Usability. *Mol Biol Evol* [Internet]. 2013 Apr
1;30(4):772–80. Available from: [https://academic.oup.com/mbe/article-](https://academic.oup.com/mbe/article-lookup/doi/10.1093/molbev/mst010)
[lookup/doi/10.1093/molbev/mst010](https://academic.oup.com/mbe/article-lookup/doi/10.1093/molbev/mst010)
- 46. Letunic I, Bork P. Interactive Tree Of Life (iTOL) v5: an online tool for phylogenetic tree
display and annotation. *Nucleic Acids Res* [Internet]. 2021 Jul 2;49(W1):W293–6.
Available from: <https://academic.oup.com/nar/article/49/W1/W293/6246398>
- 47. Garrison E, Marth G. Haplotype-based variant detection from short-read sequencing. 2012
Jul 17; Available from: <http://arxiv.org/abs/1207.3907>
- 48. Aidley J, Wanford JJ, Green LR, Sheppard SK, Bayliss CD. Phasomeit: An ‘omics’
approach to cataloguing the potential breadth of phase variation in the genus

campylobacter. *Microb Genomics*. 2018;4(11).

**Table 1.** *C jejuni* strains isolated from patients with GBS including isolation year, GBS
subtypes, LOS class, presence of *cst-II/cst-III* genes, and presence of Thr/Asn at 51st positions of
*cst-II* gene.

C. jejuni strains	Year of isolation	Disease	GBS subtypes	LOS class	cst-II/cst-III	Thr/Asn at 51 st position
BD10	2006	GBS	AMAN	B	cst-II	Asn
BD22	2006	GBS	AMAN	B	cst-II	Asn
BD27	2006	GBS	AMAN	B	cst-II	Asn
BD34	2006	GBS	AMAN	B	cst-II	Asn
BD-39	2006	GBS	AMAN	A	cst-II	Thr
BD-67	2007	GBS	AMAN	B	cst-II	Asn
BD74	2007	GBS	AIDP	B	cst-II	Asn
BD-75	2007	GBS	N/D	A	cst-II	Thr
BD94	2007	GBS	AMAN	W	Absence	N/A
ZH1	2010	GBS	AMAN	C	cst-III	N/A
ZH2	2010	GBS	AMAN	B	cst-II	Thr
ZH6	2010	GBS	AMAN	A	cst-II	Thr
ZH7	2010	GBS	N/D	A	cst-II	Thr
ZH8	2011	GBS	AMAN	B	cst-II	Asn
ZH9	2011	GBS	AIDP	C	cst-III	N/A
ZH10	2011	GBS	AMAN	C	cst-III	N/A
ZH11	2011	GBS	AMAN	C	cst-III	N/A
ZH12	2011	GBS	AMAN	B	cst-II	Thr
ZH14	2011	GBS	AMAN	B	cst-II	Thr
ZH15	2012	GBS	Equivocal	B	cst-II	Asn
ZH16	2012	GBS	Inexcitable	B	cst-II	Asn
ZH18	2012	GBS	AMSAN	A	cst-II	Thr
ZH19	2012	GBS	AMAN	A	cst-II	Thr

ZH20	2012	GBS	AMAN	A	cst-II	Asn
Z191005RS	2019	GBS	AMAN	C	cst-III	N/A
Z191005SS	2019	GBS	AMAN	C	cst-III	N/A
Z201020SS	2020	GBS	AMAN	A	cst-II	Asn
Z201020RS	2020	GBS	AMAN	A	cst-II	Asn
Z211080RS	2021	GBS	AMAN	UD	Absence	N/A
Z211080SS	2021	GBS	AMAN	UD	Absence	N/A
Z211097RS	2021	GBS	AMAN	M	cst-II	Asn
Z211097SS	2021	GBS	AMAN	M	cst-II	Asn
Z221123RS	2022	GBS	AMAN	A	cst-II	Asn
Z221123SS	2022	GBS	AMAN	A	cst-II	Asn
Z221130RS	2022	GBS	AMAN	F	Absence	N/A
Z221146RS	2022	GBS	AMAN	B	cst-III	Asn
Z221146SS	2022	GBS	AMAN	B	cst-III	Asn
Z227001SS	2022	GBS	N/D	A	cst-III	Asn

*GBS, Guillain-Barré syndrome; AMAN, Acute motor axonal neuropathy; AIDP, acute*
*inflammatory demyelinating polyradiculoneuropathy; UD, Undetermined; Thr, Threonine; Asn,*
*asparagine; N/D, not done; N/A, not applicable*

**Fig. 1. Sequence Type (ST) and Clonal Complexes (CC) of intended 64 *C. jejuni*.** (A)
Neighbor-joining tree based on concatenated nucleotide sequence loci of *Campylobacter jejuni*
allele from PubMLST database. (B) Venn diagram of Sequence Type (ST) diversity among GBS
and enteritis-associated *C. jejuni* strains. (C) Venn diagram of Clonal complex (CC) diversity
among GBS and enteritis-associated *C. jejuni* strains.

**Fig. 2.** Phylogenetic tree of LOS region of *C. jejuni* strains isolated from GBS and enteritis
patients along with reference strain NCTC11168.

**Fig. 3.** Comparative genome alignment of LOS biosynthesis locus region between GBS (Fig.
3A) and enteritis-associated *C. jejuni* strains (Fig. 3B). LOS region of *C. jejuni* NCTC11168 was
used as the reference genome. Colored arrows indicate the coding sequences and various colour-
reflected homologues groups identified by the clinker. Grey/white arrows represent the
hypothetical proteins. The black and grey bars represent the percentage of amino acid identity as
indicated in the figure.

**Fig. 4.** Distribution of nucleotide variations within the LOS region between GBS-associated and
enteritis-associated *C.jejuni*. (A) Distribution of total nucleotide variations, (B) Distribution of
SNPs, (C) Distribution of MNPs, (D) Distribution of complex (combination of SNP/MNP).

**Fig. 5.** The presence of phase variation within the LOS region of GBS-associated and enteritis-
associated strains.

**Genomic Variability of Lipooligosaccharide Biosynthesis Locus and Sequence Type among**
***Campylobacter jejuni* Isolated from Patients with Guillain-Barre´ Syndrome**

Md. Abu Jaher Nayeem,^a Shoma Hayat,^a Asaduzzaman Asad,^a Md. Golam Mostafa,^a Shah
Nayeem Faruque,^a Ruma Begum,^a Mosabbir Ahmed,^a Yearul Kabir,^b Israt Jahan,^{a,c} Zhahirul
Islam,^{a*}

7 ^aGut-Brain Axis Laboratory, Infectious Diseases Division (IDD), icddr,b, Dhaka 1212,
Bangladesh.

9 ^bDepartment of Biochemistry and Molecular Biology, University of Dhaka, Dhaka 1000,
Bangladesh.

11 ^cDepartment of Medical Microbiology and Infectious Diseases, Erasmus University Medical
Center, Rotterdam, The Netherlands.

Running Title: Lipooligosaccharide and sequence types in *C. jejuni* associated with GBS

***Corresponding author**

Zhahirul Islam, Ph.D.

Gut-Brain Axis Laboratory

Infectious Diseases Division (IDD)

icddr,b, Dhaka, Bangladesh

68, Shaheed Tajuddin Ahmad Sarani, Mohakhali, Dhaka-1212, Bangladesh

Phone: +880 2 9886464, Fax: +880 2 8812529

E-mail: zislam@icddr.org

**ABSTRACT**

Lipooligosaccharide (LOS) is a crucial component of the *Campylobacter jejuni* (*C. jejuni*) cell
membrane that mimics human gangliosides and can induce autoimmune neuropathies like
Guillain-Barré syndrome (GBS). Approximately 1 in 1000 *C. jejuni* infections lead to GBS, with
LOS variations influencing immune response and disease induction. We aimed to investigate the
sequence type and genomic variation of the LOS region of *C. jejuni* strains isolated from patients
with GBS and enteritis. We isolated 38 GBS- and 24 enteritis-associated *C. jejuni* strains from
Bangladesh, sequenced them on the Illumina platform and compared their LOS regions using
various bioinformatics approaches. Comparative genomic analysis between GBS-associated and
enteritis-associated strains revealed that LOS class A, B and C were prevalent ($p = 0.0001$) in
GBS-associated *C. jejuni* strains. Sequence types ST-43 and ST-2042 and clonal complexes ST-
403 cc and ST-21 cc were overrepresented in GBS compared to enteritis-associated strains. All
GBS-associated strains carrying LOS class C belonged to clonal complex ST-21 cc. A
significantly higher ($p = 0.025$) number of nucleotide variations (NVs) were detected within the
LOS region of GBS-associated strains. In GBS strains, phase variation (PV) within the LOS
region was significantly higher ($p = 0.004$). In conclusion, the over-representation of GBS-
specific LOS classes with higher rates of nucleotide and phase variations might be the
determining factor for GBS. However, further research on integrating DNA epigenetic analysis
and metabolic profiling is needed to establish the *C. jejuni*-specific risk factors of GBS.

**IMPORTANCE**

The lipooligosaccharide (LOS) biosynthesis region is the most variable and virulent genomic
region of *Campylobacter jejuni*. This study addressed the critical gap in understanding the
genomic features contributing to GBS by analyzing the LOS region of *C. jejuni* strains from

GBS and enteritis patients. Our analysis of *C. jejuni* isolates from GBS and enteritis cases
revealed genomic variations in the sequence type and LOS region, including distinct LOS classes
(A, B, C), higher prevalence of the *cst-II* gene and significant presence of phase variability
among GBS-associated *C. jejuni* strains. The findings highlight the association of *C. jejuni* LOS-
region in the pathogenesis of GBS. In addition, the results of this study will facilitate future
comparisons by identifying genomic differences in lipooligosaccharide biosynthesis and
sequence types between *C. jejuni* strains linked to GBS. This will enhance our understanding of
the genomic variation of the LOS region and their role in GBS development.

**KEYWORDS**

Guillain-Barré syndrome, *Campylobacter jejuni*, lipooligosaccharide, comparative genomics,
phase variation

Introduction

*Campylobacter jejuni* (*C. jejuni*) is a leading cause of gastroenteritis and has been causatively
associated with the onset of peripheral autoimmune neuropathy such as Guillain-Barré syndrome
(GBS) (1). This association is linked to specific genetic and structural variations within the
bacterial strains, particularly within the lipooligosaccharide (LOS) locus. LOS is the most
important virulence factor of *C. jejuni*, structurally mimicking human gangliosides and thereby
triggering an autoimmune response that can lead to GBS (2). *C. jejuni* infection may develop
GBS in approximately 1 in 1000 enteritis (ENT) patients as a result of a cross-reactive immune
response that induces anti-ganglioside antibodies (3–5). In Bangladesh, *C. jejuni* is recognized as
the most prevalent pathogen associated with GBS (3). The onset of autoimmune neuropathies
following *C. jejuni* infection is primarily due to molecular mimicry between LOS on *C. jejuni*'s
cell surface and gangliosides on human peripheral nerves (6,7). Previous studies have reported
that *C. jejuni* LOS loci class A, B, and C, as well as the serotype *C. jejuni* HS:23 and clonal
complex ST-403 cc, were associated with a high frequency of *C. jejuni*-associated GBS (8–10).
Moreover, no studies have conducted a comparative sequence-based approach to examine
genomic variations within the LOS region of *C. jejuni*.

Previously, a comparative genotyping approach revealed remarkable diversity among the
sequence types (STs) and clonal complexes (CCs) of various *C. jejuni* strains isolated from
patients with enteritis and patients with GBS in Bangladesh (8). This diversity highlights the
genetic variability of *C. jejuni* strains and their different associations with gastrointestinal and
neurological diseases (11). Cohort-based studies on GBS-associated *C. jejuni* cases have
indicated that genomic variation in the LOS region of *C. jejuni* may be responsible for the
endemicity of GBS in Bangladesh (3,8). These studies suggest that specific genetic differences

within the LOS locus contribute to the high prevalence of GBS in this region, underscoring the
need for further research into these genomic variations. Biochemical and structural analyses of
the LOS outer core oligosaccharides of *C. jejuni* have revealed sialylated moieties structurally
similar to various gangliosides (2,12). This structural resemblance, known as molecular mimicry,
facilitates the production of anti-ganglioside antibodies during infection and contributes to the
onset of GBS (2,13,14). The LOS region of *C. jejuni* isolated from GBS patients has been shown
to mimic the peripheral nerve gangliosides, mainly GM1, GD1a, and GQ1b (15). The
sialyltransferase (*cst-II*) gene is a key in adding neuraminic acid to LOS which is linked to GBS.
Strains with *cst-II* (Thr51) express GM1- and GD1a-like LOS, while *cst-II* (Asn51) variants
produce GT1a- and GD1c-like LOS, influencing immune responses and GBS risk (16). During
infection, the induction of these LOS may prompt antibodies to bind to similar ganglioside
structures on peripheral nerves. This cross-reactivity can trigger complement-mediated immune
activation, resulting in nerve damage and the onset of GBS (17).

*C. jejuni* LOS biosynthesis region displays considerable genetic variation among strains,
featuring notable differences in the arrangement of genes which encode LOS carbohydrate
moieties and their linkages. Furthermore, there is significant structural variety between strains
due to the susceptibility of the LOS gene locus to periodic phase fluctuations (18,19). There are
notable differences, especially in the outer section of the oligosaccharide moiety, with variations
in lipid A phosphorylation and the number of amide linkages (20). The complete genome
sequence analysis of *C. jejuni* NCTC 11168 revealed the involvement of a cluster of genes that
extends from Cj1131c (*galE*) to Cj1151c (*rfaD*) with lower GC content (18,21,22). The LOS
biosynthesis locus, a gene cluster that is genetically quite varied and interchangeable between
strains, harbors the *C. jejuni* genes involved in ganglioside mimicry (23). A cross-reactive

antibody response is induced in nerve tissue due to the molecular similarity between
*Campylobacter* LOS and nerve gangliosides; although, the precise pathophysiology of post-
*Campylobacter* neuropathy, like GBS, is unclear (13). The variations in LOS class, gene
alterations, mutations, and mechanisms such as phase variation (PV) within the LOS locus
contribute to structural variations in the ganglioside mimics (18,19,23).

[revised manuscript text omitted]

**Extraction of genomic DNA and whole-genome sequencing**

The genomic DNA of 50 *C. jejuni* strains (GBS strain, $n = 38$ and enteritis strains, $n = 12$) were
extracted using the Wizard® genomic DNA purification kit (Promega, USA) according to the
manufacturer's instructions. The quality of DNA was assessed using NanoDrop
spectrophotometer (Thermo Scientific, USA) and quantified with Qubit 2.0 fluorimeter (Life
Technologies). Whole-genome sequencing was performed using Illumina technology at the
icddr,b Genome Centre (iGC), and the sequences were processed and assembled using a
previously described approach (33). Sequence coverage was checked by mapping the reads to
reference sequence with Burrows-Wheeler Aligner (BWA) v0.7.18 and coverage was calculated

using SAMtools v1.20 (34,35). The quality of the assembled genome was assessed using
QUAST v5.2.0 (36). Confirmation of genus and detection of possible contaminated contigs was
carried out on Kraken v1.1.1 (37). The assemblies were further filtered to remove contigs <
200bp with a k-mer coverage of $\geq 5X$. The genome sequences are publicly available on the
NCBI-SRA database and are associated with the BioProjects PRJNA378546, PRJNA717137,
and PRJNA607058 respectively (Table S1). Annotation was performed on the NCBI Prokaryotic
Genome Annotation Pipeline (PGAP) v6.7 and rapid annotation using subsystem technology
(RASTtk) (<https://rast.nmpdr.org/rast.cgi>) platform using default parameters (38,39). Sequence
type and clonal complex were determined using the PubMLST database (40).

**Identification of LOS biosynthesis gene cluster**

The LOS region was extracted from each whole genome through alignment with *C. jejuni*
reference genome NCTC11168 using minimap2 (-c --cs -x asm20) (41). The LOS region in the
*Campylobacter jejuni* reference genome (NCTC11168) is located between two specific genes:
the *Cj1131c* (*galE*) gene and the *Cj1151c* (*rfaD*) gene (18). The coordinate range
(AL11168.1:1064895-1084736) was then lifted over using minimap2 utility paftools.js (with
“liftover”) and the specified genomic region was separated from each genome with SAMtools
(42). Multiple sequence alignment of LOS loci was performed for all GBS, enteritis, and
reference genomes NCTC11168 and ICDCJ07001. The extracted LOS region sequences were
annotated in the RAST annotation server (39). After annotation, each open reading frame (ORF)
was manually verified for their homology to specific genes from the NCBI database using blastp
(<https://blast.ncbi.nlm.nih.gov/Blast.cgi>).

***In silico* LOS classification**

The genome sequences of the LOS region from reference strains and subject strains (GBS and
enteritis) were aligned using Nucleotide BLAST (<https://blast.ncbi.nlm.nih.gov>). The LOS
classification was performed following the methodology and references described by *Hameed et*
*al.* 2022 (43). The LOS classes were confirmed by sequence alignment with $\geq 90\%$ coverage and
$\geq 90\%$ nucleotide identity.

**Genome alignment and phylogenetic analysis of LOS biosynthesis locus**

The annotated data of each LOS region was compared with blastn and visualized in clinker and
clustermap.js using the GenBank (GBK) file of the specified LOS region of each *C. jejuni*
genome (44). A phylogenetic tree was developed to understand the genomic variation of the LOS
region in GBS-associated and enteritis-associated *C. jejuni* strains. The LOS regions of *C. jejuni*
isolate were aligned on MAFFT online server version 7 using default parameters, and UPGMA
clustering was performed on the sequences of the LOS locus (45). The phylogenetic tree was
visualized using iTOL v6.8.1 (46).

**Variation of nucleotide in *Campylobacter jejuni* LOS biosynthesis locus**

Nucleotide variations in the LOS biosynthesis locus region were detected through SNIPPY
v4.6.0 (<https://github.com/tseemann/snippy>). Snippy uses the BWA-MEM algorithm to perform
local alignment, and SAMtool for parsing and manipulating alignment in SAM/BAM format.
Freebayes was used for variant calling (47). An unpaired t-test was performed using GraphPad
Prism v9 to compare the number of unique single nucleotide polymorphisms (SNPs), multiple
nucleotide polymorphisms (MNPs), complex polymorphisms (combinations of SNPs/MNPs),
and total nucleotide variations within the LOS region between GBS-associated and enteritis-
associated strains.

**Identification of phase variable genes within the LOS locus of *C. jejuni***

PhasomeIt, a novel program which identifies phase variable genes in *C. jejuni* genome through
association with simple sequence repeats was used to detect Phase Variable (PV) genes in the
LOS locus. To identify all putatively phase-variable loci in *C. jejuni* LOS regions, a cut-off value
of '7 6 0 5 5' and filter cutoff of 'W9' was used (48). Fisher's exact test was performed for 'G/C'
tracts containing PV genes of strains isolated from GBS and enteritis patients, depending on
whether their PV genes are ON or OFF.

***Acknowledgements***

This research activity was funded by the Fogarty International Center (FIC), National Institute of
Neurological Disorders and Stroke (NINDS) of the National Institutes of Health (NIH), USA
under Award Number K43TW011447. icddr,b gratefully acknowledges the commitment of the
Government of The People's Republic of Bangladesh and Global Affairs Canada for their
unrestricted support. We acknowledge all study personnel who contributed to patient enrollment
and specimen collection; data management and laboratory support. We are indebted to the
neurologists who referred their patients to this genomic study. We are thankful to the patients
who participated in the study and provided their valuable data.

***Author Contributions***

ZI, MAJN and SH conceptualized and designed the study. The research methodology and
execution plan were conceived by MAJN, SH and AA. MAJN, AA and MGM contributed to
data acquisition and genomic analysis. ZI contributed to the acquisition of funding. MAJN, SNF,
RB, MA, and AA did materials acquisition, laboratory experiments and data acquisition. MAJN
drafted the manuscript, which was critically reviewed by SH, AA, MGM, SNF, RB, MA, YK, IJ,
and ZI for intellectual content. All authors read and approved the final manuscript before
submission.

***Materials and Correspondence***

The correspondence and material requests would be addressed to ZI.

***Conflicts of interest***

The authors declare that there are no conflicts of interest.

***Ethical statement***

Each participant provided written informed consent, and all studies were carried out following
relevant guidelines and regulations. The study protocol was reviewed and approved by the
Institutional Review Board (IRB) and the Ethical Review Committee (ERC) of icddr,b, Dhaka,
Bangladesh under protocol number PR-19048.

**Data availability**

The authors confirm that all supporting data, and protocols have been provided within the article
or through supplementary data files. The raw sequencing data for this project and primary
assemblies are made publicly available under projects PRJNA378546, PRJNA717137, and
PRJNA607058. The accession numbers are provided in supplementary file **Table S1** which is
available in the online version of this article. Other metadata about study subjects and genome
sequences are provided in **Table 1**.

**References**

- 1. Malik A, Brudvig JM, Gadsden BJ, Ethridge AD, Mansfield LS. *Campylobacter jejuni*
induces autoimmune peripheral neuropathy via Sialoadhesin and Interleukin-4 axes. *Gut*
*Microbes*. 2022;14(1).
- 2. Islam Z, Gilbert M, Mohammad QD, Klaij K, Li J, van Rijs W, et al. Guillain-Barré
Syndrome-Related *Campylobacter jejuni* in Bangladesh: Ganglioside Mimicry and Cross-
Reactive Antibodies. Bereswill S, editor. *PLoS One* [Internet]. 2012 Aug 27;7(8):e43976.
Available from: <https://dx.plos.org/10.1371/journal.pone.0043976>
- 3. Islam Z, Jacobs BC, Van Belkum A, Mohammad QD, Islam MB, Herbrink P, et al.
Axonal variant of Guillain-Barré syndrome associated with *Campylobacter* infection in
Bangladesh. *Neurology*. 2010;74(7):581–7.
- 4. Allos BM. Association between *Campylobacter* infection and Guillain-Barre syndrome. *J*
*Infect Dis*. 1997;176(6 SUPPL.).
- 5. Mccarthy N, Andersson Y, Jormanainen V, Gustavsson O, Giesecke J. The risk of
Guillain-Barre syndrome following infection with *Campylobacter jejuni*. *Epidemiol*
*Infect*. 1999;122(1):15–7.
- 6. Godschalk PCR, Gilbert M, Jacobs BC, Kramers T, Tio-Gillen AP, Ang CW, et al. Co-
infection with two different *Campylobacter jejuni* strains in a patient with the Guillain-
Barré syndrome. *Microbes Infect* [Internet]. 2006 Jan;8(1):248–53. Available from:
<https://linkinghub.elsevier.com/retrieve/pii/S1286457905002601>
- 7. Godschalk PCR, Kuijf ML, Li J, St. Michael F, Ang CW, Jacobs BC, et al. Structural
characterization of *Campylobacter jejuni* lipooligosaccharide outer cores associated with
Guillain-Barré and Miller Fisher syndromes. *Infect Immun*. 2007;75(3):1245–54.
- 8. Islam Z, van Belkum A, Wagenaar JA, Cody AJ, de Boer AG, Tabor H, et al.
Comparative genotyping of *Campylobacter jejuni* strains from patients with Guillain-
Barré syndrome. *PLoS One*. 2009;4(9).
- 9. Heikema AP, Strepis N, Horst-Kreft D, Huynh S, Zomer A, Kelly DJ, et al. Biomolecule
sulphation and novel methylations related to Guillain-Barré syndrome-associated
*Campylobacter jejuni* serotype HS: 19. *Microb Genomics*. 2021;7(11):660.
- 10. Godschalk PCR, Heikema AP, Gilbert M, Komagamine T, Ang CW, Glerum J, et al. The
crucial role of *Campylobacter jejuni* genes in anti-ganglioside antibody induction in
Guillain-Barre syndrome. *J Clin Invest*. 2004;114(11):1659–65.
- 11. Djeghout B, Bloomfield SJ, Rudder S, Elumogo N, Mather AE, Wain J, et al.
Comparative genomics of *Campylobacter jejuni* from clinical campylobacteriosis stool
specimens. *Gut Pathog* [Internet]. 2022 Dec 7;14(1):45. Available from:
<https://gutpathogens.biomedcentral.com/articles/10.1186/s13099-022-00520-1>
- 12. Bowes T, Wagner ER, Boffey J, Nicholl D, Cochrane L, Benboubetra M, et al. Tolerance
to Self Gangliosides Is the Major Factor Restricting the Antibody Response to
Lipopolysaccharide Core Oligosaccharides in *Campylobacter jejuni* Strains Associated

- with Guillain-Barré Syndrome. *Infect Immun* [Internet]. 2002 Sep;70(9):5008–18.
 Available from: <https://journals.asm.org/doi/10.1128/IAI.70.9.5008-5018.2002>
- 13. Yuki N. Molecular mimicry between gangliosides and lipopolysaccharides of
 *Campylobacter jejuni* isolated from patients with Guillain-Barre syndrome and Miller
 Fisher syndrome. *J Infect Dis*. 1997;176(6 SUPPL.):S150–3.
- 14. Yu RK, Usuki S, Ariga T. Ganglioside Molecular Mimicry and Its Pathological Roles in
 Guillain-Barré Syndrome and Related Diseases. *Infect Immun* [Internet]. 2006
 Dec;74(12):6517–27. Available from: <https://journals.asm.org/doi/10.1128/IAI.00967-06>
- 15. Hughes RAC, Cornblath DR. Guillain-barre syndrome. *Lancet*. 2005;366(9497):1653–66.
- 16. Yuki N. *Campylobacter* sialyltransferase gene polymorphism directs clinical features of
 Guillain–Barré syndrome. *J Neurochem* [Internet]. 2007 Nov 6;103(s1):150–8. Available
 from: <https://onlinelibrary.wiley.com/doi/10.1111/j.1471-4159.2007.04707.x>
- 17. Laman JD, Huizinga R, Boons G-J, Jacobs BC. Guillain-Barré syndrome: expanding the
 concept of molecular mimicry. *Trends Immunol* [Internet]. 2022 Apr;43(4):296–308.
 Available from: <https://linkinghub.elsevier.com/retrieve/pii/S147149062200028X>
- 18. Parker CT, Horn ST, Gilbert M, Miller WG, Woodward DL, Mandrell RE. Comparison of
 *Campylobacter jejuni* Lipooligosaccharide Biosynthesis Loci from a Variety of Sources. *J*
 *Clin Microbiol* [Internet]. 2005 Jun;43(6):2771–81. Available from:
 <https://journals.asm.org/doi/10.1128/JCM.43.6.2771-2781.2005>
- 19. Parker CT, Gilbert M, Yuki N, Endtz HP, Mandrell RE. Characterization of
 lipooligosaccharide-biosynthetic loci of *Campylobacter jejuni* reveals new
 lipooligosaccharide classes: Evidence of mosaic organizations. *J Bacteriol*.
 2008;190(16):5681–9.
- 20. Stephenson HN, John CM, Naz N, Gundogdu O, Dorrell N, Wren BW, et al.
 *Campylobacter jejuni* Lipooligosaccharide Sialylation, Phosphorylation, and Amide/Ester
 Linkage Modifications Fine-tune Human Toll-like Receptor 4 Activation. *J Biol Chem*
 [Internet]. 2013 Jul;288(27):19661–72. Available from:
 <https://linkinghub.elsevier.com/retrieve/pii/S0021925820456832>
- 21. Linton D, Karlyshev A V., Hitchen PG, Morris HR, Dell A, Gregson NA, et al. Multiple
 N-acetyl neuraminic acid synthetase (*neuB*) genes in *Campylobacter jejuni*: Identification
 and characterization of the gene involved in sialylation of lipo-oligosaccharide. *Mol*
 *Microbiol*. 2000;35(5):1120–34.
- 22. Parkhill J, Wren BW, Mungall K, Ketley JM, Churcher C, Basham D, et al. The genome
 sequence of the food-borne pathogen *Campylobacter jejuni* reveals hypervariable
 sequences. *Nature* [Internet]. 2000;403(6770):665–8. Available from:
 [file://localhost/Users/cliftonfranklund/Documents/Journal Articles/Parkhill\(2000\).pdf](file://localhost/Users/cliftonfranklund/Documents/Journal%20Articles/Parkhill(2000).pdf)
- 23. Gilbert M, Karwaski MF, Bernatchez S, Young NM, Taboada E, Michniewicz J, et al. The
 genetic bases for the variation in the lipo-oligosaccharide of the mucosal pathogen,
 *Campylobacter jejuni*. Biosynthesis of sialylated ganglioside mimics in the core
 oligosaccharide. *J Biol Chem*. 2002;277(1):327–37.

- 24. Semchenko EA, Day CJ, Moutin M, Wilson JC, Tiralongo J, Korolik V. Structural
heterogeneity of terminal glycans in *Campylobacter jejuni* Lipooligosaccharides. *PLoS*
*One*. 2012;7(7).
- 25. Parker CT, Gilbert M, Yuki N, Endtz HP, Mandrell RE. Characterization of
lipooligosaccharide-biosynthetic loci of *Campylobacter jejuni* reveals new
lipooligosaccharide classes: Evidence of mosaic organizations. *J Bacteriol*.
2008;190(16):5681–9.
- 26. Heikema AP, Islam Z, Horst-Kreft D, Huizinga R, Jacobs BC, Wagenaar JA, et al.
*Campylobacter jejuni* capsular genotypes are related to Guillain-Barré syndrome. *Clin*
*Microbiol Infect*. 2015;21(9):852.e1-852.e9.
- 27. Koga M, Gilbert M, Takahashi M, Li J, Koike S, Hirata K, et al. Comprehensive analysis
of bacterial risk factors for the development of Guillain-Barré syndrome after
*Campylobacter jejuni* enteritis. *J Infect Dis*. 2006;193(4):547–55.
- 28. Heikema AP, Strepis N, Horst-Kreft D, Huynh S, Zomer A, Kelly DJ, et al. Biomolecule
sulphation and novel methylations related to guillain-barré syndrome-associated
*campylobacter jejuni* serotype hs:19. *Microb Genomics*. 2021;7(11).
- 29. Quino W, Caro-Castro J, Mestanza O, Hurtado V, Zamudio ML, Cruz-Gonzales G, et al.
Emergence and Molecular Epidemiology of *Campylobacter jejuni* ST-2993 Associated
with a Large Outbreak of Guillain-Barré Syndrome in Peru. Denes TG, editor. *Microbiol*
*Spectr* [Internet]. 2022 Oct 26;10(5). Available from:
<https://journals.asm.org/doi/10.1128/spectrum.01187-22>
- 30. Gilbert M, Karwaski MF, Bernatchez S, Young NM, Taboada E, Michniewicz J, et al. The
genetic bases for the variation in the lipo-oligosaccharide of the mucosal pathogen,
*Campylobacter jejuni*. Biosynthesis of sialylated ganglioside mimics in the core
oligosaccharide. *J Biol Chem*. 2002;277(1):327–37.
- 31. Guerry P, Szymanski CM, Prendergast MM, Hickey TE, Ewing CP, Pattarini DL, et al.
Phase variation of *Campylobacter jejuni* 81-176 lipooligosaccharide affects ganglioside
mimicry and invasiveness in vitro. *Infect Immun*. 2002;70(2):787–93.
- 32. Islam Z, Nabila FH, Asad A, Begum R, Jahan I, Hayat S, et al. Draft Genome Sequences
of Three Strains of *Campylobacter jejuni* Isolated from Patients with Guillain-Barré
Syndrome in Bangladesh. Rasko D, editor. *Microbiol Resour Announc* [Internet]. 2021
Apr 29;10(17). Available from: <https://journals.asm.org/doi/10.1128/MRA.00005-21>
- 33. Hayat S, Nabila FH, Asad A, Begum R, Jahan I, Endtz HP, et al. Draft Genome
Sequences of Four Strains of *Campylobacter jejuni* Isolated from Patients with Axonal
Variant of Guillain-Barré Syndrome in Bangladesh. Putonti C, editor. *Microbiol Resour*
*Announc* [Internet]. 2022 Feb 17;11(2). Available from:
<https://journals.asm.org/doi/10.1128/mra.01146-21>
- 34. Heng L, Richard D. Fast and accurate short read alignment with Burrows-Wheeler
Transform. *Bioinformatics*. 2009;25(14):1754–60.
- 35. Li H, Handsaker B, Wysoker A, Fennell T, Ruan J, Homer N, et al. The Sequence

- Alignment/Map format and SAMtools. *Bioinformatics* [Internet]. 2009 Aug
15;25(16):2078–9. Available from:
<https://academic.oup.com/bioinformatics/article/25/16/2078/204688>
- 36. Gurevich A, Saveliev V, Vyahhi N, Tesler G. QUASt: Quality assessment tool for
genome assemblies. *Bioinformatics*. 2013;29(8):1072–5.
- 37. Wood DE, Salzberg SL. Kraken: ultrafast metagenomic sequence classification using
exact alignments. *Genome Biol* [Internet]. 2014 Mar 3;15(3):R46. Available from:
<https://genomebiology.biomedcentral.com/articles/10.1186/gb-2014-15-3-r46>
- 38. Tatusova T, DiCuccio M, Badretdin A, Chetvernin V, Nawrocki EP, Zaslavsky L, et al.
NCBI prokaryotic genome annotation pipeline. *Nucleic Acids Res* [Internet]. 2016 Aug
19;44(14):6614–24. Available from: [https://academic.oup.com/nar/article-](https://academic.oup.com/nar/article-lookup/doi/10.1093/nar/gkw569)
[lookup/doi/10.1093/nar/gkw569](https://academic.oup.com/nar/article-lookup/doi/10.1093/nar/gkw569)
- 39. Aziz RK, Bartels D, Best A, DeJongh M, Disz T, Edwards RA, et al. The RAST Server:
Rapid annotations using subsystems technology. *BMC Genomics*. 2008;9.
- 40. Jolley KA, Bray JE, Maiden MCJ. Open-access bacterial population genomics: BIGSdb
software, the PubMLST.org website and their applications. *Wellcome Open Res*
[Internet]. 2018 Sep 24;3:124. Available from:
<https://wellcomeopenresearch.org/articles/3-124/v1>
- 41. Li H. New strategies to improve minimap2 alignment accuracy. *Bioinformatics*.
2021;37(23):4572–4.
- 42. Danecek P, JK B, J L, J M, V O, MO P, et al. Twelve years of SAMtools and BCFtools.
*Gigascience*. 2021;10(2).
- 43. Hameed A, Woodacre A, Machado LR, Marsden GL. An Updated Classification System
and Review of the Lipooligosaccharide Biosynthesis Gene Locus in *Campylobacter jejuni*
[Internet]. Vol. 11, *Frontiers in Microbiology*. 2020. p. 677. Available from:
<https://www.frontiersin.org/article/10.3389/fmicb.2020.00677>
- 44. Gilchrist CLM, Chooi YH. Clinker & clustermap.js: Automatic generation of gene cluster
comparison figures. *Bioinformatics*. 2021;37(16):2473–5.
- 45. Katoh K, Standley DM. MAFFT Multiple Sequence Alignment Software Version 7:
Improvements in Performance and Usability. *Mol Biol Evol* [Internet]. 2013 Apr
1;30(4):772–80. Available from: [https://academic.oup.com/mbe/article-](https://academic.oup.com/mbe/article-lookup/doi/10.1093/molbev/mst010)
[lookup/doi/10.1093/molbev/mst010](https://academic.oup.com/mbe/article-lookup/doi/10.1093/molbev/mst010)
- 46. Letunic I, Bork P. Interactive Tree Of Life (iTOL) v5: an online tool for phylogenetic tree
display and annotation. *Nucleic Acids Res* [Internet]. 2021 Jul 2;49(W1):W293–6.
Available from: <https://academic.oup.com/nar/article/49/W1/W293/6246398>
- 47. Garrison E, Marth G. Haplotype-based variant detection from short-read sequencing. 2012
Jul 17; Available from: <http://arxiv.org/abs/1207.3907>
- 48. Aidley J, Wanford JJ, Green LR, Sheppard SK, Bayliss CD. Phasomeit: An ‘omics’
approach to cataloguing the potential breadth of phase variation in the genus

campylobacter. *Microb Genomics*. 2018;4(11).

**Table 1.** *C. jejuni* strains isolated from patients with GBS including isolation year, GBS
subtypes, LOS class, presence of *cst-II/cst-III* genes, and presence of Thr/Asn at 51st positions of
*cst-II* gene.

C. jejuni strains	Year of isolation	Disease	GBS subtypes	LOS class	cst-II/cst-III	Thr/Asn at 51 st position
BD10	2006	GBS	AMAN	B	cst-II	Asn
BD22	2006	GBS	AMAN	B	cst-II	Asn
BD27	2006	GBS	AMAN	B	cst-II	Asn
BD34	2006	GBS	AMAN	B	cst-II	Asn
BD-39	2006	GBS	AMAN	A	cst-II	Thr
BD-67	2007	GBS	AMAN	B	cst-II	Asn
BD74	2007	GBS	AIDP	B	cst-II	Asn
BD-75	2007	GBS	N/D	A	cst-II	Thr
BD94	2007	GBS	AMAN	W	Absence	N/A
ZH1	2010	GBS	AMAN	C	cst-III	N/A
ZH2	2010	GBS	AMAN	B	cst-II	Thr
ZH6	2010	GBS	AMAN	A	cst-II	Thr
ZH7	2010	GBS	N/D	A	cst-II	Thr
ZH8	2011	GBS	AMAN	B	cst-II	Asn
ZH9	2011	GBS	AIDP	C	cst-III	N/A
ZH10	2011	GBS	AMAN	C	cst-III	N/A
ZH11	2011	GBS	AMAN	C	cst-III	N/A
ZH12	2011	GBS	AMAN	B	cst-II	Thr
ZH14	2011	GBS	AMAN	B	cst-II	Thr
ZH15	2012	GBS	Equivocal	B	cst-II	Asn
ZH16	2012	GBS	Inexcitable	B	cst-II	Asn
ZH18	2012	GBS	AMSAN	A	cst-II	Thr
ZH19	2012	GBS	AMAN	A	cst-II	Thr

ZH20	2012	GBS	AMAN	A	cst-II	Asn
Z191005RS	2019	GBS	AMAN	C	cst-III	N/A
Z191005SS	2019	GBS	AMAN	C	cst-III	N/A
Z201020SS	2020	GBS	AMAN	A	cst-II	Asn
Z201020RS	2020	GBS	AMAN	A	cst-II	Asn
Z211080RS	2021	GBS	AMAN	UD	Absence	N/A
Z211080SS	2021	GBS	AMAN	UD	Absence	N/A
Z211097RS	2021	GBS	AMAN	M	cst-II	Asn
Z211097SS	2021	GBS	AMAN	M	cst-II	Asn
Z221123RS	2022	GBS	AMAN	A	cst-II	Asn
Z221123SS	2022	GBS	AMAN	A	cst-II	Asn
Z221130RS	2022	GBS	AMAN	F	Absence	N/A
Z221146RS	2022	GBS	AMAN	B	cst-III	Asn
Z221146SS	2022	GBS	AMAN	B	cst-III	Asn
Z227001SS	2022	GBS	N/D	A	cst-III	Asn

GBS, Guillain-Barré syndrome; AMAN, Acute motor axonal neuropathy; AIDP, acute
inflammatory demyelinating polyradiculoneuropathy; UD, Undetermined; Thr, Threonine; Asn,
asparagine; N/D, not done; N/A, not applicable

**Fig. 1. Sequence Type (ST) and Clonal Complexes (CC) of intended 64 *C. jejuni*.** (A)
Neighbor-joining tree based on concatenated nucleotide sequence loci of *Campylobacter jejuni*
allele from PubMLST database. (B) Venn diagram of Sequence Type (ST) diversity among GBS
and enteritis-associated *C. jejuni* strains. (C) Venn diagram of Clonal complex (CC) diversity
among GBS and enteritis-associated *C. jejuni* strains.

**Fig. 2.** Phylogenetic tree of LOS region of *C. jejuni* strains isolated from GBS and enteritis
patients along with reference strain NCTC11168.

**Fig. 3.** Comparative genome alignment of LOS biosynthesis locus region between GBS (Fig.
3A) and enteritis-associated *C. jejuni* strains (Fig. 3B). LOS region of *C. jejuni* NCTC11168 was
used as the reference genome. Colored arrows indicate the coding sequences and various colour-
reflected homologues groups identified by the clinker. Grey/white arrows represent the
hypothetical proteins. The black and grey bars represent the percentage of amino acid identity as
indicated in the figure.

**Fig. 4.** Distribution of nucleotide variations within the LOS region between GBS-associated and
enteritis-associated *C.jejuni*. (A) Distribution of total nucleotide variations, (B) Distribution of
SNPs, (C) Distribution of MNPs, (D) Distribution of complex (combination of SNP/MNP).

**Fig. 5.** The presence of phase variation within the LOS region of GBS-associated and enteritis-
associated strains.

**Genomic Variability of Lipooligosaccharide Biosynthesis Locus and Sequence Type among**
***Campylobacter jejuni* Isolated from Patients with Guillain-Barre´ Syndrome**

Md. Abu Jaher Nayeem,^a Shoma Hayat,^a Asaduzzaman Asad,^a Md. Golam Mostafa,^a Shah
Nayeem Faruque,^a Ruma Begum,^a Mosabbir Ahmed,^a Yearul Kabir,^b Israt Jahan,^{a,c} Zhahirul
Islam,^{a*}

7 ^aGut-Brain Axis Laboratory, Infectious Diseases Division (IDD), icddr,b, Dhaka 1212,
Bangladesh.

9 ^bDepartment of Biochemistry and Molecular Biology, University of Dhaka, Dhaka 1000,
Bangladesh.

11 ^cDepartment of Medical Microbiology and Infectious Diseases, Erasmus University Medical
Center, Rotterdam, The Netherlands.

Running Title: Lipooligosaccharide and sequence types in *C. jejuni* associated with GBS

***Corresponding author**

Zhahirul Islam, Ph.D.

Gut-Brain Axis Laboratory

Infectious Diseases Division (IDD)

icddr,b, Dhaka, Bangladesh

68, Shaheed Tajuddin Ahmad Sarani, Mohakhali, Dhaka-1212, Bangladesh

Phone: +880 2 9886464, Fax: +880 2 8812529

E-mail: zislam@icddr.org

**ABSTRACT**

Lipooligosaccharide (LOS) is a crucial component of the *Campylobacter jejuni* (*C. jejuni*) cell
membrane that mimics human gangliosides and can induce autoimmune neuropathies like
Guillain-Barré syndrome (GBS). Approximately 1 in 1000 *C. jejuni* infections lead to GBS, with
LOS variations influencing immune response and disease induction. We aimed to investigate the
sequence type and genomic variation of the LOS region of *C. jejuni* strains isolated from patients
with GBS and enteritis. We isolated 38 GBS- and 24 enteritis-associated *C. jejuni* strains from
Bangladesh, sequenced them on the Illumina platform and compared their LOS regions using
various bioinformatics approaches. Comparative genomic analysis between GBS-associated and
enteritis-associated strains revealed that LOS class A, B and C were prevalent ($p = 0.0001$) in
GBS-associated *C. jejuni* strains. Sequence types ST-43 and ST-2042 and clonal complexes ST-
403 cc and ST-21 cc were overrepresented in GBS compared to enteritis-associated strains. All
GBS-associated strains carrying LOS class C belonged to clonal complex ST-21 cc. A
significantly higher ($p = 0.025$) number of nucleotide variations (NVs) were detected within the
LOS region of GBS-associated strains. In GBS strains, phase variation (PV) within the LOS
region was significantly higher ($p = 0.004$). In conclusion, the over-representation of GBS-
specific LOS classes with higher rates of nucleotide and phase variations might be the
determining factor for GBS. However, further research on integrating DNA epigenetic analysis
and metabolic profiling is needed to establish the *C. jejuni*-specific risk factors of GBS.

**IMPORTANCE**

The lipooligosaccharide (LOS) biosynthesis region is the most variable and virulent genomic
region of *Campylobacter jejuni*. This study addressed the critical gap in understanding the
genomic features contributing to GBS by analyzing the LOS region of *C. jejuni* strains from

GBS and enteritis patients. Our analysis of *C. jejuni* isolates from GBS and enteritis cases
revealed genomic variations in the sequence type and LOS region, including distinct LOS classes
(A, B, C), higher prevalence of the *cst-II* gene and significant presence of phase variability
among GBS-associated *C. jejuni* strains. The findings highlight the association of *C. jejuni* LOS-
region in the pathogenesis of GBS. In addition, the results of this study will facilitate future
comparisons by identifying genomic differences in lipooligosaccharide biosynthesis and
sequence types between *C. jejuni* strains linked to GBS. This will enhance our understanding of
the genomic variation of the LOS region and their role in GBS development.

**KEYWORDS**

Guillain-Barré syndrome, *Campylobacter jejuni*, lipooligosaccharide, comparative genomics,
phase variation

Introduction

*Campylobacter jejuni* (*C. jejuni*) is a leading cause of gastroenteritis and has been causatively
associated with the onset of peripheral autoimmune neuropathy such as Guillain-Barré syndrome
(GBS) (1). This association is linked to specific genetic and structural variations within the
bacterial strains, particularly within the lipooligosaccharide (LOS) locus. LOS is the most
important virulence factor of *C. jejuni*, structurally mimicking human gangliosides and thereby
triggering an autoimmune response that can lead to GBS (2). *C. jejuni* infection may develop
GBS in approximately 1 in 1000 enteritis (ENT) patients as a result of a cross-reactive immune
response that induces anti-ganglioside antibodies (3–5). In Bangladesh, *C. jejuni* is recognized as
the most prevalent pathogen associated with GBS (3). The onset of autoimmune neuropathies
following *C. jejuni* infection is primarily due to molecular mimicry between LOS on *C. jejuni*'s
cell surface and gangliosides on human peripheral nerves (6,7). Previous studies have reported
that *C. jejuni* LOS loci class A, B, and C, as well as the serotype *C. jejuni* HS:23 and clonal
complex ST-403 cc, were associated with a high frequency of *C. jejuni*-associated GBS (8–10).
Moreover, no studies have conducted a comparative sequence-based approach to examine
genomic variations within the LOS region of *C. jejuni*.

Previously, a comparative genotyping approach revealed remarkable diversity among the
sequence types (STs) and clonal complexes (CCs) of various *C. jejuni* strains isolated from
patients with enteritis and patients with GBS in Bangladesh (8). This diversity highlights the
genetic variability of *C. jejuni* strains and their different associations with gastrointestinal and
neurological diseases (11). Cohort-based studies on GBS-associated *C. jejuni* cases have
indicated that genomic variation in the LOS region of *C. jejuni* may be responsible for the
endemicity of GBS in Bangladesh (3,8). These studies suggest that specific genetic differences

within the LOS locus contribute to the high prevalence of GBS in this region, underscoring the
need for further research into these genomic variations. Biochemical and structural analyses of
the LOS outer core oligosaccharides of *C. jejuni* have revealed sialylated moieties structurally
similar to various gangliosides (2,12). This structural resemblance, known as molecular mimicry,
facilitates the production of anti-ganglioside antibodies during infection and contributes to the
onset of GBS (2,13,14). The LOS region of *C. jejuni* isolated from GBS patients has been shown
to mimic the peripheral nerve gangliosides, mainly GM1, GD1a, and GQ1b (15). The
sialyltransferase (*cst-II*) gene is a key in adding neuraminic acid to LOS which is linked to GBS.
Strains with *cst-II* (Thr51) express GM1- and GD1a-like LOS, while *cst-II* (Asn51) variants
produce GT1a- and GD1c-like LOS, influencing immune responses and GBS risk (16). During
infection, the induction of these LOS may prompt antibodies to bind to similar ganglioside
structures on peripheral nerves. This cross-reactivity can trigger complement-mediated immune
activation, resulting in nerve damage and the onset of GBS (17).

*C. jejuni* LOS biosynthesis region displays considerable genetic variation among strains,
featuring notable differences in the arrangement of genes which encode LOS carbohydrate
moieties and their linkages. Furthermore, there is significant structural variety between strains
due to the susceptibility of the LOS gene locus to periodic phase fluctuations (18,19). There are
notable differences, especially in the outer section of the oligosaccharide moiety, with variations
in lipid A phosphorylation and the number of amide linkages (20). The complete genome
sequence analysis of *C. jejuni* NCTC 11168 revealed the involvement of a cluster of genes that
extends from Cj1131c (*galE*) to Cj1151c (*rfaD*) with lower GC content (18,21,22). The LOS
biosynthesis locus, a gene cluster that is genetically quite varied and interchangeable between
strains, harbors the *C. jejuni* genes involved in ganglioside mimicry (23). A cross-reactive

antibody response is induced in nerve tissue due to the molecular similarity between
*Campylobacter* LOS and nerve gangliosides; although, the precise pathophysiology of post-
*Campylobacter* neuropathy, like GBS, is unclear (13). The variations in LOS class, gene
alterations, mutations, and mechanisms such as phase variation (PV) within the LOS locus
contribute to structural variations in the ganglioside mimics (18,19,23).

[revised manuscript text omitted]

**Extraction of genomic DNA and whole-genome sequencing**

The genomic DNA of 50 *C. jejuni* strains (GBS strain, $n = 38$ and enteritis strains, $n = 12$) were
extracted using the Wizard® genomic DNA purification kit (Promega, USA) according to the
manufacturer's instructions. The quality of DNA was assessed using NanoDrop
spectrophotometer (Thermo Scientific, USA) and quantified with Qubit 2.0 fluorimeter (Life
Technologies). Whole-genome sequencing was performed using Illumina technology at the
icddr,b Genome Centre (iGC), and the sequences were processed and assembled using a
previously described approach (33). Sequence coverage was checked by mapping the reads to
reference sequence with Burrows-Wheeler Aligner (BWA) v0.7.18 and coverage was calculated

using SAMtools v1.20 (34,35). The quality of the assembled genome was assessed using
QUAST v5.2.0 (36). Confirmation of genus and detection of possible contaminated contigs was
carried out on Kraken v1.1.1 (37). The assemblies were further filtered to remove contigs <
200bp with a k-mer coverage of $\geq 5X$. The genome sequences are publicly available on the
NCBI-SRA database and are associated with the BioProjects PRJNA378546, PRJNA717137,
and PRJNA607058 respectively (Table S1). Annotation was performed on the NCBI Prokaryotic
Genome Annotation Pipeline (PGAP) v6.7 and rapid annotation using subsystem technology
(RASTtk) (<https://rast.nmpdr.org/rast.cgi>) platform using default parameters (38,39). Sequence
type and clonal complex were determined using the PubMLST database (40).

**Identification of LOS biosynthesis gene cluster**

The LOS region was extracted from each whole genome through alignment with *C. jejuni*
reference genome NCTC11168 using minimap2 (-c --cs -x asm20) (41). The LOS region in the
*Campylobacter jejuni* reference genome (NCTC11168) is located between two specific genes:
the *Cj1131c* (*galE*) gene and the *Cj1151c* (*rfaD*) gene (18). The coordinate range
(AL11168.1:1064895-1084736) was then lifted over using minimap2 utility paftools.js (with
“liftover”) and the specified genomic region was separated from each genome with SAMtools
(42). Multiple sequence alignment of LOS loci was performed for all GBS, enteritis, and
reference genomes NCTC11168 and ICDCJ07001. The extracted LOS region sequences were
annotated in the RAST annotation server (39). After annotation, each open reading frame (ORF)
was manually verified for their homology to specific genes from the NCBI database using blastp
(<https://blast.ncbi.nlm.nih.gov/Blast.cgi>).

***In silico* LOS classification**

The genome sequences of the LOS region from reference strains and subject strains (GBS and
enteritis) were aligned using Nucleotide BLAST (<https://blast.ncbi.nlm.nih.gov>). The LOS
classification was performed following the methodology and references described by *Hameed et*
*al.* 2022 (43). The LOS classes were confirmed by sequence alignment with $\geq 90\%$ coverage and
$\geq 90\%$ nucleotide identity.

**Genome alignment and phylogenetic analysis of LOS biosynthesis locus**

The annotated data of each LOS region was compared with blastn and visualized in clinker and
clustermap.js using the GenBank (GBK) file of the specified LOS region of each *C. jejuni*
genome (44). A phylogenetic tree was developed to understand the genomic variation of the LOS
region in GBS-associated and enteritis-associated *C. jejuni* strains. The LOS regions of *C. jejuni*
isolate were aligned on MAFFT online server version 7 using default parameters, and UPGMA
clustering was performed on the sequences of the LOS locus (45). The phylogenetic tree was
visualized using iTOL v6.8.1 (46).

**Variation of nucleotide in *Campylobacter jejuni* LOS biosynthesis locus**

Nucleotide variations in the LOS biosynthesis locus region were detected through SNIPPY
v4.6.0 (<https://github.com/tseemann/snippy>). Snippy uses the BWA-MEM algorithm to perform
local alignment, and SAMtool for parsing and manipulating alignment in SAM/BAM format.
Freebayes was used for variant calling (47). An unpaired t-test was performed using GraphPad
Prism v9 to compare the number of unique single nucleotide polymorphisms (SNPs), multiple
nucleotide polymorphisms (MNPs), complex polymorphisms (combinations of SNPs/MNPs),
and total nucleotide variations within the LOS region between GBS-associated and enteritis-
associated strains.

**Identification of phase variable genes within the LOS locus of *C. jejuni***

PhasomeIt, a novel program which identifies phase variable genes in *C. jejuni* genome through
association with simple sequence repeats was used to detect Phase Variable (PV) genes in the
LOS locus. To identify all putatively phase-variable loci in *C. jejuni* LOS regions, a cut-off value
of '7 6 0 5 5' and filter cutoff of 'W9' was used (48). Fisher's exact test was performed for 'G/C'
tracts containing PV genes of strains isolated from GBS and enteritis patients, depending on
whether their PV genes are ON or OFF.

***Acknowledgements***

This research activity was funded by the Fogarty International Center (FIC), National Institute of
Neurological Disorders and Stroke (NINDS) of the National Institutes of Health (NIH), USA
under Award Number K43TW011447. icddr,b gratefully acknowledges the commitment of the
Government of The People's Republic of Bangladesh and Global Affairs Canada for their
unrestricted support. We acknowledge all study personnel who contributed to patient enrollment
and specimen collection; data management and laboratory support. We are indebted to the
neurologists who referred their patients to this genomic study. We are thankful to the patients
who participated in the study and provided their valuable data.

***Author Contributions***

ZI, MAJN and SH conceptualized and designed the study. The research methodology and
execution plan were conceived by MAJN, SH and AA. MAJN, AA and MGM contributed to
data acquisition and genomic analysis. ZI contributed to the acquisition of funding. MAJN, SNF,
RB, MA, and AA did materials acquisition, laboratory experiments and data acquisition. MAJN
drafted the manuscript, which was critically reviewed by SH, AA, MGM, SNF, RB, MA, YK, IJ,
and ZI for intellectual content. All authors read and approved the final manuscript before
submission.

***Materials and Correspondence***

The correspondence and material requests would be addressed to ZI.

***Conflicts of interest***

The authors declare that there are no conflicts of interest.

***Ethical statement***

Each participant provided written informed consent, and all studies were carried out following
relevant guidelines and regulations. The study protocol was reviewed and approved by the
Institutional Review Board (IRB) and the Ethical Review Committee (ERC) of icddr,b, Dhaka,
Bangladesh under protocol number PR-19048.

**Data availability**

The authors confirm that all supporting data, and protocols have been provided within the article
or through supplementary data files. The raw sequencing data for this project and primary
assemblies are made publicly available under projects PRJNA378546, PRJNA717137, and
PRJNA607058. The accession numbers are provided in supplementary file **Table S1** which is
available in the online version of this article. Other metadata about study subjects and genome
sequences are provided in **Table 1**.

References

- 1. Malik A, Brudvig JM, Gadsden BJ, Ethridge AD, Mansfield LS. *Campylobacter jejuni*
induces autoimmune peripheral neuropathy via Sialoadhesin and Interleukin-4 axes. *Gut*
*Microbes*. 2022;14(1).
- 2. Islam Z, Gilbert M, Mohammad QD, Klaij K, Li J, van Rijs W, et al. Guillain-Barré
Syndrome-Related *Campylobacter jejuni* in Bangladesh: Ganglioside Mimicry and Cross-
Reactive Antibodies. Bereswill S, editor. *PLoS One* [Internet]. 2012 Aug 27;7(8):e43976.
Available from: <https://dx.plos.org/10.1371/journal.pone.0043976>
- 3. Islam Z, Jacobs BC, Van Belkum A, Mohammad QD, Islam MB, Herbrink P, et al.
Axonal variant of Guillain-Barré syndrome associated with *Campylobacter* infection in
Bangladesh. *Neurology*. 2010;74(7):581–7.
- 4. Allos BM. Association between *Campylobacter* infection and Guillain-Barre syndrome. *J*
*Infect Dis*. 1997;176(6 SUPPL.).
- 5. Mccarthy N, Andersson Y, Jormanainen V, Gustavsson O, Giesecke J. The risk of
Guillain-Barre syndrome following infection with *Campylobacter jejuni*. *Epidemiol*
*Infect*. 1999;122(1):15–7.
- 6. Godschalk PCR, Gilbert M, Jacobs BC, Kramers T, Tio-Gillen AP, Ang CW, et al. Co-
infection with two different *Campylobacter jejuni* strains in a patient with the Guillain-
Barré syndrome. *Microbes Infect* [Internet]. 2006 Jan;8(1):248–53. Available from:
<https://linkinghub.elsevier.com/retrieve/pii/S1286457905002601>
- 7. Godschalk PCR, Kuijf ML, Li J, St. Michael F, Ang CW, Jacobs BC, et al. Structural
characterization of *Campylobacter jejuni* lipooligosaccharide outer cores associated with
Guillain-Barré and Miller Fisher syndromes. *Infect Immun*. 2007;75(3):1245–54.
- 8. Islam Z, van Belkum A, Wagenaar JA, Cody AJ, de Boer AG, Tabor H, et al.
Comparative genotyping of *Campylobacter jejuni* strains from patients with Guillain-
Barré syndrome. *PLoS One*. 2009;4(9).
- 9. Heikema AP, Strepis N, Horst-Kreft D, Huynh S, Zomer A, Kelly DJ, et al. Biomolecule
sulphation and novel methylations related to Guillain-Barré syndrome-associated
*Campylobacter jejuni* serotype HS: 19. *Microb Genomics*. 2021;7(11):660.
- 10. Godschalk PCR, Heikema AP, Gilbert M, Komagamine T, Ang CW, Glerum J, et al. The
crucial role of *Campylobacter jejuni* genes in anti-ganglioside antibody induction in
Guillain-Barre syndrome. *J Clin Invest*. 2004;114(11):1659–65.
- 11. Djeghout B, Bloomfield SJ, Rudder S, Elumogo N, Mather AE, Wain J, et al.
Comparative genomics of *Campylobacter jejuni* from clinical campylobacteriosis stool
specimens. *Gut Pathog* [Internet]. 2022 Dec 7;14(1):45. Available from:
<https://gutpathogens.biomedcentral.com/articles/10.1186/s13099-022-00520-1>
- 12. Bowes T, Wagner ER, Boffey J, Nicholl D, Cochrane L, Benboubetra M, et al. Tolerance
to Self Gangliosides Is the Major Factor Restricting the Antibody Response to
Lipopolysaccharide Core Oligosaccharides in *Campylobacter jejuni* Strains Associated

- with Guillain-Barré Syndrome. *Infect Immun* [Internet]. 2002 Sep;70(9):5008–18.
Available from: <https://journals.asm.org/doi/10.1128/IAI.70.9.5008-5018.2002>
- 13. Yuki N. Molecular mimicry between gangliosides and lipopolysaccharides of
*Campylobacter jejuni* isolated from patients with Guillain-Barre syndrome and Miller
Fisher syndrome. *J Infect Dis*. 1997;176(6 SUPPL.):S150–3.
- 14. Yu RK, Usuki S, Ariga T. Ganglioside Molecular Mimicry and Its Pathological Roles in
Guillain-Barré Syndrome and Related Diseases. *Infect Immun* [Internet]. 2006
Dec;74(12):6517–27. Available from: <https://journals.asm.org/doi/10.1128/IAI.00967-06>
- 15. Hughes RAC, Cornblath DR. Guillain-barre syndrome. *Lancet*. 2005;366(9497):1653–66.
- 16. Yuki N. *Campylobacter* sialyltransferase gene polymorphism directs clinical features of
Guillain–Barré syndrome. *J Neurochem* [Internet]. 2007 Nov 6;103(s1):150–8. Available
from: <https://onlinelibrary.wiley.com/doi/10.1111/j.1471-4159.2007.04707.x>
- 17. Laman JD, Huizinga R, Boons G-J, Jacobs BC. Guillain-Barré syndrome: expanding the
concept of molecular mimicry. *Trends Immunol* [Internet]. 2022 Apr;43(4):296–308.
Available from: <https://linkinghub.elsevier.com/retrieve/pii/S147149062200028X>
- 18. Parker CT, Horn ST, Gilbert M, Miller WG, Woodward DL, Mandrell RE. Comparison of
*Campylobacter jejuni* Lipooligosaccharide Biosynthesis Loci from a Variety of Sources. *J*
*Clin Microbiol* [Internet]. 2005 Jun;43(6):2771–81. Available from:
<https://journals.asm.org/doi/10.1128/JCM.43.6.2771-2781.2005>
- 19. Parker CT, Gilbert M, Yuki N, Endtz HP, Mandrell RE. Characterization of
lipooligosaccharide-biosynthetic loci of *Campylobacter jejuni* reveals new
lipooligosaccharide classes: Evidence of mosaic organizations. *J Bacteriol*.
2008;190(16):5681–9.
- 20. Stephenson HN, John CM, Naz N, Gundogdu O, Dorrell N, Wren BW, et al.
*Campylobacter jejuni* Lipooligosaccharide Sialylation, Phosphorylation, and Amide/Ester
Linkage Modifications Fine-tune Human Toll-like Receptor 4 Activation. *J Biol Chem*
[Internet]. 2013 Jul;288(27):19661–72. Available from:
<https://linkinghub.elsevier.com/retrieve/pii/S0021925820456832>
- 21. Linton D, Karlyshev A V., Hitchen PG, Morris HR, Dell A, Gregson NA, et al. Multiple
N-acetyl neuraminic acid synthetase (*neuB*) genes in *Campylobacter jejuni*: Identification
and characterization of the gene involved in sialylation of lipo-oligosaccharide. *Mol*
*Microbiol*. 2000;35(5):1120–34.
- 22. Parkhill J, Wren BW, Mungall K, Ketley JM, Churcher C, Basham D, et al. The genome
sequence of the food-borne pathogen *Campylobacter jejuni* reveals hypervariable
sequences. *Nature* [Internet]. 2000;403(6770):665–8. Available from:
[file://localhost/Users/cliftonfranklund/Documents/Journal Articles/Parkhill\(2000\).pdf](file://localhost/Users/cliftonfranklund/Documents/Journal%20Articles/Parkhill(2000).pdf)
- 23. Gilbert M, Karwaski MF, Bernatchez S, Young NM, Taboada E, Michniewicz J, et al. The
genetic bases for the variation in the lipo-oligosaccharide of the mucosal pathogen,
*Campylobacter jejuni*. Biosynthesis of sialylated ganglioside mimics in the core
oligosaccharide. *J Biol Chem*. 2002;277(1):327–37.

- 24. Semchenko EA, Day CJ, Moutin M, Wilson JC, Tiralongo J, Korolik V. Structural
heterogeneity of terminal glycans in *Campylobacter jejuni* Lipooligosaccharides. *PLoS*
*One*. 2012;7(7).
- 25. Parker CT, Gilbert M, Yuki N, Endtz HP, Mandrell RE. Characterization of
lipooligosaccharide-biosynthetic loci of *Campylobacter jejuni* reveals new
lipooligosaccharide classes: Evidence of mosaic organizations. *J Bacteriol*.
2008;190(16):5681–9.
- 26. Heikema AP, Islam Z, Horst-Kreft D, Huizinga R, Jacobs BC, Wagenaar JA, et al.
*Campylobacter jejuni* capsular genotypes are related to Guillain-Barré syndrome. *Clin*
*Microbiol Infect*. 2015;21(9):852.e1-852.e9.
- 27. Koga M, Gilbert M, Takahashi M, Li J, Koike S, Hirata K, et al. Comprehensive analysis
of bacterial risk factors for the development of Guillain-Barré syndrome after
*Campylobacter jejuni* enteritis. *J Infect Dis*. 2006;193(4):547–55.
- 28. Heikema AP, Strepis N, Horst-Kreft D, Huynh S, Zomer A, Kelly DJ, et al. Biomolecule
sulphation and novel methylations related to guillain-barré syndrome-associated
*campylobacter jejuni* serotype hs:19. *Microb Genomics*. 2021;7(11).
- 29. Quino W, Caro-Castro J, Mestanza O, Hurtado V, Zamudio ML, Cruz-Gonzales G, et al.
Emergence and Molecular Epidemiology of *Campylobacter jejuni* ST-2993 Associated
with a Large Outbreak of Guillain-Barré Syndrome in Peru. Denes TG, editor. *Microbiol*
*Spectr* [Internet]. 2022 Oct 26;10(5). Available from:
<https://journals.asm.org/doi/10.1128/spectrum.01187-22>
- 30. Gilbert M, Karwaski MF, Bernatchez S, Young NM, Taboada E, Michniewicz J, et al. The
genetic bases for the variation in the lipo-oligosaccharide of the mucosal pathogen,
*Campylobacter jejuni*. Biosynthesis of sialylated ganglioside mimics in the core
oligosaccharide. *J Biol Chem*. 2002;277(1):327–37.
- 31. Guerry P, Szymanski CM, Prendergast MM, Hickey TE, Ewing CP, Pattarini DL, et al.
Phase variation of *Campylobacter jejuni* 81-176 lipooligosaccharide affects ganglioside
mimicry and invasiveness in vitro. *Infect Immun*. 2002;70(2):787–93.
- 32. Islam Z, Nabila FH, Asad A, Begum R, Jahan I, Hayat S, et al. Draft Genome Sequences
of Three Strains of *Campylobacter jejuni* Isolated from Patients with Guillain-Barré
Syndrome in Bangladesh. Rasko D, editor. *Microbiol Resour Announc* [Internet]. 2021
Apr 29;10(17). Available from: <https://journals.asm.org/doi/10.1128/MRA.00005-21>
- 33. Hayat S, Nabila FH, Asad A, Begum R, Jahan I, Endtz HP, et al. Draft Genome
Sequences of Four Strains of *Campylobacter jejuni* Isolated from Patients with Axonal
Variant of Guillain-Barré Syndrome in Bangladesh. Putonti C, editor. *Microbiol Resour*
*Announc* [Internet]. 2022 Feb 17;11(2). Available from:
<https://journals.asm.org/doi/10.1128/mra.01146-21>
- 34. Heng L, Richard D. Fast and accurate short read alignment with Burrows-Wheeler
Transform. *Bioinformatics*. 2009;25(14):1754–60.
- 35. Li H, Handsaker B, Wysoker A, Fennell T, Ruan J, Homer N, et al. The Sequence

- Alignment/Map format and SAMtools. *Bioinformatics* [Internet]. 2009 Aug
15;25(16):2078–9. Available from:
<https://academic.oup.com/bioinformatics/article/25/16/2078/204688>
- 36. Gurevich A, Saveliev V, Vyahhi N, Tesler G. QUAST: Quality assessment tool for
genome assemblies. *Bioinformatics*. 2013;29(8):1072–5.
- 37. Wood DE, Salzberg SL. Kraken: ultrafast metagenomic sequence classification using
exact alignments. *Genome Biol* [Internet]. 2014 Mar 3;15(3):R46. Available from:
<https://genomebiology.biomedcentral.com/articles/10.1186/gb-2014-15-3-r46>
- 38. Tatusova T, DiCuccio M, Badretdin A, Chetvernin V, Nawrocki EP, Zaslavsky L, et al.
NCBI prokaryotic genome annotation pipeline. *Nucleic Acids Res* [Internet]. 2016 Aug
19;44(14):6614–24. Available from: [https://academic.oup.com/nar/article-](https://academic.oup.com/nar/article-lookup/doi/10.1093/nar/gkw569)
[lookup/doi/10.1093/nar/gkw569](https://academic.oup.com/nar/article-lookup/doi/10.1093/nar/gkw569)
- 39. Aziz RK, Bartels D, Best A, DeJongh M, Disz T, Edwards RA, et al. The RAST Server:
Rapid annotations using subsystems technology. *BMC Genomics*. 2008;9.
- 40. Jolley KA, Bray JE, Maiden MCJ. Open-access bacterial population genomics: BIGSdb
software, the PubMLST.org website and their applications. *Wellcome Open Res*
[Internet]. 2018 Sep 24;3:124. Available from:
<https://wellcomeopenresearch.org/articles/3-124/v1>
- 41. Li H. New strategies to improve minimap2 alignment accuracy. *Bioinformatics*.
2021;37(23):4572–4.
- 42. Danecek P, JK B, J L, J M, V O, MO P, et al. Twelve years of SAMtools and BCFtools.
*Gigascience*. 2021;10(2).
- 43. Hameed A, Woodacre A, Machado LR, Marsden GL. An Updated Classification System
and Review of the Lipooligosaccharide Biosynthesis Gene Locus in *Campylobacter jejuni*
[Internet]. Vol. 11, *Frontiers in Microbiology*. 2020. p. 677. Available from:
<https://www.frontiersin.org/article/10.3389/fmicb.2020.00677>
- 44. Gilchrist CLM, Chooi YH. Clinker & clustermap.js: Automatic generation of gene cluster
comparison figures. *Bioinformatics*. 2021;37(16):2473–5.
- 45. Katoh K, Standley DM. MAFFT Multiple Sequence Alignment Software Version 7:
Improvements in Performance and Usability. *Mol Biol Evol* [Internet]. 2013 Apr
1;30(4):772–80. Available from: [https://academic.oup.com/mbe/article-](https://academic.oup.com/mbe/article-lookup/doi/10.1093/molbev/mst010)
[lookup/doi/10.1093/molbev/mst010](https://academic.oup.com/mbe/article-lookup/doi/10.1093/molbev/mst010)
- 46. Letunic I, Bork P. Interactive Tree Of Life (iTOL) v5: an online tool for phylogenetic tree
display and annotation. *Nucleic Acids Res* [Internet]. 2021 Jul 2;49(W1):W293–6.
Available from: <https://academic.oup.com/nar/article/49/W1/W293/6246398>
- 47. Garrison E, Marth G. Haplotype-based variant detection from short-read sequencing. 2012
Jul 17; Available from: <http://arxiv.org/abs/1207.3907>
- 48. Aidley J, Wanford JJ, Green LR, Sheppard SK, Bayliss CD. Phasomeit: An ‘omics’
approach to cataloguing the potential breadth of phase variation in the genus

campylobacter. *Microb Genomics*. 2018;4(11).

**Table 1.** *C. jejuni* strains isolated from patients with GBS including isolation year, GBS
subtypes, LOS class, presence of *cst-II/cst-III* genes, and presence of Thr/Asn at 51st positions of
*cst-II* gene.

C. jejuni strains	Year of isolation	Disease	GBS subtypes	LOS class	cst-II/cst-III	Thr/Asn at 51 st position
BD10	2006	GBS	AMAN	B	cst-II	Asn
BD22	2006	GBS	AMAN	B	cst-II	Asn
BD27	2006	GBS	AMAN	B	cst-II	Asn
BD34	2006	GBS	AMAN	B	cst-II	Asn
BD-39	2006	GBS	AMAN	A	cst-II	Thr
BD-67	2007	GBS	AMAN	B	cst-II	Asn
BD74	2007	GBS	AIDP	B	cst-II	Asn
BD-75	2007	GBS	N/D	A	cst-II	Thr
BD94	2007	GBS	AMAN	W	Absence	N/A
ZH1	2010	GBS	AMAN	C	cst-III	N/A
ZH2	2010	GBS	AMAN	B	cst-II	Thr
ZH6	2010	GBS	AMAN	A	cst-II	Thr
ZH7	2010	GBS	N/D	A	cst-II	Thr
ZH8	2011	GBS	AMAN	B	cst-II	Asn
ZH9	2011	GBS	AIDP	C	cst-III	N/A
ZH10	2011	GBS	AMAN	C	cst-III	N/A
ZH11	2011	GBS	AMAN	C	cst-III	N/A
ZH12	2011	GBS	AMAN	B	cst-II	Thr
ZH14	2011	GBS	AMAN	B	cst-II	Thr
ZH15	2012	GBS	Equivocal	B	cst-II	Asn
ZH16	2012	GBS	Inexcitable	B	cst-II	Asn
ZH18	2012	GBS	AMSAN	A	cst-II	Thr
ZH19	2012	GBS	AMAN	A	cst-II	Thr

ZH20	2012	GBS	AMAN	A	cst-II	Asn
Z191005RS	2019	GBS	AMAN	C	cst-III	N/A
Z191005SS	2019	GBS	AMAN	C	cst-III	N/A
Z201020SS	2020	GBS	AMAN	A	cst-II	Asn
Z201020RS	2020	GBS	AMAN	A	cst-II	Asn
Z211080RS	2021	GBS	AMAN	UD	Absence	N/A
Z211080SS	2021	GBS	AMAN	UD	Absence	N/A
Z211097RS	2021	GBS	AMAN	M	cst-II	Asn
Z211097SS	2021	GBS	AMAN	M	cst-II	Asn
Z221123RS	2022	GBS	AMAN	A	cst-II	Asn
Z221123SS	2022	GBS	AMAN	A	cst-II	Asn
Z221130RS	2022	GBS	AMAN	F	Absence	N/A
Z221146RS	2022	GBS	AMAN	B	cst-III	Asn
Z221146SS	2022	GBS	AMAN	B	cst-III	Asn
Z227001SS	2022	GBS	N/D	A	cst-III	Asn

*GBS, Guillain-Barré syndrome; AMAN, Acute motor axonal neuropathy; AIDP, acute*
*inflammatory demyelinating polyradiculoneuropathy; UD, Undetermined; Thr, Threonine; Asn,*
*asparagine; N/D, not done; N/A, not applicable*

**Fig. 1. Sequence Type (ST) and Clonal Complexes (CC) of intended 64 *C. jejuni*.** (A)
Neighbor-joining tree based on concatenated nucleotide sequence loci of *Campylobacter jejuni*
allele from PubMLST database. (B) Venn diagram of Sequence Type (ST) diversity among GBS
and enteritis-associated *C. jejuni* strains. (C) Venn diagram of Clonal complex (CC) diversity
among GBS and enteritis-associated *C. jejuni* strains.

**Fig. 2.** Phylogenetic tree of LOS region of *C. jejuni* strains isolated from GBS and enteritis
patients along with reference strain NCTC11168.

**Fig. 3.** Comparative genome alignment of LOS biosynthesis locus region between GBS (Fig.
3A) and enteritis-associated *C. jejuni* strains (Fig. 3B). LOS region of *C. jejuni* NCTC11168 was
used as the reference genome. Colored arrows indicate the coding sequences and various colour-
reflected homologues groups identified by the clinker. Grey/white arrows represent the
hypothetical proteins. The black and grey bars represent the percentage of amino acid identity as
indicated in the figure.

**Fig. 4.** Distribution of nucleotide variations within the LOS region between GBS-associated and
enteritis-associated *C.jejuni*. (A) Distribution of total nucleotide variations, (B) Distribution of
SNPs, (C) Distribution of MNPs, (D) Distribution of complex (combination of SNP/MNP).

**Fig. 5.** The presence of phase variation within the LOS region of GBS-associated and enteritis-
associated strains.

The authors postulate that little is known about the significant diverse genomic variability of the LOS region of *C. jejuni* strains associated with GBS in low- and middle-income countries. Therefore, this study analyzes 38 GBS- and 24 enteritis-associated *C. jejuni* strains from Bangladesh for ST variation and LOS region diversity. Twelve enteritis-associated strains were taken from NCBI. They postulate that the over-representation of GBS-specific LOS classes with higher rates of nucleotide and phase variations might be the determining factor for GBS.

While the paper was generally well written, I'm not convinced that the conclusions of this study are fully supported by the data presented. I recommend revisiting the methodology surrounding the frequency of nucleotide and phase variation within the LOS locus, as it was not clear in the manuscript. I also suggest minor improvements to the phylogenetic trees to better summarize the different results from Table 1 (ST, CC, disease, LOS class, cst gene type and mutation) together in the same figure.

Introduction

1. The molecular-mimicry phenomenon that explains how certain *Campylobacter* can provoke a cross-reactive antibody response is repeated at many points in the Introduction. I would recommend reviewing this section to make it more concise.

Results

2. I find Figure 1a would be easier to analyze if the tree was displayed as a rectangle. I know that we usually see circularized trees in papers, but that is mostly due to the large number of sequences which would make it impossible to fit rectangularly in one page. But in your case, as there are only 63 sequences, I think it could fit well within figure dimensions and could provide extra space to include other metadata to the tree (ex. All information from Table 1).
3. Is the phylogenetic tree in Figure 1A based only on the 7 MLST loci, the cgMLST scheme or the wgMLST scheme in PubMLST? Many different schemas for *Campylobacter* exist and it was not clear in the Methods section which scheme was used, as it seems to be only roughly mentioned in Figure 1's description. No distance scale is provided either to interpret the branch lengths. Why not use Snippy on the whole genome instead?
4. I think Figure 1a and 1b could be better represented as a compiled bar graph with the STs and CCs included in the figure, which would make it

- easier to visualize the commonalities and differences between the two groups (GBS and enteritis).
5. Line 142: Why were the LOS classes not determined for some of the strains? Did they not correspond to the classes defined by Hameed et al. (2020)? An explanation here could be useful to better understand the results.
 6. Line 148: Did you mean to say “undetermined” instead of “undermined” here?
 7. Lines 162-168: It wasn't clear what reference genome was used with Snippy to identify the nucleotide variations within the LOS locus. What LOS class did the reference genome have? Could this have biased your results?
 8. Lines 168-175: I feel this section describes Figure 2, the phylogeny of the LOS locus of your 63 strains, which should be mentioned earlier in the manuscript, when Figure 2 is first mentioned. I don't see why the phylogenetic analysis and the nucleotide variation is described together here. I would recommend adding it to “Identification of frequency of *C. jejuni* LOS locus classes” and to modify the title, or to give it its own subsection.
 9. In Figure 2, do you know why LOS classes A and B cluster into different branches? In other words, classes A and B are not monophyletic, as is the case for the other LOS classes presented here. Is this expected? Do the LOS classes A and B have a lot of genetic similarity and overlap?
 10. Were the clusters 1, 2 and 3 in Figure 2 defined statistically by Snippy, or were they arbitrarily defined by the authors?
 11. Do the *cst* gene profiles (II vs III and Asn51 vs Thr51) correlate with the LOS classes or ST/CC? For example, are all *cst*-II (Asn51) within LOS class A? It could be useful to include the *cst* gene information as metadata in one of the phylogenetic trees (Figure 1 or 3) to improve visualization instead of only describing them in Table 1.
 12. Line 188: There doesn't seem to be many genes inside the LOS locus, but you claim 108 phase variable genes were found. Perhaps you meant to say “sites” instead of “genes”? Later on, you mention 13 different groups of PV genes. Were these 108 sites found within 13 different genes inside the LOS locus? Providing the name of these genes is suggested. This paragraph should be reworked to improve comprehension.
 13. Again, do the different polymeric tracts correlate with the different LOS classes and/or ST/CC observe between the GBS and enteritis groups? Are they dependent on the other variables?

14. I'm not convinced that there is more nucleotide variation inside the LOS locus in GBS-associated strains compared to enteritis strains. Could you explain in more detail how the differences were counted? Does the reference genome used for Snippy include all possible genes that could be included in the LOS locus? Because since Snippy is a reference-based alignment tool, if an LOS locus gene is absent from the reference genome, no SNPs will be found for this gene. Is it possible that the LOS locus in the reference genome shared more resemblance to the primary LOS classes detected in GBS (class A and B), and that's why you observe more nucleotide variation? Also, some LOS classes illustrated in Figure 3b (ex. class G) seem to have less gene content than the other LOS classes and is more represented in enteritis than in GBS. With that being said, perhaps the frequency of nucleotide variation in the LOS locus should be normalized on the full length of the LOS locus. This might explain why more variation is observed in GBS-associated strains due to the longer LOS locus observed on average. Same goes for the frequency in phase variation in Figure 5.
15. I would recommend including the outputs of Snippy and Phasomelt or creating a summary table describing their results as supplementary tables.

Discussion

16. Line 208: Parentheses not needed.
17. Lines 210-212: "our research indicates that LOS class, sialyltransferase variation, and phase variation within the LOS region might be the significant contributing factors for inducing GBS" ... However, it was not demonstrated if these 3 factors are co-dependent. Do only certain LOS classes (ex. A and B) have more significant sialyltransferase and phase variation? How do these 3 factors differ from one another?
18. Lines 260-262: I would recommend placing this text in the Introduction instead.
19. I would suggest mentioning that other host-related factors can influence the development of GBS (ex. state of immune system, age, etc.). Based on previous research, genotypes alone do not seem to be able to predict with 100% accuracy the development of GBS.

Methods

20. Line 313: I don't understand what has been filtered from the assemblies here... Did contigs under 5x coverage stay within the assembly? 5x coverage is not very high to generate a good quality genome assembly. What were the range of coverages in your genomes? What amount of N

bases did you have per genome on average? How many contigs were in your final assemblies? This information could be added to support the quality of the sequences generated.

21. Line 336: The Hameed reference is dated in 2020, not 2022.

22. Line 356: I would be careful with the use of the term “novel” here to describe Phasomelt, as the paper was published 6 years ago...

Spectrum00062-25: Genomic Variability of Lipooligosaccharide Biosynthesis Locus and Sequence Type among *Campylobacter jejuni* Isolated from Patients with Guillain-Barre' Syndrome

Reviewer #1 (Editor):

Comment: The study is relevant, however some gaps related to genome quality assessment are present and could affect analysis and conclusions.

Response: We sincerely appreciate your positive feedback on our submitted manuscript. All queries have been addressed point by point in the revised version.

Query-1: Authors should detail the quality of selected genomes (number of contigs, N50, length of genomes), that should reflect the robustness of analysis.

Response: Thank you for your comment. We have added detailed information on the quality of the genomes, including the number of contigs, N50, and genome lengths, in **Supplementary Table S1**.

Query-2: Authors used genomes with low coverage (5X). This is very low, considering the importance of the study. I recommend to re-analyse genomes, using only those with higher coverage 10X and more then 15X and more, to provide a more robust analysis.

Response: We thank the reviewer for the helpful comment and apologize for the confusion. The reported 5X coverage was a typo error; in fact, all genomes have an average coverage of ~200X, as now correctly noted in **Supplementary Table S1**. Short contigs (<200 bp) were removed during quality filtering, but contig coverage was not used as a filter. Given the high coverage, we believe the dataset is robust, and this approach does not compromise quality. The manuscript and supplementary materials have been corrected accordingly
(Lines: 308-309; Supplementary Table S1).

Reviewer #2 (Comments for the Author):

Comment: The study is relevant, however some gaps related to genome quality assessment are present and could affect analysis and conclusions.

Response: We sincerely appreciate your positive feedback on our submitted manuscript. All queries have been addressed point by point in the revised version.

Query-1: Authors should detail the quality of selected genomes (number of contigs, N50, length of genomes), that should reflect the robustness of analysis.

Response: Thank you for your comment. We have added detailed information on the quality of the genomes, including the number of contigs, N50, and genome lengths, in **Supplementary Table S1**.

Query-2: Authors used genomes with low coverage (5X). This is very low, considering the importance of the study. I recommend to re-analyse genomes, using only those with higher coverage 10X and more then 15X and more, to provide a more robust analysis.

Response: We thank the reviewer for the helpful comment and apologize for the confusion. The reported 5X coverage was a typo error; in fact, all genomes have an average coverage of ~200X, as now correctly noted in **Supplementary Table S1**. Short contigs (<200 bp) were removed during quality filtering, but contig coverage was not used as a filter. Given the high coverage, we believe the dataset is robust, and this approach does not compromise quality. The manuscript and supplementary materials have been corrected accordingly (**Lines: 308-309; Supplementary Table S1**).

Introduction

Comment-1: The molecular-mimicry phenomenon that explains how certain Campylobacter can provoke a cross-reactive antibody response is repeated at many points in the Introduction. I would recommend reviewing this section to make it more concise.

Response: Thanks for the suggestion. The Introduction section has been revised and made more concise. (**Pages: 4-5; Lines: 62-67, 81-88, 91-95**)

Results

Comment-2: I find Figure 1a would be easier to analyze if the tree was displayed as a rectangle. I know that we usually see circularized trees in papers, but that is mostly due to the large number of sequences which would make it impossible to fit rectangularly in one page. But in your case, as there are only 63 sequences, I think it could fit well within figure dimensions and could provide extra space to include other metadata to the tree (ex. All information from Table 1).

Response: **Fig-1A** has been revised and presented in a rectangular format. As per recommendation, we have integrated the metadata, including sequence type (ST), clonal complex (CC), Disease status, Isolation year, LOS class, *cst* gene type, and *cst-II* mutation into the updated **Fig-1A**.

Comment-3: Is the phylogenetic tree in Figure 1A based only on the 7 MLST loci, the cgmlst scheme or the wgMLST scheme in PubMLST? Many different schemas for Campylobacter exist and it was not clear in the Methods section which scheme was used, as it seems to be only roughly mentioned in Figure 1's description.

Response: The phylogenetic tree is based on 7 MLST loci. In addition, we incorporated the lines “The multi-locus sequence type (MLST) and clonal complex (CC) were determined through the PubMLST database, utilizing the sequences of seven housekeeping genes: *aspA*, *glnA*, *gltA*, *glyA*, *pgm*, *tkl*, and *uncA*” in the methodology section. (Page: 16; Lines: 313-316)

No distance scale is provided either to interpret the branch lengths.

Response: Tree distance scale has been added to the revised phylogenetic tree in **Fig-1A** and **Fig-2**.

Why not use Snippy on the whole genome instead?

Response: The study focused on the lipooligosaccharide (LOS) biosynthesis locus of *C. jejuni* as it is directly related to the molecular mimicry pathogenesis mechanism of GBS. Moreover, we could not find any phylogenetic clustering between GBS and enteritis-associated *C. jejuni* while considering the whole genome, but the LOS region showed the group-wise presentation (Fig-2).

Comment-4: I think Figure 1a and 1b could be better represented as a compiled bar graph with the STs and CCs included in the figure, which would make it easier to visualize the commonalities and differences between the two groups (GBS and enteritis).

Response: Thanks for the suggestions. The figures have been revised to include bar graphs grouped by sequence types (STs) and clonal complexes (CCs) in revised version (**Fig-1B**, **Fig-1C**).

Comment-5: - Line 142: Why were the LOS classes not determined for some of the strains? Did they not correspond to the classes defined by Hameed et al. (2020)? An explanation here could be useful to better understand the results.

Response: We followed the method described by Hameed et al. (2020) to determine the LOS classes of all *C. jejuni* strains. Strains that could not be classified into the defined LOS classes by Hameed et al. 2020 were labelled as "undetermined." An explanation has been added in the discussion section. (Page: 12; Lines: 228-231)

Comment-6: -Line 148: Did you mean to say “undetermined” instead of “undermined” here?

Response: This is a typo error of “undetermined”. The term 'undermined' has been corrected to 'undetermined' accordingly. (Page: 8; Lines: 141)

Comment-7: -Lines 162-168: It wasn't clear what reference genome was used with Snippy to identify the nucleotide variations within the LOS locus. What LOS class did the reference genome have? Could this have biased your results?

Response: We used the *C. jejuni* NCTC11168, which confers LOS class C, as the reference genome to identify nucleotide variations within the LOS locus. This information has also been included in the Methods section (**Page: 17; Line: 346**).

LOS classes mainly differ due to the presence/absence of a genic/non-genic region, not based on SNP/MNP. Moreover, we performed mapping-based nucleotide variation analysis, therefore, nucleotide variations were only reported when the comparing LOS region also possessed the same gene present in the reference genome.

Comment-8: -Lines 168-175: I feel this section describes Figure 2, the phylogeny of the LOS locus of your 63 strains, which should be mentioned earlier in the manuscript, when Figure 2 is first mentioned. I don't see why the phylogenetic analysis and the nucleotide variation is described together here. I would recommend adding it to "Identification of frequency of *C. jejuni* LOS locus classes" and to modify the title, or to give it its own subsection.

Response: Thanks for the suggestion. We have incorporated the suggested section into the 'Identification of LOS classes and phylogenetic analysis of the *C. jejuni* LOS region' and have modified the subsection title as per your recommendation (**Page: 8; Lines: 141-148**)

Comment-9: In Figure 2, do you know why LOS classes A and B cluster into different branches? In other words, classes A and B are not monophyletic, as is the case for the other LOS classes presented here. Is this expected? Do the LOS classes A and B have a lot of genetic similarity and overlap?

Response: Yes, it was expected. LOS classes A and B in *C. jejuni* share the highest genetic similarity among all the LOS classes, as both possess the genes involved in sialic acid biosynthesis and the production of ganglioside-like structures, but differ in the specific glycosyltransferases they utilize, resulting in variations in the final LOS structure. Therefore, LOS A and LOS B can be close together in the phylogenetic tree.

Comment-10: Were the clusters 1, 2 and 3 in Figure 2 defined statistically by Snippy, or were they arbitrarily defined by the authors?

Response: The clusters mentioned in Fig. 2 did not indicate any clade or lineage. They were defined arbitrarily for a better understanding of the genetic similarity and closeness.

Comment-11: Do the *cst* gene profiles (II vs III and Asn51 vs Thr51) correlate with the LOS classes or ST/CC? For example, are all *cst*-II (Asn51) within LOS class A? It could be useful to include the *cst* gene information as metadata in one of the phylogenetic trees (Figure 1 or 3) to improve visualization instead of only describing them in Table 1.

Response: The Sialyltransferase (*cst*) gene profiles (*cst-II* vs. *cst-III* and Asn51 vs. Thr51) showed no correlation with LOS classes or sequence type (ST)/clonal complex (CC). The sialyltransferase gene (*cst*) information as metadata has been included in the phylogenetic tree of **Fig-1A**.

Comment-12: -Line 188: There doesn't seem to be many genes inside the LOS locus, but you claim 108 phase variable genes were found. Perhaps you meant to say "sites" instead of "genes"? Later on, you mention 13 different groups of PV genes. Were these 108 sites found within 13 different genes inside the LOS locus? Providing the name of these genes is suggested. This paragraph should be reworked to improve comprehension.

Response: A total of 108 phase-variable (PV) sites have been found in the 13 different groups of PV genes. The line has been edited, and the name list of phase-variable genes has been provided in the Supplementary Table S3 (**Page: 10; Line: 181-183, & Table: S3, S4**).

Comment-13: Again, do the different polymeric tracts correlate with the different LOS classes and/or ST/CC observe between the GBS and enteritis groups? Are they dependent on the other variables?

Response: Phase variation is independent, considering LOS classes and/or ST/CC. Genomic phase-variation enables bacteria to adapt to rapidly changing environments by reversibly altering genes through mechanisms such as slipped-strand mispairing, recombination, and insertion/excision events without relying on permanent mutations.

Comment-14. I'm not convinced that there is more nucleotide variation inside the LOS locus in GBS-associated strains compared to enteritis strains. Could you explain in more detail how the differences were counted? Does the reference genome used for Snippy include all possible genes that could be included in the LOS locus? Because since Snippy is a reference-based alignment tool, if an LOS locus gene is absent from the reference genome, no SNPs will be found for this gene. Is it possible that the LOS locus in the reference genome shared more resemblance to the primary LOS classes detected in GBS (class A and B), and that's why you observe more nucleotide variation? Also, some LOS classes illustrated in Figure 3b (ex. class G) seem to have less gene content than the other LOS classes and is more represented in enteritis than in GBS. With that being said, perhaps the frequency of nucleotide variation in the LOS locus should be normalized on the full length of the LOS locus. This might explain why more variation is observed in GBS-associated strains due to the longer LOS locus observed on average. Same goes for the frequency in phase variation in Figure 5.

Response:

- a) We used the LOS region of *C. jejuni* NCTC11168 as a reference for Snippy and gene content representation, which confers LOS class C. LOS classes mainly differ due to the presence/absence of a genic/non-genic region, not based on SNP.
- b) Moreover, we performed mapping-based SNP analysis, therefore, SNPs were only reported if comparing the LOS region also possessed the same gene present in the

reference genome, minimizing potential bias. We used reference genome that belongs to Class-C LOS which is more genetically closer to LOS class A and B. Therefore, there is a minimum chance of overrepresentation of SNV in GBS-related *C. jejuni*.

- c) For phase variation detection, PhasomeIt was used. PhasomeIt employs a custom algorithm that identifies phase-variable genes in *Campylobacter* species by searching for simple sequence repeats (SSRs), particularly those involving homopolymeric tracts such as poly-G and poly-C. One of its key features is that it does not rely on reference genomes; instead, it scans the genome for repeat motifs associated with phase variation, allowing it to detect phase variation even in novel or draft genomes without predefined annotations.

Comment-15: I would recommend including the outputs of Snippy and PhasomeIt or creating a summary table describing their results as supplementary tables.

Response: Thanks for the suggestion. We have included the summary table of outputs of Snippy (**Supplementary Table S2**) and PhasomeIt (**Supplementary Table S3**) in the supplementary section.

Discussion

Comment-16: Line 208: Parentheses not needed.

Response: The parentheses have been removed.

Comment-17: Lines 210-212: “our research indicates that LOS class, sialyltransferase variation, and phase variation within the LOS region might be the significant contributing factors for inducing GBS” ... However, it was not demonstrated if these 3 factors are co-dependent. Do only certain LOS classes (ex. A and B) have more significant sialyltransferase and phase variation? How do these 3 factors differ from one another?

Response: LOS class, sialyltransferase variation, and phase variation are not co-dependent, but they are functionally interconnected. Each component influences and is influenced by the others in shaping the final structure of the LOS and the pathogen’s ability to evade the host immune system.

How do these 3 factors differ from one another?

Response: LOS class defines the structural gene framework of the LOS locus; sialyltransferase variation influences the ability to sialylate LOS and mimic host gangliosides; and phase variation governs the on/off expression of genes, allowing rapid phenotypic switching. While these mechanisms are interrelated, they serve distinct roles, including structural organization, enzymatic function, and gene regulation, respectively.

Comment-18: -Lines 260-262: I would recommend placing this text in the Introduction instead.

Response: As per reviewer suggestion we incorporated the intended text “Gilbert *et al.* reported that two types of *cst-II* gene alleles cause the translated enzyme to express either threonine (Thr) or asparagine (Asn) at position 51st, resulting in monofunctional or bifunctional activity with one or two sialic acids” in the Introduction section. (Page: 5; Lines: 100-102)

Comment-19: I would suggest mentioning that other host-related factors can influence the development of GBS (ex. state of immune system, age, etc.). Based on previous research, genotypes alone do not seem to be able to predict with 100% accuracy the development of GBS.

Response: Thank you for the suggestion. We have added the influence of host-related factors in the Discussion. (Page: 14; Line: 273-277)

Methods

Comment-20: -Line 313: I don't understand what has been filtered from the assemblies here... Did contigs under 5x coverage stay within the assembly? 5x coverage is not very high to generate a good quality genome assembly. What were the range of coverages in your genomes? What amount of N bases did you have per genome on average? How many contigs were in your final assemblies? This information could be added to support the quality of the sequences generated.

Response: This is a typo error and has been corrected accordingly in the revised version of the manuscript. Short contigs that are <200bp have been removed/filtered (Page: 16; Lines: 309-310).

The range of genome coverage, the amount of N bases per genome average and the number of contigs in final assemblies have been included in the supplementary file. (Supplementary Table S1)

Comment-21: -Line 336: The Hameed reference is dated in 2020, not 2022.

Response: The reference has been corrected. (Page: 17; Line: 334)

Comment-22: -Line 356: I would be careful with the use of the term “novel” here to describe PhasomeIt, as the paper was published 6 years ago...

Response: We have removed the term “novel” and revised to “PhasomeIt tool was used for the identification of phase-variable (PV) genes within the LOS locus of *C. jejuni* genome by analyzing associations with simple sequence repeats”. (Page: 18; Lines: 355-356)

Re: Spectrum00062-25R1 (Genomic Variability of Lipooligosaccharide Biosynthesis Locus and Sequence Type among *Campylobacter jejuni* Isolated from Patients with Guillain-Barre´ Syndrome)

Dear Dr. Zahirul Islam:

Thank you for the privilege of reviewing your work. Below you will find my comments, instructions from the Spectrum editorial office, and the reviewer comments.

Revision Guidelines

Sincerely,
Sadjia Bekal
Editor
Microbiology Spectrum

Reviewer #1 (Comments for the Author):

Results:
Minor observations and recommendations on line 114 and 128

Discussion:

minor observations on lines 203 and 241

Materials and Methods:

minor observations on line 303, 309 and 310

Reviewer #2 (Comments for the Author):

The authors have made all suggested corrections in the manuscript and in the figures and tables to increase clarity and transparency. The quality of the genomes are now considered good quality now that more details have been provided. However, some components, like the statistical analysis of differences in genomic variations observed between GBS and enteritis cases, have not been explained or supported enough to justify its acceptance.

Reviewer comments :

1. Thank you for updating Figure 1 with more metadata and including the bar charts as suggested! It makes it immensely easier to analyze and compare results. The only caveat that I have is the overlap between the Disease labels/sequence identifiers and the end of the branches of the tree. Since they are touching, it makes it difficult to see when the sequences are shared on the same branch (when there is a vertical line for the branches for identical ST). I would suggest adding a small gap between the tree and the sequence identifiers, if possible.
2. In the Figure 1 caption, you mention the 53rd position of cst-II, but in the figure, it's written « cst-II 51st position ».
3. At the added Lines 347-348, the term « local alignment » is repeated twice, but not necessary. SNIPPY/Snippy should be written in the same way as well.
4. Thank you for your comments concerning Lines 162-168 and the reference genome alignment. So, if the reference genome is LOS class C, then only the genes present in LOS class C would be taken into consideration by Snippy for SNP analysis with all other LOS classes. I understand that some classes resemble LOS class C more than others. For example, perhaps LOS class B has 16 common genes with LOS class C, but LOS class W only has 11 genes in common since many other genes are included in its locus, but seem absent in the LOS class C from the reference according to your Figure 3. You also mention that: « The LOS region of GBS-associated strains showed a similar pattern of gene alignment, with the majority of them (84%) belonging to LOS classes A, B, and C. Whereas, the LOS regions of enteritis-associated strains exhibited a more varied gene pattern with higher presence of hypothetical proteins compared to GBS-associated strains which belong to seven different LOS classes » (Lines 156-160).
Could this effect or bias your results demonstrated in Figures 4 and 5 (due to less gene alignments compared to the reference for the Enteritis group)? This is why I thought to normalize the number of SNP differences and phase variable regions based on the length of the alignment, as I would expect to find less genetic variation if a smaller genomic region matches the reference genome used to detect such genetic variation (SNP, MNP and complex nucleotide variations; 16 genes vs 11 genes). In other words, if the y-axis in Figures 4 and 5 were « No. of *insert type of variation (SNP, phase variation, etc.)* per 1 kb of alignment », would there still be significant differences between GBS and Enteritis groups? This is just a thought/concern, but if you feel that your current analysis is sufficient, keep it that way, but I would suggest justifying why this bias does not exist.
5. I would recommend adding more details in the Figure 2 caption. For example, adding a bit of the methods used here (SNP-based, number of SNPs used, etc.).
6. Thank you for providing the list of phase variable genes in Supplementary Table S3! Is it possible to add the gene names in the table to match the gene names from Figure 3? It would be helpful for readers to connect both items.

**Genomic Variability of Lipooligosaccharide Biosynthesis Locus and Sequence Type among**
***Campylobacter jejuni* Isolated from Patients with Guillain-Barre´ Syndrome**

Md. Abu Jaher Nayeem,^a Shoma Hayat,^a Asaduzzaman Asad,^a Md. Golam Mostafa,^a Shah
Nayeem Faruque,^a Ruma Begum,^a Mosabbir Ahmed,^a Yearul Kabir,^b Israt Jahan,^{a,c} Zhahirul
Islam,^{a*}

[revised manuscript text omitted]

**References**

- 1. Malik A, Brudvig JM, Gadsden BJ, Ethridge AD, Mansfield LS. *Campylobacter jejuni*
induces autoimmune peripheral neuropathy via Sialoadhesin and Interleukin-4 axes. *Gut*
*Microbes*. 2022;14(1).
- 2. Islam Z, Gilbert M, Mohammad QD, Klaij K, Li J, van Rijs W, et al. Guillain-Barré
Syndrome-Related *Campylobacter jejuni* in Bangladesh: Ganglioside Mimicry and Cross-
Reactive Antibodies. Bereswill S, editor. *PLoS One* [Internet]. 2012 Aug 27;7(8):e43976.
Available from: <https://dx.plos.org/10.1371/journal.pone.0043976>
- 3. Tam CC, Rodrigues LC, Petersen I, Islam A, Hayward A, O'Brien SJ. Incidence of
Guillain-Barré syndrome among patients with *Campylobacter* infection: A general
practice research database study. *J Infect Dis* [Internet]. 2006 Jul;194(1):95–7. Available
from: <https://academic.oup.com/jid/article-lookup/doi/10.1086/504294>
- 4. Islam Z, Jacobs BC, Van Belkum A, Mohammad QD, Islam MB, Herbrink P, et al.
Axonal variant of Guillain-Barré syndrome associated with *Campylobacter* infection in
Bangladesh. *Neurology*. 2010;74(7):581–7.
- 5. Islam Z, van Belkum A, Wagenaar JA, Cody AJ, de Boer AG, Tabor H, et al.
Comparative genotyping of *Campylobacter jejuni* strains from patients with Guillain-
Barré syndrome. *PLoS One*. 2009;4(9).
- 6. Heikema AP, Strepis N, Horst-Kreft D, Huynh S, Zomer A, Kelly DJ, et al. Biomolecule
sulphation and novel methylations related to Guillain-Barré syndrome-associated
*Campylobacter jejuni* serotype HS: 19. *Microb Genomics*. 2021;7(11):660.
- 7. Godschalk PCR, Heikema AP, Gilbert M, Komagamine T, Ang CW, Glerum J, et al. The
crucial role of *Campylobacter jejuni* genes in anti-ganglioside antibody induction in
Guillain-Barre syndrome. *J Clin Invest*. 2004;114(11):1659–65.
- 8. Djeghout B, Bloomfield SJ, Rudder S, Elumogo N, Mather AE, Wain J, et al.
Comparative genomics of *Campylobacter jejuni* from clinical campylobacteriosis stool
specimens. *Gut Pathog* [Internet]. 2022 Dec 7;14(1):45. Available from:
<https://gutpathogens.biomedcentral.com/articles/10.1186/s13099-022-00520-1>
- 9. Yuki N. Molecular mimicry between gangliosides and lipopolysaccharides of
*Campylobacter jejuni* isolated from patients with Guillain-Barre syndrome and Miller
Fisher syndrome. *J Infect Dis*. 1997;176(6 SUPPL.):S150–3.
- 10. Yu RK, Usuki S, Ariga T. Ganglioside Molecular Mimicry and Its Pathological Roles in
Guillain-Barré Syndrome and Related Diseases. *Infect Immun* [Internet]. 2006
Dec;74(12):6517–27. Available from: <https://journals.asm.org/doi/10.1128/IAI.00967-06>
- 11. Hughes RAC, Cornblath DR. Guillain-barre syndrome. *Lancet*. 2005;366(9497):1653–66.
- 12. Yuki N. *Campylobacter* sialyltransferase gene polymorphism directs clinical features of
Guillain–Barré syndrome. *J Neurochem* [Internet]. 2007 Nov 6;103(s1):150–8. Available
from: <https://onlinelibrary.wiley.com/doi/10.1111/j.1471-4159.2007.04707.x>
- 13. Laman JD, Huizinga R, Boons G-J, Jacobs BC. Guillain-Barré syndrome: expanding the

- concept of molecular mimicry. *Trends Immunol* [Internet]. 2022 Apr;43(4):296–308.
Available from: <https://linkinghub.elsevier.com/retrieve/pii/S147149062200028X>
- 14. Parker CT, Horn ST, Gilbert M, Miller WG, Woodward DL, Mandrell RE. Comparison of
*Campylobacter jejuni* Lipooligosaccharide Biosynthesis Loci from a Variety of Sources. *J*
*Clin Microbiol* [Internet]. 2005 Jun;43(6):2771–81. Available from:
<https://journals.asm.org/doi/10.1128/JCM.43.6.2771-2781.2005>
- 15. Parker CT, Gilbert M, Yuki N, Endtz HP, Mandrell RE. Characterization of
lipooligosaccharide-biosynthetic loci of *Campylobacter jejuni* reveals new
lipooligosaccharide classes: Evidence of mosaic organizations. *J Bacteriol*.
2008;190(16):5681–9.
- 16. Stephenson HN, John CM, Naz N, Gundogdu O, Dorrell N, Wren BW, et al.
*Campylobacter jejuni* Lipooligosaccharide Sialylation, Phosphorylation, and Amide/Ester
Linkage Modifications Fine-tune Human Toll-like Receptor 4 Activation. *J Biol Chem*
[Internet]. 2013 Jul;288(27):19661–72. Available from:
<https://linkinghub.elsevier.com/retrieve/pii/S0021925820456832>
- 17. Linton D, Karlyshev A V., Hitchen PG, Morris HR, Dell A, Gregson NA, et al. Multiple
N-acetyl neuraminic acid synthetase (*neuB*) genes in *Campylobacter jejuni*: Identification
and characterization of the gene involved in sialylation of lipo-oligosaccharide. *Mol*
*Microbiol*. 2000;35(5):1120–34.
- 18. Parkhill J, Wren BW, Mungall K, Ketley JM, Churcher C, Basham D, et al. The genome
sequence of the food-borne pathogen *Campylobacter jejuni* reveals hypervariable
sequences. *Nature* [Internet]. 2000;403(6770):665–8. Available from:
[file://localhost/Users/cliftonfranklund/Documents/Journal Articles/Parkhill\(2000\).pdf](file://localhost/Users/cliftonfranklund/Documents/Journal%20Articles/Parkhill(2000).pdf)
- 19. Semchenko EA, Day CJ, Moutin M, Wilson JC, Tiralongo J, Korolik V. Structural
heterogeneity of terminal glycans in *Campylobacter jejuni* Lipooligosaccharides. *PLoS*
*One*. 2012;7(7).
- 20. Gilbert M, Karwaski MF, Bernatchez S, Young NM, Taboada E, Michniewicz J, et al. The
genetic bases for the variation in the lipo-oligosaccharide of the mucosal pathogen,
*Campylobacter jejuni*. Biosynthesis of sialylated ganglioside mimics in the core
oligosaccharide. *J Biol Chem*. 2002;277(1):327–37.
- 21. Parker CT, Gilbert M, Yuki N, Endtz HP, Mandrell RE. Characterization of
lipooligosaccharide-biosynthetic loci of *Campylobacter jejuni* reveals new
lipooligosaccharide classes: Evidence of mosaic organizations. *J Bacteriol*.
2008;190(16):5681–9.
- 22. Gilbert M, Karwaski MF, Bernatchez S, Young NM, Taboada E, Michniewicz J, et al. The
genetic bases for the variation in the lipo-oligosaccharide of the mucosal pathogen,
*Campylobacter jejuni*. Biosynthesis of sialylated ganglioside mimics in the core
oligosaccharide. *J Biol Chem*. 2002;277(1):327–37.
- 23. Heikema AP, Islam Z, Horst-Kreft D, Huizinga R, Jacobs BC, Wagenaar JA, et al.
*Campylobacter jejuni* capsular genotypes are related to Guillain-Barré syndrome. *Clin*
*Microbiol Infect*. 2015;21(9):852.e1-852.e9.

- 24. Koga M, Gilbert M, Takahashi M, Li J, Koike S, Hirata K, et al. Comprehensive analysis
of bacterial risk factors for the development of Guillain-Barré syndrome after
*Campylobacter jejuni* enteritis. *J Infect Dis.* 2006;193(4):547–55.
- 25. Heikema AP, Strepis N, Horst-Kreft D, Huynh S, Zomer A, Kelly DJ, et al. Biomolecule
sulphation and novel methylations related to guillain-barré syndrome-associated
*campylobacter jejuni* serotype hs:19. *Microb Genomics.* 2021;7(11).
- 26. Quino W, Caro-Castro J, Mestanza O, Hurtado V, Zamudio ML, Cruz-Gonzales G, et al.
Emergence and Molecular Epidemiology of *Campylobacter jejuni* ST-2993 Associated
with a Large Outbreak of Guillain-Barré Syndrome in Peru. Denes TG, editor. *Microbiol*
*Spectr* [Internet]. 2022 Oct 26;10(5). Available from:
<https://journals.asm.org/doi/10.1128/spectrum.01187-22>
- 27. Guerry P, Szymanski CM, Prendergast MM, Hickey TE, Ewing CP, Pattarini DL, et al.
Phase variation of *Campylobacter jejuni* 81-176 lipooligosaccharide affects ganglioside
mimicry and invasiveness in vitro. *Infect Immun.* 2002;70(2):787–93.
- 28. Esposito S, Longo MR. Guillain–Barré syndrome. *Autoimmun Rev.* 2017;16(1):96–101.
- 29. Nyati KK, Nyati R. Role of *campylobacter jejuni* infection in the pathogenesis of
Guillain-Barré syndrome: An update. *Biomed Res Int.* 2013;2013.
- 30. Islam Z, Nabila FH, Asad A, Begum R, Jahan I, Hayat S, et al. Draft Genome Sequences
of Three Strains of *Campylobacter jejuni* Isolated from Patients with Guillain-Barré
Syndrome in Bangladesh. Rasko D, editor. *Microbiol Resour Announc* [Internet]. 2021
Apr 29;10(17). Available from: <https://journals.asm.org/doi/10.1128/MRA.00005-21>
- 31. Hayat S, Nabila FH, Asad A, Begum R, Jahan I, Endtz HP, et al. Draft Genome
Sequences of Four Strains of *Campylobacter jejuni* Isolated from Patients with Axonal
Variant of Guillain-Barré Syndrome in Bangladesh. Putonti C, editor. *Microbiol Resour*
*Announc* [Internet]. 2022 Feb 17;11(2). Available from:
<https://journals.asm.org/doi/10.1128/mra.01146-21>
- 32. Heng L, Richard D. Fast and accurate short read alignment with Burrows-Wheeler
Transform. *Bioinformatics.* 2009;25(14):1754–60.
- 33. Li H, Handsaker B, Wysoker A, Fennell T, Ruan J, Homer N, et al. The Sequence
Alignment/Map format and SAMtools. *Bioinformatics* [Internet]. 2009 Aug
15;25(16):2078–9. Available from:
<https://academic.oup.com/bioinformatics/article/25/16/2078/204688>
- 34. Gurevich A, Saveliev V, Vyahhi N, Tesler G. QUAST: Quality assessment tool for
genome assemblies. *Bioinformatics.* 2013;29(8):1072–5.
- 35. Wood DE, Salzberg SL. Kraken: ultrafast metagenomic sequence classification using
exact alignments. *Genome Biol* [Internet]. 2014 Mar 3;15(3):R46. Available from:
<https://genomebiology.biomedcentral.com/articles/10.1186/gb-2014-15-3-r46>
- 36. Tatusova T, DiCuccio M, Badretdin A, Chetvernin V, Nawrocki EP, Zaslavsky L, et al.
NCBI prokaryotic genome annotation pipeline. *Nucleic Acids Res* [Internet]. 2016 Aug
19;44(14):6614–24. Available from: <https://academic.oup.com/nar/article->

lookup/doi/10.1093/nar/gkw569

37. Aziz RK, Bartels D, Best A, DeJongh M, Disz T, Edwards RA, et al. The RAST Server:
Rapid annotations using subsystems technology. *BMC Genomics*. 2008;9.

38. Jolley KA, Bray JE, Maiden MCJ. Open-access bacterial population genomics: BIGSdb
software, the PubMLST.org website and their applications. *Wellcome Open Res*
[Internet]. 2018 Sep 24;3:124. Available from:
<https://wellcomeopenresearch.org/articles/3-124/v1>

39. Li H. New strategies to improve minimap2 alignment accuracy. *Bioinformatics*.
2021;37(23):4572–4.

40. Danecek P, JK B, J L, J M, V O, MO P, et al. Twelve years of SAMtools and BCFtools.
*Gigascience*. 2021;10(2).

41. Hameed A, Woodacre A, Machado LR, Marsden GL. An Updated Classification System
and Review of the Lipooligosaccharide Biosynthesis Gene Locus in *Campylobacter jejuni*
[Internet]. Vol. 11, *Frontiers in Microbiology*. 2020. p. 677. Available from:
<https://www.frontiersin.org/article/10.3389/fmicb.2020.00677>

42. Gilchrist CLM, Chooi YH. Clinker & clustermap.js: Automatic generation of gene cluster
comparison figures. *Bioinformatics*. 2021;37(16):2473–5.

43. Katoh K, Standley DM. MAFFT Multiple Sequence Alignment Software Version 7:
Improvements in Performance and Usability. *Mol Biol Evol* [Internet]. 2013 Apr
1;30(4):772–80. Available from: [https://academic.oup.com/mbe/article-](https://academic.oup.com/mbe/article-lookup/doi/10.1093/molbev/mst010)
[lookup/doi/10.1093/molbev/mst010](https://academic.oup.com/mbe/article-lookup/doi/10.1093/molbev/mst010)

44. Letunic I, Bork P. Interactive Tree Of Life (iTOL) v5: an online tool for phylogenetic tree
display and annotation. *Nucleic Acids Res* [Internet]. 2021 Jul 2;49(W1):W293–6.
Available from: <https://academic.oup.com/nar/article/49/W1/W293/6246398>

45. Garrison E, Marth G. Haplotype-based variant detection from short-read sequencing. 2012
Jul 17; Available from: <http://arxiv.org/abs/1207.3907>

46. Aidley J, Wanford JJ, Green LR, Sheppard SK, Bayliss CD. Phasomeit: An ‘omics’
approach to cataloguing the potential breadth of phase variation in the genus
*campylobacter*. *Microb Genomics*. 2018;4(11).

**Fig. 1.** (A) Phylogenetic tree based on concatenated nucleotide sequence loci of *Campylobacter*
*jejuni* allele from PubMLST database based on 7 MLST loci. The tree also includes metadata
such as clonal complex, year of isolation, LOS class, sialyltransferase genes (*cst-II/cst-III*), and
the 53rd position of *cst-II*. (B) Bar chart of Sequence Type (ST) diversity among GBS and
enteritis-associated *C. jejuni* strains. (C) Bar chart of Clonal complex (CC) diversity among GBS
and enteritis-associated *C. jejuni* strains.

**Fig. 2.** Phylogenetic tree of LOS region of *C. jejuni* strains isolated from GBS and enteritis
patients along with reference strain NCTC11168.

**Fig. 3.** Comparative genome alignment of LOS biosynthesis locus region between GBS (Fig.
3A) and enteritis-associated *C. jejuni* strains (Fig. 3B). LOS region of *C. jejuni* NCTC11168 was
used as the reference genome. Colored arrows indicate the coding sequences and various colour-
reflected homologues groups identified by the clinker. Grey/white arrows represent the
hypothetical proteins. The black and grey bars represent the percentage of amino acid identity as
indicated in the figure.

**Fig. 4.** Distribution of nucleotide variations within the LOS region between GBS-associated and
enteritis-associated *C. jejuni*. (A) Distribution of total nucleotide variations, (B) Distribution of
SNPs, (C) Distribution of MNPs, (D) Distribution of complex (combination of SNP/MNP).

**Fig. 5.** The presence of phase variation within the LOS region of GBS-associated and enteritis-
associated strains.

Spectrum00062-25: Genomic Variability of Lipooligosaccharide Biosynthesis Locus and Sequence Type among *Campylobacter jejuni* Isolated from Patients with Guillain-Barré Syndrome

The authors have made all suggested corrections in the manuscript and in the figures and tables to increase clarity and transparency. The quality of the genomes are now considered good quality now that more details have been provided. However, some components, like the statistical analysis of differences in genomic variations observed between GBS and enteritis cases, have not been explained or supported enough to justify it's acceptance.

Reviewer comments :

1. Thank you for updating Figure 1 with more metadata and including the bar charts as suggested! It makes it immensely easier to analyze and compare results. The only caveat that I have is the overlap between the Disease labels/sequence identifiers and the end of the branches of the tree. Since they are touching, it makes it difficult to see when the sequences are shared on the same branch (when there is a vertical line for the branches for identical ST). I would suggest adding a small gap between the tree and the sequence identifiers, if possible.
2. In the Figure 1 caption, you mention the **53rd** position of cst-II, but in the figure, it's written « cst-II **51st** position ».
3. At the added Lines 347-348, the term « local alignment » is repeated twice, but not necessary. SNIPPY/Snippy should be written in the same way as well.
4. Thank you for your comments concerning Lines 162-168 and the reference genome alignment. So, if the reference genome is LOS class C, then only the genes present in LOS class C would be taken into consideration by Snippy for SNP analysis with all other LOS classes. I understand that some classes resemble LOS class C more than others. For example, perhaps LOS class B has 16 common genes with LOS class C, but LOS class W only has 11 genes in common since many other genes are included in its locus, but seem absent in the LOS class C from the reference according to your Figure 3. You also mention that:

« The LOS region of GBS-associated strains showed a similar pattern of gene alignment, with the majority of them (84%) belonging to LOS classes A, B, and C. Whereas, the LOS regions of enteritis-associated strains exhibited a more varied gene pattern with higher presence of hypothetical proteins compared to GBS-associated strains which belong to seven different LOS classes » (Lines 156-160).

Could this effect or bias your results demonstrated in Figures 4 and 5 (due to less gene alignments compared to the reference for the Enteritis group)? This is why I thought to normalize the number of SNP differences and phase variable regions based on the length of the alignment, as I would expect to find less genetic variation if a smaller genomic region matches the reference genome used to detect such genetic variation (SNP, MNP and complex nucleotide variations; 16 genes vs 11 genes). In other words, if the y-axis in Figures 4 and 5 were « No. of *insert type of variation (SNP, phase variation, etc.)* **per 1 kb of alignment** », would there still be significant differences between GBS and Enteritis groups? This is just a thought/concern, but if you feel that your current analysis is sufficient, keep it that way, but I would suggest justifying why this bias does not exist.

5. I would recommend adding more details in the Figure 2 caption. For example, adding a bit of the methods used here (SNP-based, number of SNPs used, etc.).
6. Thank you for providing the list of phase variable genes in Supplementary Table S3! Is it possible to add the gene names in the table to match the gene names from Figure 3? It would be helpful for readers to connect both items.

Spectrum00062-25R2: Genomic Variability of Lipooligosaccharide Biosynthesis Locus and Sequence Type among *Campylobacter jejuni* Isolated from Patients with Guillain-Barre' Syndrome

Reviewer #1 (Comments for the Author):

Results:

Comment: Minor observations and recommendations on line 114 and 128.

Response: As per recommendation, we have revised line 114 (revised version highlighted: Page: 7; line: 115) and line 128 (revised version highlighted: Page: 7; line: 129).

Discussion:

Comment: Minor observations on lines 203 and 241

Response: We have revised line 203 (revised version highlighted: Page: 11; lines: 203-204) and line 241 (revised version highlighted: Pages: 12-13; lines: 240-243).

Materials and Methods:

Comment: Minor observations on line 303, 309 and 310

Response: We have revised line 303 (revised version highlighted: Page: 15; lines: 302-306), 309 & 310 (revised version highlighted: Page: 16; lines: 310-312)

Reviewer #2 (Comments for the Author):

Comment: *The authors have made all suggested corrections in the manuscript and in the figures and tables to increase clarity and transparency. The quality of the genomes are now considered good quality now that more details have been provided. However, some components, like the statistical analysis of differences in genomic variations observed between GBS and enteritis cases, have not been explained or supported enough to justify its acceptance.*

Response: Thanks for reviewer's very positive comments regarding the revised manuscript. In the revised version of the manuscript, we have also addressed the statistical significance of genomic variations between GBS and enteritis and data interpretation to support the statement.

Reviewer comments:

Comment-1: *Thank you for updating Figure 1 with more metadata and including the bar charts as suggested! It makes it immensely easier to analyze and compare results. The only caveat that I have is the overlap between the Disease labels/sequence identifiers and the end of the branches of the tree. Since they are touching, it makes it difficult to see when the sequences are shared on the same branch (when there is a vertical line for the branches for identical ST). I would suggest adding a small gap between the tree and the sequence identifiers, if possible.*

Response: We appreciate the reviewer's positive remarks about Figure 1. As per suggestion, we have updated **Figure 1** by removing the background colour and adding the possible gap between the tree and the sequence identifiers.

Comment-2: *In the Figure 1 caption, you mention the 53rd position of cst-II, but in the figure, it's written « cst-II 51st position ».*

Response: Apology for typo error. We have corrected this in the revised version. (revised version highlighted: Page: 25; line: 547)

Comment-3: *At the added Lines 347-348, the term « local alignment » is repeated twice, but not necessary. SNIPPY/Snippy should be written in the same way as well.*

Response: We have removed the repeated use of the term "local alignment" and ensured that "Snippy" is written consistently throughout the manuscript. (revised version highlighted: Page: 17; lines: 345-347)

Comment-4: *Thank you for your comments concerning Lines 162-168 and the reference genome alignment. So, if the reference genome is LOS class C, then only the genes present in LOS class C would be taken into consideration by Snippy for SNP analysis with all other LOS classes. I understand that some classes resemble LOS class C more than others. For*

example, perhaps LOS class B has 16 common genes with LOS class C, but LOS class W only has 11 genes in common since many other genes are included in its locus, but seem absent in the LOS class C from the reference according to your Figure 3. You also mention that:

« The LOS region of GBS-associated strains showed a similar pattern of gene alignment, with the majority of them (84%) belonging to LOS classes A, B, and C. Whereas, the LOS regions of enteritis-associated strains exhibited a more varied gene pattern with higher presence of hypothetical proteins compared to GBS-associated strains which belong to seven different LOS classes » (Lines 156-160).

Could this effect or bias your results demonstrated in Figures 4 and 5 (due to less gene alignments compared to the reference for the Enteritis group)? This is why I thought to normalize the number of SNP differences and phase variable regions based on the length of the alignment, as I would expect to find less genetic variation if a smaller genomic region matches the reference genome used to detect such genetic variation (SNP, MNP and complex nucleotide variations; 16 genes vs 11 genes). In other words, if the y-axis in Figures 4 and 5 were « No. of *insert type of variation (SNP, phase variation, etc.)* per 1 kb of alignment », would there still be significant differences between GBS and Enteritis groups? This is just a thought/concern, but if you feel that your current analysis is sufficient, keep it that way, but I would suggest justifying why this bias does not exist.

Response: As per reviewer suggestion, we normalized the number of nucleotide variations based on the length of the alignment, using the metric: “Number of *insert type of variation* per 1 kb of alignment,” to eliminate potential bias. The manuscript has been updated accordingly, and **Figure 4** has been revised to reflect the results of this updated analysis. (revised version highlighted: Pages:14 & 17-18; lines: 164-169 & 349-354; & Fig-4, Supplementary Table S2).

However, we used PhasmeIt, a reference-free tool that detects simple sequence repeats (SSRs), particularly homopolymeric tracts, to identify phase-variable genes in *C. jejuni*. This approach eliminates reference-based bias; therefore, the phase variation profile remains unchanged.

Comment-5: I would recommend adding more details in the Figure 2 caption. For example, adding a bit of the methods used here (SNP-based, number of SNPs used, etc.).

Response: Thank you for your suggestion. We have revised the Figure 2 caption as suggested. (revised version highlighted: Pages: 25; lines: 553-558)

Comment-6: Thank you for providing the list of phase variable genes in Supplementary Table S3! Is it possible to add the gene names in the table to match the gene names from Figure 3? It would be helpful for readers to connect both items.

Response: As per the reviewer's suggestion, we have added the gene names in the Supplementary Table S3 from Figure 3.

Re: Spectrum00062-25R2 (Genomic Variability of Lipooligosaccharide Biosynthesis Locus and Sequence Type among *Campylobacter jejuni* Isolated from Patients with Guillain-Barre´ Syndrome)

Dear Dr. Zahirul Islam:

Your manuscript has been accepted, and I am forwarding it to the ASM production staff for publication. Your paper will first be checked to make sure all elements meet the technical requirements. ASM staff will contact you if anything needs to be revised before copyediting and production can begin. Otherwise, you will be notified when your proofs are ready to be viewed.

Sincerely,
Sadjia Bekal
Editor
Microbiology Spectrum